# How Tight Can PAC-Bayes be in the Small Data Regime?

**Andrew Y. K. Foong**[*]
University of Cambridge
ykf21@cam.ac.uk

**Wessel P. Bruinsma**[*]
University of Cambridge
Invenia Labs
wpb23@cam.ac.uk

**David R. Burt**
University of Cambridge
drb62@cam.ac.uk

**Richard E. Turner**
University of Cambridge
ret26@cam.ac.uk

## Abstract

In this paper, we investigate the question: *Given a small number of datapoints, for example $N = 30$, how tight can PAC-Bayes and test set bounds be made?* For such small datasets, test set bounds adversely affect generalisation performance by withholding data from the training procedure. In this setting, PAC-Bayes bounds are especially attractive, due to their ability to use all the data to simultaneously learn a posterior and bound its generalisation risk. We focus on the case of i.i.d. data with a bounded loss and consider the generic PAC-Bayes theorem of Germain et al. While their theorem is known to recover many existing PAC-Bayes bounds, it is unclear what the tightest bound derivable from their framework is. For a fixed learning algorithm and dataset, we show that the tightest possible bound coincides with a bound considered by Catoni; and, in the more natural case of distributions over datasets, we establish a lower bound on the best bound achievable in expectation. Interestingly, this lower bound recovers the Chernoff test set bound if the posterior is equal to the prior. Moreover, to illustrate how tight these bounds can be, we study synthetic one-dimensional classification tasks in which it is feasible to meta-learn both the prior and the form of the bound to numerically optimise for the tightest bounds possible. We find that in this simple, controlled scenario, PAC-Bayes bounds are competitive with comparable, commonly used Chernoff test set bounds. However, the sharpest test set bounds still lead to better guarantees on the generalisation error than the PAC-Bayes bounds we consider.

## 1 Introduction

Generalisation bounds are of both practical and theoretical importance. Practically, tight bounds provide certificates that algorithms will perform well on unseen data. Theoretically, the bounds and underlying proof techniques can help explain the phenomenon of learning. Among the tightest known bounds are *PAC-Bayes* (McAllester, 1999) and *test set* bounds (Langford, 2002). In this paper, we investigate their numerical tightness when applied to small datasets ($N \approx 30$–$60$ datapoints). The comparison between PAC-Bayes and test set bounds is particularly interesting in this setting as one cannot discard data to compute a test set bound without significantly harming post-training performance due to a reduced training set size. PAC-Bayes on the other hand provides valid bounds while using all of the data for learning, since it provides bounds that hold uniformly. The small

---

[*]Equal contribution.

35th Conference on Neural Information Processing Systems (NeurIPS 2021).

data setting can also be quite different from the big data setting, as lower-order terms in PAC-Bayes bounds have a non-negligible contribution, and the detailed structure of the bound becomes important.

Fortunately, we do not have to study each PAC-Bayes bound separately: remarkably, Germain et al. (2009) showed that a wide range of bounds can be obtained as special cases of a single *generic PAC-Bayes theorem* that captures the central ideas of many PAC-Bayes proofs (see also Bégin et al. (2016)). This theorem has a free parameter: it holds for any convex function, $\Delta$. By choosing $\Delta$ appropriately, one can recover the well-known bounds of Langford and Seeger (2001), Catoni (2007) and other bounds. We focus on two questions related to this set-up. *First, what is the tightest bound achievable by any convex function $\Delta$?* An answer would characterise the limits of the generic PAC-Bayes theorem, and thereby of a wide range of bounds, by telling us how much improvement could be obtained before new ideas or assumptions are needed. *Second*, since test set bounds are the *de facto* standard for larger datasets, but PAC-Bayes has benefits when $N$ is small, we ask: *in the small data regime, can PAC-Bayes be tighter than test set bounds?*

In Section 3, Theorem 4, we show that in the (artificial) case when $\Delta$ can be chosen depending on the dataset (without taking a union bound), the tightest version of the generic PAC-Bayes theorem is obtained by one of the *Catoni bounds* (Catoni, 2007). In the more realistic case when $\Delta$ must be chosen *before* sampling the dataset, we do not fully characterise the tightest bound, but in Corollary 3 we *lower bound* the tightest bound achievable (in expectation) with any $\Delta$. We also provide numerical evidence in Figure 2 that suggests this lower bound can in some cases be attained, by flexibly parameterising a convex function $\Delta$ with a constrained neural network. Interestingly, this lower bound coincides with removing a lower-order term from the Langford and Seeger (2001) bound (something that Langford (2002) conjectured was possible), and relaxes to the well-known *Chernoff test set bound* (see Theorem 2 below) when the PAC-Bayes posterior is equal to the prior.

In Section 4, we investigate the tightness of PAC-Bayes and test set bounds in synthetic 1D classification. The goal of this experiment is to find out how tight the bounds could be made in principle. We use meta-learning to adapt all aspects of the bounds and learning algorithms, producing meta-learners that are trained to optimise the value of the bounds on this task distribution. We find that, in this setting, PAC-Bayes can be competitive with the Chernoff test set bound, but is outperformed by the *binomial tail test set bound*, of which the Chernoff bound is a relaxation. This suggests that, for standard PAC-Bayes to be quantitatively competitive with the best test set bounds on small datasets, a new proof technique leading to bounds that gracefully relax to the binomial tail bound is required. Code to reproduce all experiments can be found at `https://github.com/cambridge-mlg/pac-bayes-tightness-small-data`.

## 2 Background and Related Work

We consider supervised learning. Let $\mathcal{X}$ and $\mathcal{Y}$ denote the *input space* and *output space*, and let $\mathcal{Z} = \mathcal{X} \times \mathcal{Y}$. Assume there is an (unknown) probability measure[2] $D$ over $\mathcal{Z}$, with the dataset $S \sim D^N$. Denote the *hypothesis space* by $\mathcal{H} \subseteq \mathcal{Y}^{\mathcal{X}}$. A learning algorithm is then a map $\mathcal{Z}^N \to \mathcal{H}$. In PAC-Bayes, we also consider maps $\mathcal{Z}^N \to \mathcal{M}_1(\mathcal{H})$, where $\mathcal{M}_1$ is the set of probability measures on its argument. The performance of a hypothesis $h \in \mathcal{H}$ is measured by a *loss function* $\ell\colon \mathcal{Z} \times \mathcal{H} \to [0,1]$. The *(generalisation) risk* of $h$ is $R_D(h) \coloneqq \mathbb{E}_{(x,y)\sim D}[\ell((x,y),h)]$ and its *empirical risk on $S$* is $R_S(h) \coloneqq \frac{1}{N}\sum_{(x,y)\in S}\ell((x,y),h)$. For $Q \in \mathcal{M}_1(\mathcal{H})$ its *(generalisation Gibbs) risk* is $\overline{R}_D(Q) \coloneqq \mathbb{E}_{h\sim Q}[R_D(h)]$ and its *empirical (Gibbs) risk* is $\overline{R}_S(Q) \coloneqq \mathbb{E}_{h\sim Q}[R_S(h)]$. In PAC-Bayes, we usually fix a *prior* $P \in \mathcal{M}_1(\mathcal{H})$, chosen without reference to $S$ and learn a *posterior* $Q \in \mathcal{M}_1(\mathcal{H})$ which can depend on $S$. The *KL-divergence* between $Q$ and $P$ is defined as $\mathrm{KL}(Q\|P) = \int \log \frac{\mathrm{d}Q}{\mathrm{d}P}\,\mathrm{d}Q$ if $Q \ll P$ and $\infty$ otherwise. Let $\mathcal{C}$ denote the set of proper, convex, lower semicontinuous (l.s.c.) functions $\mathbb{R}^2 \to \mathbb{R} \cup \{+\infty\}$; if a convex function's domain is a subset of $\mathbb{R}^2$, extend it to all of $\mathbb{R}^2$ with the value $+\infty$. See Appendix C for more details on convex analysis, which we use in Section 3.

**Test Set Bounds.** *Test set bounds* rely on a subset of data which is not used to select the hypothesis, called a *test set* or *held-out set*. Let $S = S_{\mathrm{train}} \cup S_{\mathrm{test}}$, with $|S| = N$, $|S_{\mathrm{train}}| = N_{\mathrm{train}}$ and $|S_{\mathrm{test}}| = N_{\mathrm{test}}$. In Theorems 1 and 2, we assume $h$ is chosen independently of $S_{\mathrm{test}}$. For the zero-one loss, $\ell((x,y),h) \coloneqq \mathbb{1}[h(x) \neq y]$, we have that $N_{\mathrm{test}}R_{S_{\mathrm{test}}}(h)$ is a binomial random variable with parameters $(N_{\mathrm{test}}, R_D(h))$. This leads to the following simple bound, which, for $\ell \in \{0,1\}$, is tight among test set bounds:

---

[2]We will colloquially refer to measures on sets without specifying a $\sigma$-algebra. We implicitly assume functions are measurable with respect to the $\sigma$-algebras on which the relevant measures are defined.

**Theorem 1** (Binomial tail test set bound, Langford, 2005, Theorem 3.3)**.**
Let $\overline{e}(M, k, \delta) := \sup \left\{ p : \delta \leq \sum_{i=0}^{k} \binom{M}{i} p^i (1-p)^{M-i} \right\}$. For any $h \in \mathcal{H}$, $\ell \in \{0, 1\}$ and $\delta \in (0, 1)$,

$$\Pr\left( R_D(h) \leq \overline{e}(N_{\text{test}}, N_{\text{test}} R_{S_{\text{test}}}(h), \delta) \right) \geq 1 - \delta. \tag{1}$$

Often, looser bounds with a simpler form are applied. These can be obtained via the Chernoff method:

**Theorem 2** (Chernoff test set bound, Langford, 2005, Corollary 3.7)**.**
For $q, p \in [0, 1]$, let $\mathrm{kl}(q, p) := q \log \frac{q}{p} + (1-q) \log \frac{1-q}{1-p}$. For any $h \in \mathcal{H}$, $\ell \in [0, 1]$, and $\delta \in (0, 1)$,

$$\Pr\left( \mathrm{kl}(R_{S_{\text{test}}}(h), R_D(h)) \leq \frac{1}{N_{\text{test}}} \log \frac{1}{\delta} \right) \geq 1 - \delta. \tag{2}$$

**PAC-Bayes Bounds.** The PAC-Bayes approach bounds the generalisation Gibbs risk of *stochastic classifiers*, and does not require discarding data, as all the data can be used to choose the posterior, while still obtaining a valid generalisation bound. Since the seminal paper of McAllester (1999), a large variety of PAC-Bayes bounds have been derived. Germain et al. (2009) prove a very general form of the PAC-Bayes theorem which encompasses many of these (see also Bégin et al. (2016) and Rivasplata et al. (2020)). Their proof technique consists of a series of inequalities shared by PAC-Bayes proofs (Jensen's, change of measure, Markov's, supremum over risk[3]), and reveals their common structure. Thus understanding the properties of this generic theorem can give insight into many PAC-Bayes bounds at once:

**Theorem 3** (Generic PAC-Bayes theorem, Bégin et al. (2016) and Germain et al. (2009))**.**[4] *Fix $P \in \mathcal{M}_1(\mathcal{H})$, $\ell \in [0, 1]$, $\delta \in (0, 1)$, and $\Delta$ a proper, convex, l.s.c. function $[0, 1]^2 \to \mathbb{R} \cup \{+\infty\}$. Then*

$$\Pr\left( (\forall Q) \; \Delta(\overline{R}_S(Q), \overline{R}_D(Q)) \leq \frac{1}{N} \left[ \mathrm{KL}(Q\|P) + \log \frac{\mathcal{I}_\Delta(N)}{\delta} \right] \right) \geq 1 - \delta, \tag{3}$$

*where $\mathcal{I}_\Delta(N) := \sup_{r \in [0,1]} \sum_{k=0}^{N} \binom{N}{k} r^k (1-r)^{N-k} e^{N\Delta(k/N, r)}$.*

**Remark 1.** *We lose no generality in assuming $\Delta(q, \cdot)$ is monotonically increasing for all $q \in [0, 1]$, i.e. for any convex $\Delta$ we can define a $\Delta'$ that is monotonically increasing in its second argument and produces a bound that is at least as tight as the bound produced by $\Delta$. See Appendix D for a proof.*

Note that the PAC-Bayes bound holds simultaneously for all posteriors $Q$, and hence is valid even when $Q$ is chosen by minimising the bound. For completeness, we provide a proof of Theorem 3 in Appendix B. Following Germain et al. (2009), we briefly recap some of the bounds that can be recovered as special cases (or looser versions) of Theorem 3. Setting $\Delta(q, p) = C_\beta(q, p) := -\log(1 + p(e^{-\beta} - 1)) - \beta q$ for $\beta > 0$, we recover the *Catoni bounds*:

**Corollary 1** (Catoni, 2007, Theorem 1.2.6)**.** *For any $\beta > 0$,*

$$\Pr\left( (\forall Q) \; \overline{R}_D(Q) \leq \frac{1}{1 - e^{-\beta}} \left[ 1 - \exp\left( -\beta \overline{R}_S(Q) - \frac{1}{N} \left( \mathrm{KL}(Q\|P) + \log \frac{1}{\delta} \right) \right) \right] \right) \geq 1 - \delta. \tag{4}$$

This specifies a bound for every value of $\beta > 0$. If we instead choose $\Delta(q, p) = \mathrm{kl}(q, p)$, we obtain the bound of Langford and Seeger (2001), also called the *PAC-Bayes-kl bound*, but with the slightly sharper dependence on $N$ established by Maurer (2004):

**Corollary 2** (Langford and Seeger, 2001, Theorem 3, Maurer, 2004, Theorem 5)**.**

$$\Pr\left( (\forall Q) \; \mathrm{kl}(\overline{R}_S(Q), \overline{R}_D(Q)) \leq \frac{1}{N} \left[ \mathrm{KL}(Q\|P) + \log \frac{2\sqrt{N}}{\delta} \right] \right) \geq 1 - \delta. \tag{5}$$

Corollary 2 is actually very slightly looser than Theorem 3 with $\Delta = \mathrm{kl}$, since Maurer (2004) upper bounds $\mathcal{I}_{\mathrm{kl}}(N)$ by $2\sqrt{N}$ using Stirling's formula.[5] The Catoni and PAC-Bayes-kl bounds are among the tightest PAC-Bayes bounds known and have been applied in settings where numerical tightness is key, such as obtaining generalisation bounds for stochastic neural networks (Dziugaite & Roy, 2017; Zhou et al., 2019). Many other bounds can be obtained by loosening these bounds. Applying Pinsker's inequality $\mathrm{kl}(q, p) \geq 2(q - p)^2$ to Equation (5) yields the "square-root" version of the

---

[3]The supremum over risk step was introduced in Bégin et al. (2016), although for certain $\Delta$ it can be omitted.
[4]We state a simpler version of their result WLOG, absorbing a free parameter into the function $\Delta$.
[5]Maurer (2004) only proves this bound for $N \geq 8$, but the cases where $1 \leq N \leq 7$ can be easily verified numerically (Germain et al., 2015, Lemma 19).

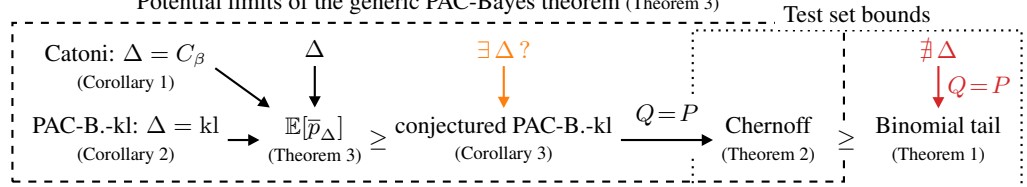

Figure 1: **Illustration of the relationship between various PAC-Bayes and test set bounds**; see Section 3. It is unclear if there always exists a $\Delta$ that recovers the conjectured PAC-Bayes-kl bound (and hence the Chernoff bound when $Q = P$; see Open Problem 1), but there certainly does not exist a $\Delta$ that recovers the Binomial tail bound when $Q = P$.

PAC-Bayes theorem (McAllester, 1999, 2003). The "PAC-Bayes-$\lambda$" (Thiemann et al., 2017) and "PAC-Bayes-quadratic" bounds (Rivasplata et al., 2019) can be derived as loosened versions of the PAC-Bayes-kl bound using the inequality $\mathrm{kl}(q, p) \geq (q - p)^2/(2p)$, valid for $q < p$. The "linear" bound in McAllester (2013) can be derived by loosening the Catoni bound using: $C_\beta(q, p) \leq A \implies p \leq \frac{1}{1 - \beta/2}(q + \frac{1}{\beta}A)$, which is valid for $\beta \in (0, 2)$.

**How Tight Are PAC-Bayes Bounds?** A fundamental question we can ask about a generalisation bound is how tight it is, and whether it can be tightened. Comparing the PAC-Bayes-kl and Chernoff test set bounds when $Q = P$ (so the PAC-Bayes bound essentially becomes a test set bound) shows they are identical except for a $\log(2\sqrt{N})/N$ on the RHS of the PAC-Bayes-kl bound. Whether this term (or similar discrepancies between PAC-Bayes and *Occam bounds* (Langford, 2002, Corollary 4.6.2); see Appendix A) can be removed has been an open question since Langford (2002, Problem 6.1.2). Maurer (2004) reduced this term to its current form, improving on work by Langford and Seeger (2001). Interestingly, Germain et al. (2009, Proposition 2.1) shows that the expression obtained by dropping $\log(2\sqrt{N})/N$ from the PAC-Bayes-kl bound is identical to that obtained by *illegally*[6] minimising the Catoni bound with respect to $\beta$; Catoni (2007, Theorem 1.2.8) shows that a union bound can be used to, in a legal way, approximately optimise with respect to $\beta$ at the cost of an additional lower order term. The Chernoff test set bound is itself a looser version of the binomial tail bound, raising the question of whether a PAC-Bayes bound can be found that reduces to the binomial tail bound when $Q = P$. We provide new insights into these problems in Section 3.

Researchers have also compared PAC-Bayes bounds numerically on actual learning problems. Langford (2005) and Germain et al. (2009) were able to obtain reasonable guarantees on small datasets. However, Langford (2005) found that on datasets with $N \approx 145$, PAC-Bayes was outperformed by test set bounds. Dziugaite and Roy (2017), Langford and Caruana (2001), and Pérez-Ortiz et al. (2021) provide non-vacuous bounds for neural networks using PAC-Bayes. Even so, Dziugaite et al. (2021) states that tighter bounds would be obtained using a test set instead. In Section 4 we find that if the bounds and learning algorithms are optimised for a task distribution, PAC-Bayes can be tight enough to compete with the Chernoff test set bound, but not the binomial tail test set bound.

## 3 Characterising the Limits of the Generic PAC-Bayes Proof Technique

This section establishes our main theoretical contributions, which characterise the limits of the generic PAC-Bayes theorem (Theorem 3). For a convex $\Delta \in \mathcal{C}$, Theorem 3 gives a high-probability upper bound on $\Delta(\overline{R}_S(Q), \overline{R}_D(Q))$. Define $B[f, y] \coloneqq \sup\{p \in [0, 1] : f(p) \leq y\}$ for $f \colon [0, 1] \to \mathbb{R}$ and $y \in \mathbb{R}$, where we take $\sup \varnothing = 1$. This upper bound (Theorem 3) can be "inverted" to obtain a high-probability upper bound on $\overline{R}_D(Q)$: with probability at least $1 - \delta$, for all $Q \in \mathcal{M}_1(\mathcal{H})$,

$$\overline{R}_D(Q) \leq \overline{p}_\Delta \quad \text{where} \quad \overline{p}_\Delta \coloneqq B\left[\Delta(\overline{R}_S(Q), \cdot), \frac{1}{N}\left(\mathrm{KL}(Q\|P) + \log\frac{\mathcal{I}_\Delta(N)}{\delta}\right)\right]. \tag{6}$$

Since (6) holds for all $\Delta \in \mathcal{C}$, a natural question is: *Which $\Delta$ minimises $\overline{p}_\Delta$?* This would characterise how tight, numerically, PAC-Bayes theorems can be made without introducing ideas beyond those needed to prove the bounds stated in Section 2. Before considering the case when $\Delta$ is selected before observing $S \sim D^N$, we first characterise the optimal $\Delta$ in the simplified scenario where $\Delta$ can depend on the dataset $S$ and the posterior $Q$ (Theorem 4). This setting is artificial, since choosing

---

[6]That is, optimising $\beta$ depending on the dataset $S$ without taking a union bound.

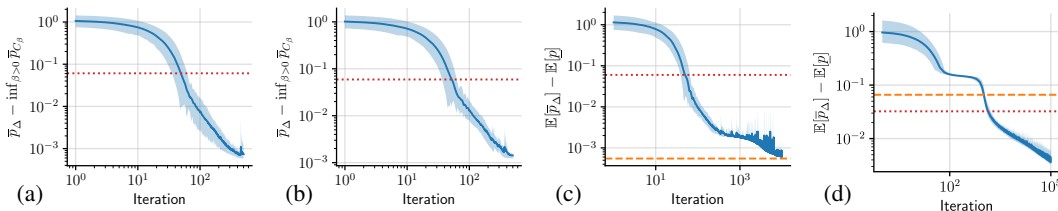

Figure 2: **The tightest Catoni bound is the optimal generic PAC-Bayes bound for a fixed dataset, but not the optimal expected bound for a random dataset.** We optimise a convex function with $H$ hidden units to minimise $\overline{p}_\Delta$ with $\delta = 0.1, N = 30$. **(a) and (b) consider fixed** $q$ **and** KL (precise values below) and show the difference with the best Catoni bound (Theorem 4); **(c) and (d) consider random** $q$ **and** KL and show the *expected* difference with the conjectured PAC-Bayes-kl bound (Corollary 3). Shaded regions show the minimum and maximum over ten initialisations. All plots show the PAC-Bayes-kl bound (dotted red) and (c) and (d) show the optimal Catoni bound with parameter $\beta^*$ (dashed orange). All runs quickly converged to non-vacuous values. (a): $(q, \mathrm{KL}) = (2\%, 1)$, $\beta^* \approx 2.24$, $H = 256$. (b): $(q, \mathrm{KL}) = (5\%, 2)$, $\beta^* \approx 1.84$, $H = 256$. (c): $(q, \mathrm{KL}) \in \{(2\%, 1), (5\%, 2)\}$ uniformly, $\beta^* \approx 1.99$, $H = 512$. (d): $(q, \mathrm{KL}) \in \{(30\%, 1), (40\%, 50)\}$ uniformly, $\beta^* \approx 2.32$, $H = 1024$.

$\Delta$ based on $S$ (without taking a union bound) does not yield a valid generalisation bound. However, using Theorem 4 as a building block, we later derive a *lower bound* on the best possible generic PAC-Bayes bound (in expectation) in the more realistic case when we cannot choose $\Delta$ based on $S$ (Corollary 3). We then connect this lower bound to various existing PAC-Bayes and test set bounds. An overview is shown in Figure 1. We now state our first result:

**Theorem 4.** *Given any fixed dataset $S$ and any $Q, P \in \mathcal{M}_1(\mathcal{H})$, the tightest Catoni bound is as tight as the tightest bound possible within the generic PAC-Bayes theorem (Theorem 3). Precisely, let $\Delta \in \mathcal{C}$ and $\delta \in (0, 1)$. Choose some fixed values for $\overline{R}_S(Q) =: q \in [0, 1]$ and $\mathrm{KL}(Q\|P) =: \mathrm{KL} \in [0, \infty)$. If $q > 0$, then there exists a $\beta \in (0, \infty)$ such that $\overline{p}_\Delta \geq \overline{p}_{C_\beta}$, where $\overline{p}$ is defined in Equation (6). Moreover, if $q = 0$, then $\overline{p}_\Delta \geq \lim_{\beta \to \infty} \overline{p}_{C_\beta}$.*

**Remark 2.** *By Theorem 4, for all $\Delta \in \mathcal{C}$, we have $\overline{p}_\Delta \geq \inf_{\beta > 0} \overline{p}_{C_\beta}$, and, by Proposition 2.1 of Germain et al. (2009), $\inf_{\beta > 0} \overline{p}_{C_\beta} = B[\mathrm{kl}(q, \cdot), \frac{1}{N}(\mathrm{KL} + \log \frac{1}{\delta})]$. Hence, for all $\Delta \in \mathcal{C}$, it holds that $\overline{p}_\Delta \geq B[\mathrm{kl}(q, \cdot), \frac{1}{N}(\mathrm{KL} + \log \frac{1}{\delta})]$, which is also shown directly in the proof of Theorem 4 (Equation (17)). Note that optimising $\beta$ in this way is illegal in the general case when the dataset $S$ (and hence $q$ and $\mathrm{KL}$) is stochastic, and would typically require a union bound to be valid.*

We defer the proof of Theorem 4 to the end of this section. We numerically verify Theorem 4 by optimising $\overline{p}_\Delta$ with respect to an arbitrary convex $\Delta$ for various settings of fixed $q$ and KL. To parametrise a convex $\Delta$, we use a one-hidden-layer neural network with positive weights at the output layer and softplus nonlinearities. The inversion performed by $B$ is approximated numerically by discretising the second argument of $\Delta$ and detecting an upcrossing. Gradients are then approximated using the inverse function theorem: $\frac{\mathrm{d}}{\mathrm{d}\theta} B[f_\theta, c(\theta)] = (\partial_\theta c(\theta) - \partial_\theta f_\theta(x))/\partial_x f_\theta(x)$. See Appendix F for details.[7] Figures 2a and 2b show the difference between the numerically optimised $\Delta$ and the best Catoni bound for two settings of fixed $q \in [0, 1]$ and $\mathrm{KL} \in [0, \infty)$. In both cases, $\overline{p}_\Delta - \inf_{\beta > 0} \overline{p}_{C_\beta}$ appears to converge to zero from above, as expected from Theorem 4. Interestingly, Appendix F shows that the learned $\Delta$ can deviate substantially from $C_\beta$, suggesting that there are choices for $\Delta$ besides Catoni's which achieve $\inf_{\Delta \in \mathcal{C}} \overline{p}_\Delta$.

For any *fixed* dataset $S$, Theorem 4 states that the tightest bound is one of the Catoni bounds; precisely: $\inf_{\Delta \in \mathcal{C}} \overline{p}_\Delta = \inf_{\beta > 0} \overline{p}_{C_\beta}$. Note that the optimal value of $\beta$ may depend on the dataset $S$. The more interesting question is whether, when $S \sim D^N$ is sampled randomly, one of the Catoni bounds can still achieve the tightest bound (in expectation) for a *single* value of $\beta$ that is chosen *before* sampling $S$. The answer is no: Figure 2d gives a numerical counterexample where $\inf_{\Delta \in \mathcal{C}} \mathbb{E}[\overline{p}_\Delta] < \mathbb{E}[\overline{p}_{\mathrm{kl}}] < \inf_{\beta > 0} \mathbb{E}[\overline{p}_{C_\beta}]$. Since the Catoni family of bounds cannot generally achieve the tightest bound in expectation, which $\Delta$ do? And how tight is $\inf_{\Delta \in \mathcal{C}} \mathbb{E}[\overline{p}_\Delta]$? Whilst we do not have a full answer, we establish a simple lower bound on $\inf_{\Delta \in \mathcal{C}} \mathbb{E}[\overline{p}_\Delta]$. Define the *conjectured*

---

[7]Numerical inversion of $\Delta$ when $\Delta = \mathrm{kl}$ has been considered by many authors, including Dziugaite and Roy (2017) who use Newton's method and Majumdar and Goldstein (2018) who propose using convex optimisation methods. However, to our knowledge, the specific inversion algorithm we propose for general convex $\Delta$, along with the method for backpropagating through the inverse, are novel in the PAC-Bayes setting.

*PAC-Bayes-kl bound $\underline{p}$* as the quantity from Remark 2, which equals the PAC-Bayes-kl bound *without the $\frac{1}{N} \log \mathcal{I}_{\mathrm{kl}}(N)$ term* on the RHS:

$$\underline{p} := B[\mathrm{kl}(\overline{R}_S(Q), \cdot\,), \tfrac{1}{N}(\mathrm{KL}(Q\|P) + \log \tfrac{1}{\delta})]. \tag{7}$$

The conjectured PAC-Bayes-kl bound has *not* been proven to be a valid generalisation bound. When $S \sim D^N$ is random, Remark 2 tells us that $\inf_{\Delta \in \mathcal{C}} \overline{p}_\Delta = \underline{p}$ a.s. Taking expectations and interchanging the expectation and infimum yields the following corollary:

**Corollary 3.** *Consider the setting from Theorem 3. Then the expected conjectured PAC-Bayes-kl bound $\mathbb{E}[\underline{p}]$ gives a lower bound on all expected generalisation bounds obtained through the generic PAC-Bayes theorem (Theorem 3). That is, for any distribution over datasets, any prior, and any learning algorithm,*

$$\inf_{\Delta \in \mathcal{C}} \mathbb{E}[\overline{p}_\Delta] \geq \mathbb{E}[\underline{p}] \tag{8}$$

*Moreover, there exists a distribution over datasets, a prior, and a posterior such that equality holds. For example, let $(x, y)$ be constant almost surely, which reduces to the setting of Theorem 4. Note that in (8), $\Delta$ is chosen* not *depending on $S$, which leads to a valid generalisation bound on the LHS.*

Figure 1 shows how Corollary 3 fits into the picture so far. The conjectured PAC-Bayes-kl bound is at least as tight as the bound achieved by any $\Delta$, but Corollary 3 does not establish the existence of a $\Delta$ which achieves it. Corollary 3 has practical utility: the conjectured PAC-Bayes-kl bound can be used to prove optimality of a choice of $\Delta$. Specifically, if a practitioner computes a valid bound based on the generic PAC-Bayes theorem, and finds that it is close to the conjectured PAC-Bayes-kl bound, they can be assured by Corollary 3 that they would not have gotten a much better bound (in expectation) with any other choice of $\Delta$. Conversely, the conjectured PAC-Bayes-kl bound can quantify potential slack in the bound due to a suboptimal choice of $\Delta$. Appendix H considers an example of this application of Corollary 3 in the simplified scenario where $\overline{R}_S(Q) = \frac{1}{2}$ almost surely.

The conjectured PAC-Bayes-kl bound also recovers the Chernoff test set bound (Theorem 2) when setting $Q = P$. Since the binomial tail bound (Theorem 1) is strictly tighter than the Chernoff bound, this shows there does not exist a $\Delta$ such that the generic PAC-Bayes bound (Theorem 3) recovers the Binomial tail bound when $Q = P$; this is illustrated in Figure 1. What is unclear, however, is whether there always exists a $\Delta$ such that Theorem 3 recovers the Chernoff test set bound; or, alternatively, such that the conjectured PAC-Bayes-kl bound is attained. A positive answer to the latter would establish that the conjectured PAC-Bayes-kl bound is a valid generalisation bound.[8] As a first piece of evidence, the traces from Figures 2c and 2d suggest that a convex function could actually achieve $\mathbb{E}[\underline{p}]$; see Appendix G for more traces. We leave a full resolution of this question as an open problem; see Section 5. Interestingly, Figure 2c shows that a Catoni bound is sometimes *nearly* optimal even in the stochastic case; we will see another example of this in Figure 3.

We end this section with the proof of Theorem 4. Recall that the Catoni family of bounds follows from Theorem 3 by considering $\Delta(q, p) = C_\beta(q, p) := \mathcal{F}_\beta(p) - \beta q$ with $\mathcal{F}_\beta(p) := -\log(p(e^{-\beta} - 1) + 1)$ and $\beta > 0$. To simplify the notation, we denote $\alpha = \frac{1}{N}(\mathrm{KL} + \log \frac{1}{\delta}) \in (0, \infty)$.

*Proof of Theorem 4.* The proof proceeds in three steps. In the first two steps, we lower bound $\frac{1}{N} \log \mathcal{I}_\Delta(N)$ and upper bound $\Delta$. In the third step, we use these bounds to lower bound $B[\Delta(q, \cdot\,), \alpha + \frac{1}{N} \log \mathcal{I}_\Delta(N)]$ and identify the result with a particular Catoni bound.

**Lower bound on $\frac{1}{N} \log \mathcal{I}_\Delta(N)$:** Since $\Delta \in \mathcal{C}$, it is equal to its own double convex conjugate: $\Delta(q, p) = \Delta^{**}(q, p) = \sup_{c_q, c_p \in \mathbb{R}} (c_q q + c_p p - \Delta^*(c_q, c_p))$, where $^*$ denotes convex conjugation. Let $X \sim \mathrm{Bin}(r, N)$. Then

$$\mathcal{I}_\Delta(N) = \sup_{r \in [0,1]} \mathbb{E}[e^{N\Delta(X/N, r)}] = \sup_{r \in [0,1]} \mathbb{E}[e^{\sup_{c_q, c_p \in \mathbb{R}}(c_q X + N c_p r - N\Delta^*(c_q, c_p))}] \tag{9}$$

$$\geq \sup_{r \in [0,1]} \sup_{c_q, c_p \in \mathbb{R}} e^{N c_p r - N\Delta^*(c_q, c_p)} \mathbb{E}[e^{c_q X}] \tag{10}$$

where $\mathbb{E}[e^{c_q X}] = (r(e^{c_q} - 1) + 1)^N$ is the moment-generating function of $X$. Consequently, taking log, dividing by $N$, and noting that $\frac{1}{N} \log \mathbb{E}[e^{c_q X}] = -\mathcal{F}_{-c_q}(r)$,

$$\frac{1}{N} \log \mathcal{I}_\Delta(N) \geq A \quad \text{where} \quad A := \sup_{c_q, c_p \in \mathbb{R}} [-\Delta^*(c_q, c_p) + \sup_{r \in [0,1]} (c_p r - \mathcal{F}_{-c_q}(r))]. \tag{11}$$

---

[8] By Remark 2, $\overline{p}_\Delta \geq \underline{p}$, so $\mathbb{E}[\overline{p}_\Delta] = \mathbb{E}[\underline{p}]$ implies that $\overline{p}_\Delta = \underline{p}$ a.s., meaning that $\underline{p}$ is a valid gen. bound.

**Upper bound on $\Delta$:** We upper bound $\Delta$ by making $\Delta^*$ as small as possible without exceeding the supremum from (11). Note that $A$ is finite, because $\Delta^*$ is proper. Define $\tilde{\Delta}^*$ as follows: $\tilde{\Delta}^*(c_q, c_p) = -A + \sup_{r \in [0,1]}(c_p r - \mathcal{F}_{-c_q}(r))$. Note that $\tilde{\Delta}^*$ is proper, convex as a pointwise supremum of convex functions, and l.s.c. as a supremum of l.s.c. functions. In fact, $\tilde{\Delta}^*$ is finite for all inputs. As the notation suggests, define $\tilde{\Delta} := (\tilde{\Delta}^*)^*$. Then $\tilde{\Delta}^*$ is indeed the convex conjugate of $\tilde{\Delta}$, because $\tilde{\Delta}^* \in \mathcal{C}$, so it is equal to its own double convex conjugate. Moreover,

$$\tilde{\Delta}(q, p) = A + \sup_{c_q, c_p \in \mathbb{R}}[c_q q + c_p p - \sup_{r \in [0,1]}(c_p r - \mathcal{F}_{-c_q}(r))] \tag{12}$$

$$= A + \sup_{c_q \in \mathbb{R}}[c_q q + \sup_{c_p \in \mathbb{R}}[c_p p - \mathcal{F}^*_{-c_q}(c_p)]] \tag{13}$$

$$= A + \sup_{c_q \in \mathbb{R}}[c_q q + \mathcal{F}_{-c_q}(p)], \tag{14}$$

by observing that $p \mapsto \mathcal{F}_{-c_q}(p) \in \mathcal{C}$, so it is equal to its own double convex conjugate. Therefore,

$$\tilde{\Delta}(q, p) = A + \sup_{c_q \in \mathbb{R}} C_{-c_q}(q, p) \overset{\text{(i)}}{=} A + \mathrm{kl}(q, p) \tag{15}$$

where (i) follows from a direct computation; see Lemma E.1 (Appendix E). *Claim:* For all $q, p \in [0, 1]$, $\tilde{\Delta}(q, p) \geq \Delta(q, p)$. This follows from the definitions and finiteness of $\tilde{\Delta}^*$ and $A$: for all $c_q, c_p \in \mathbb{R}$,

$$-\tilde{\Delta}^*(c_q, c_p) + \sup_{r \in [0,1]}(c_p r - \mathcal{F}_{-c_q}(r)) = A \geq -\Delta^*(c_q, c_p) + \sup_{r \in [0,1]}(c_p r - \mathcal{F}_{-c_q}(r)), \tag{16}$$

which means that $\tilde{\Delta}^* \leq \Delta^*$, so $\tilde{\Delta} \geq \Delta$ by the order-reversing property of the convex conjugate.

**Conclusion:** Assume that $\overline{p}_\Delta < 1$; otherwise, any $\beta > 0$ works. To begin with, use the previous steps:

$$\overline{p}_\Delta = B[\Delta(q, \cdot), \alpha + \tfrac{1}{N} \log \mathcal{I}_\Delta(N)] \overset{\text{(11), claim}}{\geq} B[\tilde{\Delta}(q, \cdot), \alpha + A] \overset{\text{(15)}}{=} B[\mathrm{kl}(q, \cdot), \alpha] = \underline{p}. \tag{17}$$

Since $\alpha > 0$, clearly $\underline{p} > q$, so $0 \leq q < \underline{p} < 1$. Hence, if $q > 0$, then there exists a $\beta > 0$ such that $\mathrm{kl}(q, \underline{p}) = C_\beta(q, \underline{p})$ (Lemma E.2; Appendix E). Using that $p \mapsto C_\beta(q, p)$ is continuous and strictly increasing for all $\beta > 0$, we have that $\underline{p} = B[C_\beta(q, \cdot), \alpha]$, so

$$\overline{p}_\Delta \geq B[C_\beta(q, \cdot), \alpha] \overset{\text{(i)}}{=} B[C_\beta(q, \cdot), \alpha + \tfrac{1}{N} \log \mathcal{I}_{C_\beta}(N)] = \overline{p}_{C_\beta}, \tag{18}$$

where (i) uses that $\tfrac{1}{N} \log \mathcal{I}_{C_\beta}(N) = 0$ (Lemma E.3; Appendix E). If $q = 0$, then $\mathrm{kl}(0, \underline{p}) = \lim_{\beta \to \infty} C_\beta(0, \underline{p})$ (Lemma E.2; Appendix E), so $\overline{p}_\Delta \geq B[\lim_{\beta \to \infty} C_\beta(0, \cdot), \alpha]$, and conclude like in (18) using Lemma E.4 (Appendix E). $\qquad\square$

# 4  Meta-Learning the Tightest Bounds for Synthetic Classification

We now consider, for a particular distribution over tasks, how tight each bound can be made in expectation. Two questions naturally arise: *Which PAC-Bayes bounds are tightest?* and *Can PAC-Bayes bounds be tighter than test set bounds?* While test set bounds have traditionally been considered tighter than PAC-Bayes bounds, here we work in the small data regime where a substantial proportion of the data must be removed to form a test set, which could impact generalisation performance and hence lead to worse bounds. Our goal is *not* to compare these bounds when using standard practice, but to see how tight they can be *in principle* if we use every tool in our toolbox to minimise the expected bounds.[9] While these optimisations will be impractical for large models and datasets, they can provide some statistical insight.

**Learning Algorithm.** Certain learning algorithms may work better with test set bounds, and others with PAC-Bayes bounds. Instead of choosing a fixed algorithm, we *meta-learn* (Schmidhuber, 1987; Thrun & Pratt, 2012) separate algorithms to optimise each bound in expectation: we parametrise a hypothesis space $\mathcal{H}_\theta$ and a *posterior map* $Q_\theta \colon \mathcal{Z}^N \to \mathcal{M}_1(\mathcal{H}_\theta)$ by a finite dimensional vector $\theta$, which is trained to optimise the expected bound (we will amalgamate all meta-learnable parameters into the single vector $\theta$). This is explained in more detail below. This way, we obtain algorithms that are optimised for each bound. After meta-learning, we can further refine each PAC-Bayes posterior by minimising the PAC-Bayes bound, see Appendix I.4.

**Task Distribution.** In meta-learning, we refer to a data-generating distribution $D$ and dataset $S \sim D^N$ as a *task*. We consider a *distribution* over tasks, $D \sim \mathcal{T}$, where $\mathcal{T}$ is a distribution over

---

[9]Our goal here is to minimise *high probability* PAC-Bayes and test set bounds *in expectation*. See Dziugaite et al. (2021, Appendix J) for a relevant discussion.

data-generating distributions, and aim to find the best expected bounds for this distribution achievable by an optimised algorithm.[10] We choose especially simple learning tasks — synthetic 1-dimensional binary classification problems, generated by thresholding Gaussian process (GP) samples — which allows us to fully control the task distribution and easily inspect predictive distributions visually to diagnose learning. Appendix I.1 contains full details.

**Priors.** The choice of prior is crucial in PAC-Bayes, and the role of *data-dependent priors* (DDPs) (Ambroladze et al., 2007; Parrado-Hernández et al., 2012; Pérez-Ortiz et al., 2021) has been gaining increased attention. This involves splitting the dataset into $N = N_{\text{prior}} + N_{\text{risk}}$ datapoints. The DDP is allowed to depend on the *prior set* of size $N_{\text{prior}}$ (standard priors use $N_{\text{prior}} = 0$), and the risk bound is computed on the *risk set* of size $N_{\text{risk}}$. Crucially, *the bound is valid when the posterior depends on all $N$ datapoints*. Recently, Dziugaite et al. (2021) showed that DDPs can lead to tighter expected bounds than the optimal non-data-dependent prior, and are sometimes even *required* to obtain non-vacuous bounds. Pérez-Ortiz et al. (2021) also report much tighter bounds when using DDPs. In our experiments we meta-learn a DDP as a map from the prior set to the prior, $P_\theta \colon \mathcal{Z}^{N_{\text{prior}}} \to \mathcal{M}_1(\mathcal{H})$. To compare PAC-Bayes DDPs against test set bounds, we sweep the prior/train set proportion from 0 to 0.8 and see what the tightest value obtained is. Strictly this would require a union bound over the proportions, but here we are primarily interested in comparing the various bounds against each other on an even footing and vary the proportion for illustrative purposes.

**The Meta-Learning Objective.** We now discuss meta-learning in more detail. During meta-training, $\theta$ is trained to optimise the expected PAC-Bayes generalisation bound over the task distribution:

$$\mathbb{E}_{D\sim\mathcal{T}}\mathbb{E}_{S\sim D^N} B\big[\Delta_\theta(\overline{R}_{S_{\text{risk}}}(Q_\theta(S)), \, \cdot\,), \tfrac{1}{N_{\text{risk}}}\big(\text{KL}(Q_\theta(S)\|P_\theta(S_{\text{prior}})) + \log\tfrac{\mathcal{I}_{\Delta_\theta}(N_{\text{risk}})}{\delta}\big)\big], \quad (19)$$

where the $\theta$ in $\Delta_\theta$ denotes that some bounds (Catoni and learned convex function) have meta-learnable parameters. Alternatively, for a meta-learner that minimises a test set bound, the objective is simply $\mathbb{E}_{D\sim\mathcal{T}}\mathbb{E}_{S\sim D^N}\overline{R}_{S_{\text{test}}}(Q_\theta(S_{\text{train}}))$, since all test set bounds are monotonic in the test set risk. We use the $0/1$ loss. As the classifiers are stochastic, the empirical risk is still differentiable with respect to $\theta$. In contrast to PAC-Bayes, the predictor that minimises the test set bound can be made deterministic after $\theta$ is learned, since it tends to eventually learn essentially deterministic classifiers; see Appendix I.2. We sample $T = 80\,000$ tasks $D_t \sim \mathcal{T}$, with associated datasets $S_t \sim D_t^N$. These form the *meta-trainset*. Additionally, we sample 1024 tasks that form a *meta-testset* used to estimate the average bounds over $\mathcal{T}$ after meta-training. For the PAC-Bayes bounds, we then Monte Carlo estimate (19). Hence, the final objective for a PAC-Bayes meta-learner is (a minibatched version of):

$$\tfrac{1}{T}\sum_{t=1}^T B\big[\Delta_\theta(\overline{R}_{S_{t,\text{risk}}}(Q_\theta(S_t)), \, \cdot\,), \tfrac{1}{N_{\text{risk}}}\big(\text{KL}(Q_\theta(S_t)\|P_\theta(S_{t,\text{prior}})) + \log\tfrac{\mathcal{I}_{\Delta_\theta}(N_{\text{risk}})}{\delta}\big)\big]. \quad (20)$$

Similarly, the objective for the test set bound meta-learner is $\tfrac{1}{T}\sum_{t=1}^T \overline{R}_{S_{t,\text{test}}}(Q_\theta(S_{t,\text{train}}))$. The bounds we compute on datasets in the meta-testset, after meta-training is complete and $\theta$ is frozen, are valid even though $\theta$ was optimised on the meta-trainset. This highlights a contrast between our procedure and the PAC-Bayes meta-learning in Amit and Meir (2018), Liu et al. (2021), and Rothfuss et al. (2021) and Farid and Majumdar (2021). While those works use PAC-Bayes to analyse generalisation of a meta-learner on new tasks, we use PAC-Bayes to analyse generalisation *within* individual tasks.

**Parametrising the Meta-Learner and Hypothesis Space.** We now describe how to parametrise the hypothesis space $\mathcal{H}_\theta$ and the maps $Q_\theta, P_\theta$. We meta-learn a feature map $\phi_\theta \colon \mathbb{R} \to \mathbb{R}^K$ and choose[11] $\mathcal{H}_\theta = \{h_w : h_w(x) = \text{sign}\langle w, \phi_\theta(x)\rangle, \, w \in \mathbb{R}^K\}$. For $Q_\theta$ and $P_\theta$ Gaussian, this hypothesis space allows us to compute the empirical Gibbs risk without Monte Carlo integration; see Appendix I.3 for details. For the form of $Q_\theta$, we take inspiration from Neural Processes (NPs) (Garnelo, Rosenbaum, et al., 2018; Garnelo, Schwarz, et al., 2018; Kim et al., 2019). NPs use neural networks to flexibly parametrise a map from datasets to predictive distributions that respects the permutation invariance of datasets (Zaheer et al., 2017). They are regularly benchmarked on 1D meta-learning tasks, making them ideally suited. We make a straightforward modification to NPs to make them output Gaussian measures over weight vectors $w \in \mathbb{R}^K$. Hence, they act as parametrisable maps from $\mathcal{Z}^N$ to the set of Gaussian measures on $\mathbb{R}^K$.

---

[10]We could also consider drawing all datasets from a *single* task $D$, which would more directly match Section 3. We regard this case as less interesting, since we would often want a bound to perform well on a variety of tasks.

[11]The dimensionality $K$ is fixed a priori.

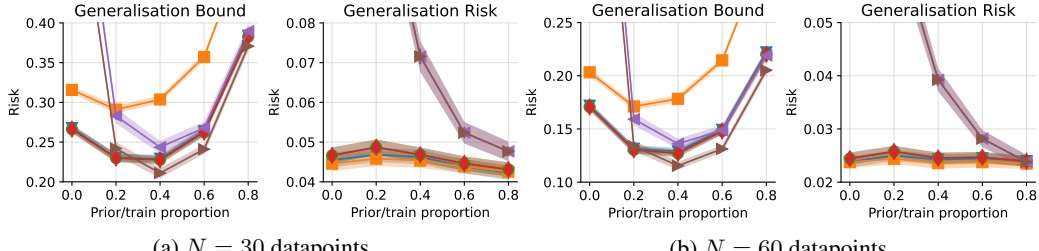

(a) $N = 30$ datapoints.  (b) $N = 60$ datapoints.

Figure 3: **Average generalisation bound and actual generalisation risk** ($\pm$ two standard errors) for CNN-NP meta-learners trained to optimise Catoni (▼), PAC-Bayes-kl (■), conjectured PAC-Bayes-kl (●), learned convex (◆), Chernoff test set (◀), and binomial tail test set (▶) bounds. Catoni, conjectured PAC-Bayes-kl, and learned convex overlap. The generalisation risks for Chernoff and binomial tail test set bounds are identical as they share the same meta-learner; only the bound computation differs. The bounds are valid with failure probability $\delta = 0.1$ except for conjectured PAC-Bayes-kl, which should be a lower bound on the best bound achievable with Theorem 3. Corresponding plots for the MLP-NP are in Appendix J.2.

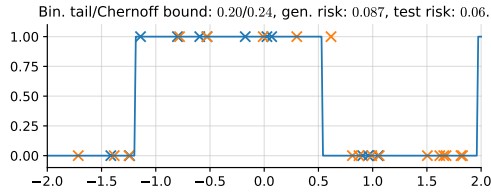

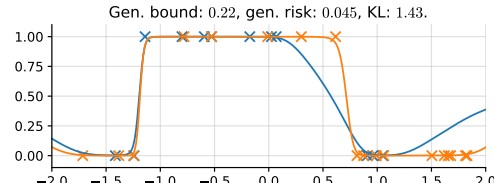

(a) **Binomial tail/Chernoff test set** bounds, showing the learned hypothesis (—), the train set (✕) of size 12 and the test set (✕) of size 18.

(b) **Learned convex** bound with data-dependent prior, showing the prior (—) and posterior (—) predictive, prior set (✕) of size 12 and risk set (✕) of size 18.

Figure 4: **Predictions on one of the 1D datasets in the meta-test set** with $N = 30$ and prior/train proportion $0.4$. For each method, we report the generalisation bound and actual generalisation risk. For the test set model, we also show the risk on the test set, and for the PAC-Bayes model we show the KL-divergence. The learned convex bound meta-learner has learned a DDP that provides a "first guess" given the prior set, which is then refined by the posterior. Figures for other PAC-Bayes bounds and datasets are provided in Appendix J.1.

We considered two kinds of NP, one based on multilayer perceptrons (MLP-NP) and another based on convolutional neural networks (CNN-NP) (detailed in Appendices I.5 and I.6) Although the MLP-NP is very flexible, the state-of-the-art in NPs on 1D tasks is given by CNN-based NPs (Bruinsma et al., 2021a; Foong et al., 2020; Gordon et al., 2020). We use an architecture closely based on the *Gaussian Neural Process* (Bruinsma et al., 2021a), which outputs full-covariance Gaussians. As expected, we found the CNN-NP to produce tighter (or comparable) average bounds to the MLP-NP, while using far fewer parameters, and training much more reliably and quickly. This is because the CNN-NP is translation equivariant, and hence exploits a key symmetry of the problem. Hence, we focus on the CNN-NP, but report some results for the MLP-NP in Appendix J.2. Hyperparameter details are given in Appendix I.7.

**Results.** We show example classification tasks and average bounds on the meta-test set in Figures 3 and 4. Note that the test set classifier became deterministic and makes hard predictions whereas the PAC-Bayes classifier shows uncertainty; see Appendix I.2 for a discussion. The PAC-Bayes-kl bound is loosest, which is unsurprising as it has no optimisable parameters to adapt to $\mathcal{T}$.[12] Surprisingly, the results for Catoni, conjectured PAC-Bayes-kl, and learned convex are nearly identical. As long as optimisation has succeeded reasonably, this suggests empirically that, in light of Corollary 3, one of the Catoni bounds may be very nearly optimal among all convex functions for this task distribution — there is not much "slack" from choosing suboptimal $\Delta$ here. We also see that the Catoni and learned convex bounds with prior proportion $0.4$ are tighter than any Chernoff test set bound. Hence, *PAC-Bayes can provide slightly tighter (or comparable) generalisation bounds to a Chernoff test set bound*. However, we see that *the binomial tail test set bound with $40\%$ of the data used for the selecting the predictor and the remaining $60\%$ used for evaluating the bound leads to*

---

[12]This is in contrast with usual applications of PAC-Bayes, where one does not have a meta-dataset with which to optimise parameters of the bound. In that setting, it can be an advantage to not have tunable parameters.

*the tightest generalisation bounds overall*. Corollary 3 sheds light on this behaviour: the optimal generic PAC-Bayes bound reduces, at best, to the Chernoff test set bound when the posterior equals the prior. However, the Chernoff bound is itself looser than the binomial tail bound. Of course, the posterior does not equal the prior here, but Corollary 3 indicates there is an extra source of looseness that PAC-Bayes has to overcome relative to the binomial tail bound. Finally, although the test set meta-learner leads to the tightest generalisation bounds, its generalisation risk is roughly double that of the PAC-Bayes meta-learner when the prior/train set proportion is $0.4$.

## 5  Conclusions, Open Problems, and Limitations

PAC-Bayes presents a potentially attractive framework for obtaining tight generalisation bounds in the small-data regime. We have investigated the tightness of PAC-Bayes and test set bounds in this regime both theoretically and experimentally. Theoretically, we showed that the generic PAC-Bayes theorem of Germain et al. (2009) and Bégin et al. (2016) which encompasses a wide range of PAC-Bayes bounds, cannot produce tighter bounds in expectation than the expression obtained by discarding the $\log(2\sqrt{N})/N$ term in the Langford and Seeger (2001) bound (*i.e.*, the *conjectured PAC-Bayes-kl bound*; Corollary 3). Although we did not prove that the conjectured PAC-Bayes-kl bound is a valid generalisation bound, numerical evidence suggests (Figures 2c and 2d) that there may exist a convex function $\Delta$ which achieves it, at least for the distributions over empirical risk and KL-divergence we considered. This suggests the following open problem:

**Open Problem 1.** *For an arbitrary distribution over datasets, does there exist a choice of $\Delta$ such that the expected conjectured PAC-Bayes-kl bound is attained (Corollary 3)? If not, how close can one get to the expected conjectured PAC-Bayes-kl bound?*

If such a $\Delta$ exists, then that would imply the conjectured PAC-Bayes-kl bound is a valid generalisation bound (see Section 3) and resolve Problem 6.1.2 of Langford (2002) in the affirmative.

We then considered, in a controlled experimental setting where meta-learning all parameters of the bounds and learning algorithms was feasible, whether PAC-Bayes bounds could be tighter than test set bounds. Although we found PAC-Bayes competitive with Chernoff bounds, both were outperformed by the binomial tail test set bound. This motivates a second open problem:

**Open Problem 2.** *Can a PAC-Bayes bound be found that relaxes gracefully to the binomial tail test set bound (Theorem 1) when the posterior is equal to the prior?*

Resolving these problems could have a significant impact on the tightness of PAC-Bayes applied to small-data, and clarify our understanding of the relationship between PAC-Bayes and test set bounds.

**Limitations.** In this paper, we concern ourselves with understanding the tightness of bounds in what might be called the *standard PAC-Bayes setting* of supervised learning: bounded losses, i.i.d. data, and Gibbs risk. We also focus on bounds that are first order in the sense that they rely only on the empirical Gibbs risk, though extending the analysis to consider other PAC-Bayes theorems (*e.g.* Rivasplata et al. (2020) and Tolstikhin and Seldin (2013)) would be of interest, especially with regards to Open Problems 1 and 2. For many practical applications in which performance guarantees are needed (*e.g.* health care), the i.i.d. assumption should be considered carefully, as it is likely an unrealistic simplification. Furthermore, Gibbs classifiers are less commonly used than deterministic classifiers in practice. To address these and other concerns, PAC-Bayes has been generalised in many directions beyond the scope of the standard setting we consider. Examples include bounds for non-i.i.d. data (Alquier & Guedj, 2018; Rivasplata et al., 2020; Seldin et al., 2012), unbounded losses (Germain et al., 2016), derandomised classifiers (Blanchard & Fleuret, 2007; Viallard et al., 2021), and Bayes risk (Germain et al., 2015; Masegosa et al., 2020). Bounds based on other divergences besides the KL have also been proposed (Alquier & Guedj, 2018; Bégin et al., 2016). As our proof relies primarily on tools from convex analysis, and Jensen's inequality is ubiquitous in PAC-Bayes bounds, it would be interesting to see if our arguments can be extended beyond the limited setting we focus on.

Finally, our meta-learning experiments only considered 1D classification, and the results might not necessarily be representative of more realistic datasets. We also only consider Gaussian prior and posterior distributions in our experiments for the sake of tractability. Scaling up the experiments and considering more flexible distributions is an important, but potentially challenging, avenue for future work.

## Acknowledgements and Funding Transparency Statement

We would like to thank Pierre Alquier and Yann Dubois for insightful discussions, John Langford for clarifying a remark on test set bounds, David Janz, Will Tebbutt, and Austin Tripp for providing helpful comments on a draft version of this paper, and Omar Rivasplata for helpful comments on an earlier version of this manuscript. Andrew Y. K. Foong gratefully acknowledges funding from a Trinity Hall Research Studentship and the George and Lilian Schiff Foundation. Wessel P. Bruinsma was supported by the Engineering and Physical Research Council (studentship number 10436152). David R. Burt acknowledges funding from the Qualcomm Innovation Fellowship and the Williams College Herchel Smith Fellowship. Richard E. Turner is supported by Google, Amazon, ARM, Improbable, Microsoft, EPSRC grant EP/T005637/1, and the UKRI Centre for Doctoral Training in the Application of Artificial Intelligence to the study of Environmental Risks (AI4ER).

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
