## A Relationship Between PAC-Bayes and Occam Bound

The well-known *Occam* bounds can be derived by a simple application of the union bound to a countable hypothesis class $\mathcal{H}$. In particular, we can consider a "prior" distribution $P$, which functions similarly to the PAC-Bayes prior. Then by applying the union bound and weighting each hypothesis $h$ with a failure probability of $P(\{h\})\delta$, we can convert any test set bound into a corresponding train set bound. Applying this to Theorem 1, we obtain:

**Theorem A.1** (Binomial tail Occam bound, Langford (2002), Theorem 4.6.1). *Let $\mathcal{H}$ be countable, and fix $P \in \mathcal{M}_1(\mathcal{H})$, $\ell \in \{0, 1\}$ and $\delta \in (0, 1)$. Then*

$$\Pr\Big((\forall h)\ R_D(h) \leq \overline{e}\big(N, N R_S(h), P(\{h\})\delta\big)\Big) \geq 1 - \delta, \tag{21}$$

*where $\overline{e}$ is defined in Theorem 1.*

Alternatively, applying this procedure to Theorem 2 yields a looser bound:

**Theorem A.2** (Chernoff Occam bound, Langford (2002), Corollary 4.6.2). *Let $\mathcal{H}$ be countable, and fix $P \in \mathcal{M}_1(\mathcal{H})$, $\ell \in [0, 1]$ and $\delta \in (0, 1)$. Then*

$$\Pr\left((\forall h)\ \mathrm{kl}(R_S(h), R_D(h)) \leq \frac{1}{N}\left[\log\frac{1}{P(\{h\})} + \log\frac{1}{\delta}\right]\right) \geq 1 - \delta. \tag{22}$$

Following Langford (2005, Sec 5.1), it is instructive to compare the Chernoff Occam bound with the PAC-Bayes-kl bound (Corollary 2) when $\mathcal{H}$ is countable and $Q$ is constrained to be a point mass, *i.e.* $Q = Q_h := \delta_h$, where $\delta_h$ denotes the Dirac measure at $h$. In that case, $\mathrm{KL}(Q_h \| P)$ reduces to $\log(1/P(\{h\}))$, and the Gibbs risks $\overline{R}_S(Q_h), \overline{R}_D(Q_h)$ simply reduce to the risks $R_S(h), R_D(h)$. Then the PAC-Bayes-kl bound states that:

$$\Pr\left((\forall h)\ \mathrm{kl}(R_S(h), R_D(h)) \leq \frac{1}{N}\left[\log\frac{1}{P(\{h\})} + \log\frac{2\sqrt{N}}{\delta}\right]\right) \geq 1 - \delta. \tag{23}$$

Comparing Equation (23) with Theorem A.2, we see that the PAC-Bayes-kl bound leads to a bound on $\mathrm{kl}(R_S(h), R_D(h))$ which is looser by an additive constant of $\log(2\sqrt{N})/N$ compared to the Chernoff Occam bound. Hence the PAC-Bayes bound does not relax gracefully to the Occam bound in this case, which motivates Open Problem 6.1.2 in Langford (2002). In fact, by Remark 2 and Corollary 3, we know that *if* we could find a convex $\Delta$ that allowed us to remove this $\log(2\sqrt{N})/N$ term (*i.e.*, the conjectured PAC-Bayes-kl bound), this would be the tightest possible bound obtainable from the generic PAC-Bayes theorem (Theorem 3). This motivates Open Problem 1. Finally, noting that Theorem A.2 is itself a looser version of Theorem A.1, we see that a PAC-Bayes bound that relaxes gracefully to Theorem A.1 is not obtainable from Theorem 3, motivating Open Problem 2.

## B Proof of Generic PAC-Bayes Theorem (Theorem 3)

We provide a proof of Theorem 3 here for convenience, which closely follows the proof given in Bégin et al. (2016). We first require a well-known lemma:

**Lemma B.1** (Kullback-Leibler change of measure inequality, Boucheron et al., 2013, Corollary 4.15). *For any set $\mathcal{H}$, probability measures $P, Q \in \mathcal{M}_1(\mathcal{H})$, and measurable function $\phi : \mathcal{H} \to \mathbb{R}$,*

$$\mathbb{E}_{h \sim Q}\phi(h) \leq \mathrm{KL}(Q\|P) + \log\left(\mathbb{E}_{h \sim P}e^{\phi(h)}\right). \tag{24}$$

In order to deal with general bounded losses $\ell \in [0, 1]$, we also use a lemma proven in Maurer (2004):

**Lemma B.2** (Maurer (2004), Lemma 3). *For any $[0, 1]$-valued random variable $z$, let $z'$ denote the unique Bernoulli random variable with $\mathbb{E}[z'] = \mathbb{E}[z]$. Let $S = (z_1, \ldots, z_N)$ and $S' = (z'_1, \ldots, z'_N)$ denote tuples of $N$ such random variables. Then for any convex function $f : [0, 1]^N \to \mathbb{R}$,*

$$\mathbb{E}[f(S)] \leq \mathbb{E}[f(S')]. \tag{25}$$

We can now prove Theorem 3:

*Proof of Theorem 3.* Applying Jensen's inequality followed by Lemma B.1, we have, for all $Q \in \mathcal{M}_1(\mathcal{H})$:

$$N\Delta(\overline{R}_S(Q), \overline{R}_D(Q)) = N\Delta(\mathbb{E}_{h \sim Q} R_S(h), \mathbb{E}_{h \sim Q} R_D(h)) \tag{26}$$

$$\leq \mathbb{E}_{h \sim Q} N\Delta(R_S(h), R_D(h)) \tag{27}$$

$$\leq \mathrm{KL}(Q \| P) + \log\left(\mathbb{E}_{h \sim P} e^{N\Delta(R_S(h), R_D(h))}\right). \tag{28}$$

Applying Markov's inequality to the random variable $\mathbb{E}_{h \sim P} e^{N\Delta(R_S(h), R_D(h))}$ (which is random through $S \sim D^N$), we obtain, for any $\delta \in (0, 1)$:

$$\Pr\left(\mathbb{E}_{h \sim P} e^{N\Delta(R_S(h), R_D(h))} \leq \frac{\mathbb{E}_{S \sim D^N} \mathbb{E}_{h \sim P} e^{N\Delta(R_S(h), R_D(h))}}{\delta}\right) \geq 1 - \delta. \tag{29}$$

Combining this with Equation (28) yields, with probability at least $1 - \delta$, for all $Q$ simultaneously:

$$\Delta(\overline{R}_S(Q), \overline{R}_D(Q)) \leq \frac{1}{N}\left[\mathrm{KL}(Q \| P) + \log \frac{\mathbb{E}_{S \sim D^N} \mathbb{E}_{h \sim P} e^{N\Delta(R_S(h), R_D(h))}}{\delta}\right]. \tag{30}$$

Finally, we upper bound $\mathbb{E}_{S \sim D^N} \mathbb{E}_{h \sim P} e^{N\Delta(R_S(h), R_D(h))}$ by a quantity than can be computed without knowing the true distribution $D$. By Tonelli's theorem:

$$\mathbb{E}_{S \sim D^N} \mathbb{E}_{h \sim P} e^{N\Delta(R_S(h), R_D(h))} = \mathbb{E}_{h \sim P} \mathbb{E}_{S \sim D^N} e^{N\Delta(R_S(h), R_D(h))}. \tag{31}$$

We will now upper bound the inner expectation by a quantity that is independent of $h$. Denote the datapoints in $S$ as $S = ((x_1, y_1), \ldots, (x_N, y_N))$. Recall that $R_S(h) := \frac{1}{N}\sum_{n=1}^{N} \ell((x_n, y_n), h)$, and note that $\ell((x_n, y_n), h)$ is a $[0, 1]$-valued random variable. Let $L$ denote the $N$-tuple of these random variables for each datapoint in $S$, *i.e.* $L = (\ell((x_1, y_1), h), \ldots, \ell((x_N, y_N), h))$, and let $M(L) := \frac{1}{N}\sum_{n=1}^{N} L_n$ be the arithmetic mean of $L$. Then the function $f: L \mapsto e^{N\Delta(M(L), R_D(h))}$ is convex since $M(L)$ is linear in $L$, $\Delta$ is convex and the exponential function is convex and nondecreasing. Hence defining $L'$ as the $N$-tuple of Bernoulli random variables such that $\mathbb{E}[L'_n] = \mathbb{E}[L_n] = R_D(h)$ for $1 \leq n \leq N$ and applying Lemma B.2,

$$\mathbb{E}_{S \sim D^N} e^{N\Delta(R_S(h), R_D(h))} = \mathbb{E}\, e^{N\Delta(M(L), R_D(h))} \tag{32}$$

$$= \mathbb{E}\, f(L) \tag{33}$$

$$\leq \mathbb{E}\, f(L') \tag{34}$$

$$= \sum_{k=0}^{N} \Pr\big(M(L') = k/N\big) e^{N\Delta(k/N, R_D(h))} \tag{35}$$

$$= \sum_{k=0}^{N} \binom{N}{k} R_D(h)^k (1 - R_D(h))^{N-k} e^{N\Delta(k/N, R_D(h))} \tag{36}$$

$$\leq \sup_{r \in [0,1]} \sum_{k=0}^{N} \binom{N}{k} r^k (1 - r)^{N-k} e^{N\Delta(k/N, r)} \tag{37}$$

$$= \mathcal{I}_\Delta(N). \tag{38}$$

Substituting this into Equation (31) and then Equation (30) completes the proof. $\qquad\square$

## C   Basic Facts from Convex Analysis

A function $f: \mathbb{R}^n \to \mathbb{R} \cup \{+\infty\}$ is called *proper* if it not everywhere $\infty$ and nowhere $-\infty$. The convex conjugate $f^*: \mathbb{R}^n \to \mathbb{R} \cup \{+\infty\}$ of $f$ is defined as $f^*(c) = \sup_{x \in \mathbb{R}^n} (\langle c, x \rangle - f(x))$. If $f$ is proper, convex, and l.s.c., then $f^*$ is also proper, convex, and l.s.c. Moreover, if $f$ is proper, convex, and l.s.c., then $f$ is equal to its double convex conjugate: $f(x) = \sup_{c \in \mathbb{R}^n} (\langle c, x \rangle - f^*(c))$. A pointwise supremum of convex functions is again convex; and a pointwise supremum of l.s.c. functions remains l.s.c. Convex functions $f: A \to \mathbb{R}$ defined on only a convex subset $A \subseteq \mathbb{R}^n$ are extended to the whole of $\mathbb{R}^n$ by setting $f|_{\mathbb{R}^n \setminus A} = +\infty$. See, for example, Rockafellar and Wets (2010) for proofs of these results and an introduction to the topic.

## D   Monotonicity of $\Delta$

**Proposition D.1.** *For* $\Delta \colon [0,1]^2 \to \mathbb{R} \cup \{+\infty\}$ *a proper, convex, and lower semi-continuous function,* $q \in [0,1]$, $\delta \in (0,1]$, *and* $\mathrm{KL} \geq 0$, *define*

$$\bar{p}_\Delta = \sup \left\{ p \in [0,1] : \Delta(q,p) \leq \frac{1}{N} \left( \mathrm{KL} + \log \frac{\mathcal{I}_\Delta(N)}{\delta} \right) \right\} \tag{39}$$

*Then there exists a proper, convex, lower semi-continuous* $\Delta' \colon [0,1]^2 \to \mathbb{R} \cup \{+\infty\}$ *such that* $\bar{p}_{\Delta'} \leq \bar{p}_\Delta$, *and for every* $q \in [0,1]$, $\Delta'(q, \cdot)$ *is monotonically increasing.*

*Proof.* Define $\Delta'(q,p) = \inf_{p' \geq p} \Delta(q,p')$. We will prove that $\Delta'$ has the desired properties. First, $\Delta'$ is not infinity everywhere, as $\Delta'(q,p) \leq \Delta(q,p)$ and $\Delta$ is proper. Second, since $\Delta$ is l.s.c. and proper, it obtains a minimum on the compact set $[0,1]^2$, hence $\Delta'$ does not take the value $-\infty$. Therefore, $\Delta'$ is proper.

Since $\Delta$ is l.s.c., the strict sublevel sets of $\Delta$ are open; that is, for all $y \in \mathbb{R}$, $\{(q,p) : \Delta(q,p) < y\}$ is open. Then,

$$\{(q,p) : \Delta'(q,p) < y\} = \{(q,p) : \inf_{p' \geq p} \Delta(q,p') < y\} \tag{40}$$

$$= \bigcup_{p' \in [p,1]} \{(q,p') : \Delta(q,p') < y\}. \tag{41}$$

The equality follows from noting that the infimum on the closed set $[p,1]$ must be achieved as $\Delta$ is l.s.c.[13] As we have written $\{(q,p) : \Delta'(q,p) < y\}$ as a union of open sets, it is open. Hence, the sublevel sets of $\Delta'$ are open, implying $\Delta'$ is l.s.c.

We next show that $\Delta'$ is convex. Define the function $D : \mathbb{R}^3 \to \mathbb{R} \cup \{+\infty\}$ by,

$$D(q,p',p) = \begin{cases} \Delta(q,p) & p' \geq p \\ +\infty & \text{otherwise.} \end{cases}$$

$D(q,p,p')$ is convex since $\Delta$ is convex and $p' \geq p$ is a convex set. Also, $\inf_{p \in \mathbb{R}} D(q,p',p) = \inf_{p' \geq p} \Delta(q,p') = \Delta'(q,p)$. As the infimum projection of a convex function is convex (Rockafellar & Wets, 2010, Proposition 2.22), $\Delta'$ is convex. Also, $\Delta'(q,p)$ is monotonically increasing in $p$ as the infimum is taken over a smaller set for larger $p$.

It remains to show that $\bar{p}_{\Delta'} \leq \bar{p}_\Delta$. For all pairs $(q,p)$, $\Delta'(q,p) = \inf_{p' \geq p} \Delta(q,p') \leq \Delta(q,p)$. From this it follows that

$$\frac{1}{N} \left( \mathrm{KL} + \log \frac{\mathcal{I}_{\Delta'}(N)}{\delta} \right) \leq \frac{1}{N} \left( \mathrm{KL} + \log \frac{\mathcal{I}_\Delta(N)}{\delta} \right). \tag{42}$$

Finally, for any $\beta \in \mathbb{R} \cup \{+\infty\}$,

$$p'_\beta := \sup \{p \in [0,1] : \Delta'(q,p) \leq \beta\} = \sup \{p \in [0,1] : \Delta(q,p) \leq \beta\} =: p_\beta. \tag{43}$$

One inequality follows from $\Delta' \leq \Delta$. For the other, for any $p' \geq p \geq p_\beta$, we have $\Delta(q,p') > \beta$. Taking an infimum over such $p'$, noting that $\Delta$ is lower semi-continuous and therefore obtains a minimum on the closed interval $[p,1]$, $\Delta'(q,p) > \beta$. Hence $p'_\beta \leq p_\beta$. The result follows from combining Equation (42) and Equation (43). □

# E   Lemmas for Theorem 4

Let $C_\beta(p,q) := -\log(p(e^{-\beta} - 1) + 1) - \beta q$ for $\beta > 0$ and

$$\mathcal{I}_\Delta(N) = \sup_{r \in [0,1]} \mathbb{E}_{X \sim \mathrm{Bin}(r,N)}[e^{N\Delta(X/N,r)}] \tag{44}$$

**Lemma E.1.** *Consider* $q,p \in [0,1]$. *Then*

$$\sup_{\beta \in \mathbb{R}} C_\beta(q,p) = \mathrm{kl}(q,p). \tag{45}$$

---

[13]The equality holds if $\Delta$ is not l.s.c by the definition of the infimum as well

*Proof.* If $q = p = 0$, then $C_\beta(q,p) = 0 = \mathrm{kl}(q,p)$; and, if $q = p = 0$, then also $C_\beta(q,p) = 0 = \mathrm{kl}(q,p)$. If, on the other hand, $q = 0$ but $p = 1$, then clearly $\sup_{\beta \in \mathbb{R}} C_\beta(q,p) = \infty = \mathrm{kl}(q,p)$; and if $q = 1$ but $p = 0$, then also clearly $\sup_{\beta \in \mathbb{R}} C_\beta(q,p) = \infty = \mathrm{kl}(q,p)$. It remains to deal with the case that $q, p \in (0,1)$. In that case, to compute the supremum, set the derivative to zero:

$$q = \frac{pe^{-\beta}}{p(e^{-\beta} - 1) + 1} \tag{46}$$

and verify that we indeed have a maximum. This gives

$$-\beta = \log \frac{1 - p}{1 - q} + \log \frac{q}{p}, \tag{47}$$

so

$$-\log(p(e^{-\beta} - 1) + 1) = \beta + \log \frac{q}{p} = \log \frac{1 - q}{1 - p}. \tag{48}$$

Therefore,

$$\sup_{\beta \in \mathbb{R}} C_\beta(q,p) = -\log \frac{1 - p}{1 - q} + q \log \frac{1 - p}{1 - q} + q \log \frac{q}{p} = \mathrm{kl}(q,p). \tag{49}$$

$\square$

The following lemma is essentially Prop 2.1 from Germain et al. (2009), but stated in a slightly more careful form:

**Lemma E.2** (Germain et al. (2009))**.** *Consider* $0 \le q < p < 1$. *If* $q > 0$*, then there exists a unique* $\beta > 0$ *such that*

$$C_\beta(q,p) = \mathrm{kl}(q,p). \tag{50}$$

*On the other hand, if* $q = 0$*, then*

$$\lim_{\beta \to \infty} C_\beta(0,p) = \mathrm{kl}(0,p). \tag{51}$$

*Proof.* If $0 < q < p < 1$, then the unique $\beta$ follows from the proof of Lemma E.1:

$$\beta = \log \frac{1 - q}{1 - p} + \log \frac{p}{q} \in (0, \infty). \tag{52}$$

On the other hand, if $q = 0 < p < 1$, then

$$\mathrm{kl}(0,p) = -\log(1 - p) = \lim_{\beta \to \infty} C_\beta(0,p). \tag{53}$$

$\square$

**Lemma E.3** (Catoni (2007) and Germain et al. (2009))**.** *For every* $\beta > 0$*, it holds that* $\mathcal{I}_{C_\beta}(N) = 1$.

*Proof.* Let $r \in [0,1]$ and $X \sim \mathrm{Bin}(r, N)$. Note that

$$\mathbb{E}[e^{-\beta X}] = (r(e^{-\beta} - 1) + 1)^N. \tag{54}$$

Therefore,

$$\mathbb{E}[e^{N C_\beta(X/N, r)}] = \mathbb{E}[e^{-N \log(r(e^{-\beta} - 1) + 1) - \beta X}] = \frac{\mathbb{E}[e^{-\beta X}]}{(r(e^{-\beta} - 1) + 1)^N} = 1. \tag{55}$$

$\square$

**Lemma E.4.** *Let* $y \ge 0$*. Then* $B[\lim_{\beta \to \infty} C_\beta(0, \cdot), y] = \lim_{\beta \to \infty} B[C_\beta(0, \cdot), y]$.

*Proof.* Note that

$$\lim_{\beta \to \infty} C_\beta(0,p) = -\log(1 - p), \qquad C_\beta(0,p) = -\log(p(e^{-\beta} - 1) + 1). \tag{56}$$

Therefore,

$$B[\lim_{\beta \to \infty} C_\beta(0, \cdot), y] = 1 - e^{-y}, \qquad B[C_\beta(0, \cdot), y] = \min\left(\frac{1 - e^{-y}}{1 - e^{-\beta}}, 1\right) \tag{57}$$

The result then follows from the observation that

$$\lim_{\beta \to \infty} \frac{1 - e^{-y}}{1 - e^{-\beta}} = 1 - e^{-y}. \tag{58}$$

$\square$

## F  Learning a Convex Function

In Section 3, we optimised an objective with respect to a function $\Delta \colon [0,1]^2 \to \mathbb{R} \cup \{+\infty\}$ that was proper, l.s.c., and convex. In this appendix, we describe how a function $\Delta \colon [0,1]^2 \to \mathbb{R}$ that is differentiable and convex can be generally parametrised. We also discuss two challenges encountered during the optimisation: (1) computing and differentiating through a supremum and (2) computing and optimising a partial inverse.

### F.1  Parametrising a Convex Function

To generally parametrise a differentiable and convex $\Delta \colon [0,1]^2 \to \mathbb{R}$, we use the sum of an affine function and a one-hidden-layer neural network with softplus activation functions and positive weights at the output layer. The combination of positive weights and softplus activation functions ensures that the neural network is a convex function. The number of hidden units used is varied between $128$ and $1024$; the precise numbers are specified in the descriptions of the experiments.

### F.2  Computing and Differentiating Through a Supremum

The generic PAC-Bayes theorem (Theorem 3) involves $\mathcal{I}_\Delta(N)$, which in turn involves a supremum of a function over $r \in [0,1]$. When optimising with respect to $\Delta$, we therefore need to compute and differentiate through a supremum. To compute the supremum, we finely discretise $[0,1]$ and compute the maximum over this grid. Technically, by approximating the supremum in this way, the bound is approximate, which means that it might not be a valid generalisation bound. However, by making the discretisation very fine, using an inter-point spacing of $10^{-5}$, the error on the generalisation bound is negligible. To differentiate the supremum, we simply run automatic differentiation on the approximation. In the remainder of this subsection, we give a plausible explanation for why this procedure also approximates the gradients correctly. The following discussion is based on `https://math.stackexchange.com/questions/3753495/derivative-of-argmin-in-a-constrained-problem`.

Consider $f \colon [0,1] \times \mathbb{R} \to \mathbb{R}$ continuously differentiable in its interior. We aim to compute

$$\frac{\mathrm{d}}{\mathrm{d}\theta} \sup_{x \in [0,1]} f(x,\theta) = \frac{\mathrm{d}}{\mathrm{d}\theta} \max_{x \in [0,1]} f(x,\theta) \tag{59}$$

where the supremum turns into a maximum by compactness of $[0,1]$ and continuity of $f$. We assume that the maximum is uniquely obtained and write

$$z(\theta) = \arg\max_{x \in [0,1]} f(x,\theta). \tag{60}$$

Then

$$\sup_{x \in [0,1]} f(x,\theta) = f(z(\theta),\theta), \tag{61}$$

so we can compute the derivative with respect to $\theta$ with the chain rule if we can compute $z'(\theta)$.

CASE 1: The constraint $x \in [0,1]$ is not binding. In that case, the stationarity condition is satisfied in a neighbourhood of $\theta$:

$$\partial_x f(z(\theta),\theta) = 0. \tag{62}$$

Therefore,

$$\frac{\mathrm{d}}{\mathrm{d}\theta} \sup_{x \in [0,1]} f(x,\theta) = \partial f(z(\theta),\theta) z'(\theta) + \partial_\theta f(z(\theta),\theta) = \partial_\theta f(z(\theta),\theta). \tag{63}$$

CASE 2: The constraint $x \in [0,1]$ is binding. In that case, $\partial_x f(z(\theta),\theta) \neq 0$, so $\partial_x f(z(\theta),\theta) < 0$ and you can argue that the optimum will remain to be attained at the constraint in a neighbourhood of $\theta$. Therefore, $z'(\theta) = 0$, which means that again

$$\frac{\mathrm{d}}{\mathrm{d}\theta} \sup_{x \in [0,1]} f(x,\theta) = \partial f(z(\theta),\theta) z'(\theta) + \partial_\theta f(z(\theta),\theta) = \partial_\theta f(z(\theta),\theta). \tag{64}$$

In either case,

$$\frac{\mathrm{d}}{\mathrm{d}\theta} \sup_{x \in [0,1]} f(x, \theta) = \partial_\theta f(z(\theta), \theta), \tag{65}$$

which shows that the derivative with respect to the maximiser can be ignored. Assuming that $z(\theta)$ can be well approximated by computing the maximiser over the fine discretisation, and that in turn leads to a good approximation of $\partial_\theta f(z(\theta), \theta)$, this provides justification for our approach of simply running automatic differentiation on the approximation to the supremum.

Finally, we note that, although computing the derivative accurately is useful for the optimisation to succeed, the bounds we compute are valid regardless of how accurate the derivative is (subject to the computation of the supremum itself being sufficiently accurate). In practice, we observe that the learned convex bound decreases steadily during optimisation (Figure 2), and that it approaches, but never goes below, the conjectured PAC-Bayes-kl bound, as per Corollary 3, which provides evidence that the implementation is sufficiently accurate for our purposes.

### F.3 Computing and Optimising a Partial Inverse

The objective that we optimise with respect to $\Delta$ is $\overline{p}_\Delta$. Recall from Equation (6) that

$$\overline{p}_\Delta := B\big[\Delta(\overline{R}_S(Q), \cdot\,), \tfrac{1}{N}\big(\mathrm{KL}(Q\|P) + \log \tfrac{\mathcal{I}_\Delta(N)}{\delta}\big)\big]. \tag{66}$$

We now abbreviate $f := \Delta(\overline{R}_S(Q), \cdot\,)$ and $c := \tfrac{1}{N}\big(\mathrm{KL}(Q\|P) + \log \tfrac{\mathcal{I}_\Delta(N)}{\delta}\big)$, so that the objective is $B[f, c] = \sup\{p \in [0, 1] : f(p) \le c\}$ for $f \colon [0, 1] \to \mathbb{R}$ convex and $c \in \mathbb{R}$. Assuming that $f = f_\theta$ and $c = c(\theta)$ depend on some parameters $\theta$ (*i.e.*, the parameters of the neural network defining $\Delta$), our goal is to compute $B[f_\theta, c(\theta)]$ and optimise it with respect to $\theta$.

A possible issue that can be run into during optimisation is that, if $f(p) \le c$ for all $p \in [0, 1]$, then $B[f, c] = 1$ and the gradient with respect to $\theta$ may be zero, which means that the optimisation may fail to make progress. A similar issue is discussed by Dziugaite and Roy (2017) when trying to optimise the PAC-Bayes-kl bound: the derivative of the inverse Bernoulli KL can be zero if $c$ is large enough. In Dziugaite and Roy (2017, Sec 2.2) this is handled by upper bounding the inverse Bernoulli KL using Pinsker's inequality. This can lead to upper bounds which are greater than 1 (whereas the exact computation of $B[f, c]$ never allows this to happen), but has the advantage of always providing a useful gradient signal.

Similarly, we can define $\overline{B}[f, c] := \sup\{p \in \mathbb{R}_{\ge 0} : f(p) \le c\}$ for $f \colon \mathbb{R}_{\ge 0} \to \mathbb{R}$ and $c \in \mathbb{R}$, which ignores the constraint that $p \le 1$. Note that in our case, since $f_\theta$ is defined by a neural network, it is trivial to extend its domain from $[0, 1]$ to $\mathbb{R}_{\ge 0}$. This will allow us to obtain a useful derivative even when the bound is vacuous. Moreover, in our case $f_\theta$ will be convex, which means that $\overline{B}[f_\theta, c(\theta)] = B[f_\theta, c(\theta)]$ if $B[f_\theta, c(\theta)] < 1$: $B[f_\theta, c(\theta)]$ is characterised by an upcrossing[14] of $c(\theta)$ by $f_\theta$, and, by convexity, $f_\theta$ can have at most one such upcrossing.

We now describe how $\overline{B}[f_\theta, c(\theta)]$ can be (approximately) computed and differentiated with respect to $\theta$. To compute $\overline{B}[f_\theta, c(\theta)]$, we evaluate $f_\theta$ on a discretisation of the interval $[0, u]$ using an inter-point spacing of $10^{-4}$ and attempt to detect an upcrossing of $c(\theta)$. Assume that this procedure finds an upcrossing; otherwise, either increase $u$ (*e.g.*, by doubling) and try again or return $u$ and set the derivative to zero (failure). Denote $x = \overline{B}[f_\theta, c(\theta)]$ and note that $f_\theta$ is continuously differentiable in its interior, because it is a neural network with softplus activations. Assume that $x > 0$, which in practice turns out to nearly always be the case. Using continuity of $f_\theta$, it holds that $f_\theta(x) = c(\theta)$. It also holds that $\partial_x f_\theta(x) > 0$ ($f_\theta$ upcrosses $c(\theta)$ at $x$). Restricting $f_\theta$ to an appropriate neighbourhood, the derivative of $\overline{B}[f_\theta, c(\theta)]$ with respect to $\theta$ comes down to computing the derivative of $f_\theta^{-1}(c(\theta))$ with respect to $\theta$. The latter derivative can be computed as follows:

$$\partial_\theta c(\theta) = \frac{\mathrm{d}}{\mathrm{d}\theta} f_\theta(f_\theta^{-1}(c(\theta))) = \partial_x f_\theta(x) \frac{\mathrm{d}}{\mathrm{d}\theta} f_\theta^{-1}(c(\theta)) + \partial_\theta f_\theta(x), \tag{67}$$

which implies that

$$\frac{\mathrm{d}}{\mathrm{d}\theta} f_\theta^{-1}(c(\theta)) = \frac{\partial_\theta c(\theta) - \partial_\theta f_\theta(x)}{\partial_x f_\theta(x)}, \tag{68}$$

---

[14]We say that a function $f \colon \mathbb{R} \to \mathbb{R}$ upcrosses $y \in \mathbb{R}$ at $x \in \mathbb{R}$ if there exists some $\varepsilon > 0$ such that $f(x') < y$ for all $x' \in (x - \varepsilon, x)$ and $f(x') > y$ for all $x' \in (x, x + \varepsilon)$.

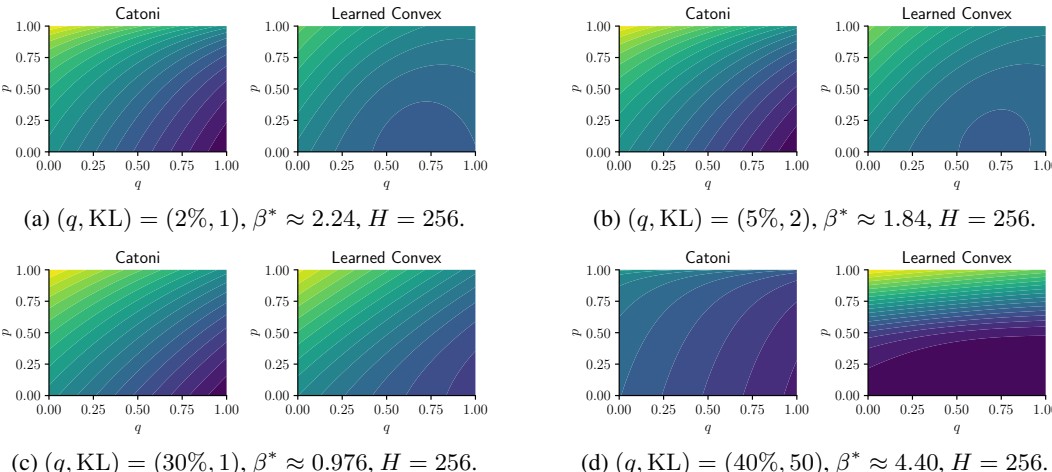

(a) $(q, \mathrm{KL}) = (2\%, 1)$, $\beta^* \approx 2.24$, $H = 256$.

(b) $(q, \mathrm{KL}) = (5\%, 2)$, $\beta^* \approx 1.84$, $H = 256$.

(c) $(q, \mathrm{KL}) = (30\%, 1)$, $\beta^* \approx 0.976$, $H = 256$.

(d) $(q, \mathrm{KL}) = (40\%, 50)$, $\beta^* \approx 4.40$, $H = 256$.

Figure 5: **Although the tightest Catoni bound is the optimal generic PAC-Bayes bound for a fixed dataset, evidence suggests that there are other choices for $\Delta$ which also achieve the tightest bound.** For various settings of fixed $q$ and KL, we optimise a convex function with $H$ hidden units to minimise $\overline{p}_\Delta$ with $\delta = 0.1$ and $N = 30$. We compare the optimal Catoni $C_{\beta^*}$ (left sides) with the learned convex function (right sides). For all runs, $\overline{p}_\Delta$ converged to $\inf_{\beta > 0} \overline{p}_{C_\beta}$ within a small tolerance.

recalling that $\partial_x f_\theta(x) > 0$.

Similarly to Appendix F.2, although computing the gradient through the partial inverse accurately is useful for optimising the convex function, the bound itself will be valid as long as the value of the partial inverse itself is computed sufficiently accurately.

## G   Additional Results for Numerical Verification of Theory

Figure 5 complements Figure 2 by comparing the optimal Catoni $C_\beta$ to examples of the learned convex functions, which demonstrates that there are other choices than $C_\beta$ which achieve bounds that are close to $\inf_{\Delta \in \mathcal{C}} \overline{p}_\Delta$ within a small tolerance. Figure 6 complements Figure 2 by considering three slightly more complicated cases of a random dataset. Note that, in Figures 6b and 6c, during iterations $10^4$–$10^6$, the optimiser struggles: the trace jumps around. The are various reasons for why this might have happened: the neural network parametrising $\Delta$ has too few hidden units, the learning rate of the optimiser is too large, the various approximations that involve $\Delta$ (Appendices F.2 and F.3) introduce too much error, or Open Problem 1 might be false.

## H   Worked Example for Corollary 3

In this section, we illustrate an application of Corollary 3 to determine when, in the simplified scenario where $R_S(Q) = \frac{1}{2}$ almost surely, the expected PAC-Bayes-kl bound is tighter than the tightest expected Catoni bound. This verifies with an analytic example, that, as we claim in Section 3, although the Catoni bound is optimal for a fixed dataset and learning algorithm, it is not optimal in expectation in the general case of a random dataset. To make the example more concrete, we also compute the bounds for $\mathrm{KL} = 1$ with probability $\frac{1}{2}$ and $\mathrm{KL} = 100$ otherwise. This choice for KL is motivated with the conclusion at the end the section.

Denote $\alpha = \frac{1}{N}(\mathrm{KL}(Q\|P) + \log \frac{1}{\delta})$. We can solve for the optimal expected Catoni bound:

$$\inf_{\beta > 0} \frac{1}{1 - e^{-\beta}} \left( 1 - e^{-\frac{1}{2}\beta} \mathbb{E}[e^{-\alpha}] \right). \tag{69}$$

Denote $u = \mathbb{E}[e^{-\alpha}]$ and set the derivative with respect to $\beta$ to zero:

$$(1 - e^{-\beta})\tfrac{1}{2}u e^{-\frac{1}{2}\beta} - e^{-\beta}(1 - u e^{-\frac{1}{2}\beta}) = 0. \tag{70}$$

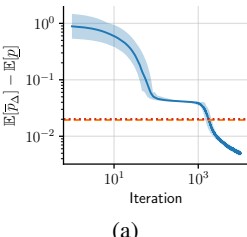
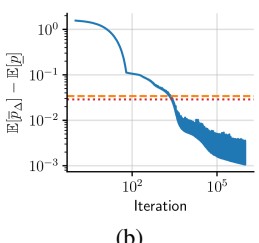
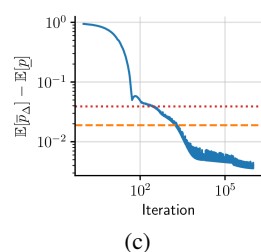

|     |     |     |
| --- | --- | --- |
| (a) | (b) | (c) |

Figure 6: **Extra results indicating that the tightest expected Catoni bound is not the optimal generic PAC-Bayes bound for a random dataset.** We optimise a convex function with $H$ hidden units to minimise $\bar{p}_\Delta$ with $\delta = 0.1$ and $N = 30$. All plots consider random $q$ and KL and show the *expected* difference with the conjectured PAC-Bayes-kl bound (Corollary 3). In (a), the shaded regions show the minimum and maximum over ten initialisations. Due to computational considerations, (b) and (c) only show one repetition, since they are run for much longer than (a) (roughly 25 hours for each run). All plots show the PAC-Bayes-kl bound (dotted red) and (c) and (d) show the optimal Catoni bound with parameter $\beta^*$ (dashed orange). All runs quickly converged to non-vacuous values. (a): $(q, \mathrm{KL}) \in \{(35\%, 5), (45\%, 30), (40\%, 7), (43\%, 25)\}$ uniformly, $\beta^* \approx 2.23$, $H = 1024$. (b): $(q, \mathrm{KL}) \in \{(3.0\%, 46.0), (9.6\%, 0.52), (14.2\%, 48.9)\}$ (rounded values) uniformly, $\beta^* \approx 3.21$, $H = 1024$. (c): $(q, \mathrm{KL}) \in \{(18.1\%, 2.56), (8.0\%, 5.83), (16.8\%, 30.0)\}$ (approximate values) uniformly, $\beta^* \approx 2.20$, $H = 1024$.

Letting $x = e^{-\frac{1}{2}\beta}$, this equation becomes a quadratic equation:

$$x^2 - \frac{2}{u}x + 1 = 0 \implies x = \frac{1}{u} \pm \sqrt{\frac{1}{u^2} - 1}. \tag{71}$$

Therefore,

$$\beta = 2\log\frac{u}{1 \pm \sqrt{1 - u^2}} = 2\log\frac{\mathbb{E}[e^{-\alpha}]}{1 \pm \sqrt{1 - \mathbb{E}[e^{-\alpha}]^2}}, \tag{72}$$

so the positive solution for $\beta$ is given by

$$\beta = 2\log\frac{\mathbb{E}[e^{-\alpha}]}{1 - \sqrt{1 - \mathbb{E}[e^{-\alpha}]^2}} \approx 2.803. \tag{73}$$

Plugging this back into the Catoni bound gives that

$$\inf_{\beta > 0}\mathbb{E}[\bar{p}_{C_\beta}] = \frac{\sqrt{1 - u^2}}{1 - \left(\frac{1 - \sqrt{1 - u^2}}{u}\right)^2} = \tfrac{1}{2}\left(1 + \sqrt{1 - \mathbb{E}[e^{-\alpha}]^2}\right) \approx 0.943. \tag{74}$$

We compare this with the choice $\Delta = \mathrm{kl}$, which corresponds to the PAC-Bayes-kl bound. In that case, the expected bound is given by

$$\mathbb{E}[\bar{p}_{\mathrm{kl}}] = \mathbb{E}\,B[\mathrm{kl}(\tfrac{1}{2}, \cdot), \alpha + \gamma] \tag{75}$$

where

$$\gamma = \frac{1}{N}\log\mathcal{I}_{\mathrm{kl}}(N) = \frac{1}{N}\log\sum_{n=0}^{N}\binom{N}{n}\left(\frac{N}{n}\right)^n\left(1 - \frac{N}{n}\right)^{N-n}. \tag{76}$$

To compute $B[\mathrm{kl}(\tfrac{1}{2}, \cdot), \alpha + \gamma]$, note that

$$\tfrac{1}{2}\log\frac{\frac{1}{2}}{p} + \tfrac{1}{2}\log\frac{\frac{1}{2}}{1 - p} = y \implies y_\pm = \tfrac{1}{2}\left(1 \pm \sqrt{1 - e^{-2y}}\right). \tag{77}$$

Therefore,

$$\mathbb{E}[\bar{p}_{\mathrm{kl}}] = \tfrac{1}{2}\left(1 + \mathbb{E}\sqrt{1 - e^{-2(\alpha+\gamma)}}\right) \approx 0.836. \tag{78}$$

This is better than the Catoni bound by more than $10\%$. Finally, by omitting $\gamma$, we find the conjectured PAC-Bayes-kl bound:

$$\mathbb{E}[\underline{p}] = \tfrac{1}{2}\left(1 + \mathbb{E}\sqrt{1 - e^{-2\alpha}}\right) \tag{79}$$

Note how similar the computed bounds are:

$$\inf_{\beta>0} \mathbb{E}[\overline{p}_{C_\beta}] = \tfrac{1}{2}\left(1 + \sqrt{1 - \mathbb{E}[e^{-\alpha}]^2}\right), \qquad \text{(optimal Catoni)} \tag{80}$$

$$\mathbb{E}[\overline{p}_{\mathrm{kl}}] = \tfrac{1}{2}\left(1 + \mathbb{E}\sqrt{1 - e^{-2(\alpha+\gamma)}}\right), \qquad \text{(PAC-Bayes-kl)} \tag{81}$$

$$\mathbb{E}[\underline{p}] = \tfrac{1}{2}\left(1 + \mathbb{E}\sqrt{1 - e^{-2\alpha}}\right). \qquad \text{(conjectured PAC-Bayes-kl)} \tag{82}$$

Define

$$\phi(x) = 1 - \tfrac{1}{2}\sqrt{1 - x^2}, \tag{83}$$

which is convex. Define the $\phi$-entropy of a random variable $X$ by

$$\mathbb{H}_\phi(X) = \mathbb{E}[\phi(X)] - \phi(\mathbb{E}[X]). \tag{84}$$

Observe that $\mathbb{H}_\phi(X)$ quantifies the slack in Jensen's inequality, which, in particular, means that $\mathbb{H}_\phi(X) \geq 0$. We then find that

$$\inf_{\beta>0} \mathbb{E}[\overline{p}_{C_\beta}] - \mathbb{E}[\underline{p}] = \mathbb{H}_\phi(e^{-\alpha}), \tag{85}$$

$$\mathbb{E}[\overline{p}_{\mathrm{kl}}] - \mathbb{E}[\underline{p}] = \mathbb{E}[\phi(e^{-\alpha}) - \phi(e^{-\gamma}e^{-\alpha})]. \tag{86}$$

Therefore, the PAC-Bayes-kl bound is tighter if and only if

$$\mathbb{E}[\phi(e^{-\alpha})] - \mathbb{E}[\phi(e^{-\gamma}e^{-\alpha})] \leq \mathbb{H}_\phi(e^{-\alpha}). \tag{87}$$

In words, the expected PAC-Bayes-kl bound is tighter than the tighest expected Catoni bound if the slack in Jensen's inequality is more than the slack introduced by scaling by $e^{-\gamma}$, which, for example, will be the case if $\mathrm{KL}(Q\|P)$ attains both small and large values.

# I  Additional Details for Synthetic Classification

## I.1  Data Generation Details

We now provide details of the task generation for the 1D classification experiments. For each task, we sample a 1D function $f$ from a Gaussian process (GP) with an exponentiated quadratic kernel with lengthscale $0.7$ and variance $1$. This is then turned into a classification problem by thresholding: $S = \big((x_n, \mathrm{sign}(f(x_n)))\big)_{n=1}^{N}$, where $x_n \sim \mathcal{U}[-2, 2]$. Finally, we only select tasks that are approximately balanced, so that the risk of a trivial predictor is $\approx 0.5$. This is done in a way that preserves the i.i.d. assumptions. In more detail, when sampling from the GP, in addition to sampling the $N$ points that make up the dataset, we also sample an additional 300 points that make up an extra held-out set which is unseen by any of the meta-learners, and whose sole purpose is for us to be able to estimate the actual generalisation risk of each posterior, which is what we report under "Generalisation Risk" in, *e.g.*, Figure 3. Furthermore, jointly with the $N + 300$ datapoints already sampled, we sample an *additional* 300 datapoints which form a "balance set". The sole purpose of the balance set is for us to check if the dataset is roughly balanced between positive and negative examples. If the prevalence of each class in the balance set is not $\approx 0.5$, then we discard the entire GP sample. Since the balance set is disjoint from the original dataset $N$ (and also the 300 datapoints forming the extra held-out set), doing this does not jeopardise the i.i.d. property within each dataset.[15] Approximately balancing the datasets in this way is convenient because it allows us to interpret results more easily, since the risk of the trivial classifier that always predicts the majority class in the observed dataset is $\approx 0.5$. We generate two disjoint meta-train sets (along with their corresponding meta-test sets) this way: one with $N = 30$ and another with $N = 60$. The meta-learners are either meta-trained and meta-tested exclusively with $N = 30$ or exclusively with $N = 60$.

## I.2  Deterministic Classification for Test Set Model

PAC-Bayes bounds naturally lead to stochastic classifiers (also known as Gibbs classifiers), whereby a fresh sample $h \sim Q$ is drawn whenever the classifier is presented with an input. However, this does

---

[15]The tasks themselves are also still i.i.d. from the same task distribution, although this does not affect the validitiy of our bounds, which only requires the i.i.d. assumption to hold *within* each dataset.

not need to be the case for *test set* bounds. In fact, it may be easier to bound the risk of a deterministic classifier with a test set bound than a stochastic one, since for a deterministic classifier, each term in the sum defining the empirical risk is a Bernoulli random variable, and hence it is trivial to apply Theorem 1, which leads to significantly tighter bounds than Theorem 2 in the small data regime. Additionally, we observed that when optimising $\frac{1}{T}\sum_{t=1}^{T}\overline{R}_{S_{t,\text{test}}}(Q_\theta(S_{t,\text{train}}))$ as in Section 4, the learned posterior map $Q_\theta$ eventually became essentially deterministic once meta-training was complete.

Another way to use the binomial tail test set bound in the case when $\ell \in [0, 1]$ (as it effectively is for Gibbs classifiers, once the zero-one loss is integrated over $Q$ to form $\mathbb{E}_{h\sim Q}[\ell_{0/1}((x, y), h)] \in [0, 1]$), is to randomise the computation of the empirical loss. In particular, for each $z \in S_{\text{test}}$, one could sample a Bernoulli random variable with parameter $\ell(z, h)$ and set the empirical risk in Theorem 1 to be the average of these Bernoulli random variables. For test set bounds, these are i.i.d. hence the sum is binomially distributed and Theorem 1 can be applied directly. We do not pursue this here, as it does not make a significant difference when the classifier is nearly deterministic, as we found.

For these reasons, at meta-*test* time we convert the test set bound meta-learners into *deterministic* classifiers by using a Bayes classifier instead of a Gibbs classifier. That is, the final predictor for a test set bound meta-learner with posterior $Q$ is given by
$$\hat{y}(x) := \text{sign}\left(\mathbb{E}_{w\sim Q}[w^\mathsf{T}\phi_\theta(x)]\right) \tag{88}$$
The risk of this predictor on a dataset $S$, which is the quantity we report and upper bound for the test set bound meta-learners in Section 4, is then simply the usual (non-Gibbs) risk: $R_S(\hat{y}) = \frac{1}{N}\sum_{(x,y)\in S}\ell_{0/1}((x, y), \hat{y})$. We emphasise that this change primarily serves to simplify the test set bound computation (and allow the use of the tighter Theorem 1 instead of just Theorem 2), and essentially does not affect the performance of the test set classifiers — the Gibbs and Bayes risks are nearly identical because the Gibbs classifier learned by the test set meta-learners was already nearly deterministic.

### I.3 Computing the Empirical Risk

In this section we provide additional details on how to compute the empirical risk term for the meta-learners. This applies for the PAC-Bayes meta-learners at both meta-train time and meta-test time, but only applies to the test set bound meta-learners during meta-train time — at meta-test time we use a Bayes classifier for the test set bound meta-learners instead of a Gibbs one; see Appendix I.2 for a discussion. Recall that we consider hypotheses of the form $h_w(x) := \text{sign}(w^\mathsf{T}\phi_\theta(x))$. Then the loss function is:
$$\ell_{0/1}((x, y), h_w) = \mathbb{1}[y \neq \text{sign}(w^\mathsf{T}\phi_\theta(x))] \tag{89}$$

We can then compute the empirical Gibbs risk as
$$\overline{R}_S(Q) = \frac{1}{|S|}\sum_{(x,y)\in S}\mathbb{E}_{w\sim Q}[\mathbb{1}[y \neq \text{sign}(w^\mathsf{T}\phi_\theta(x))]] \tag{90}$$

$$= \frac{1}{|S|}\sum_{(x,y)\in S}\text{Pr}\left(yw^\mathsf{T}\phi_\theta(x) < 0\right) \tag{91}$$

We now specialise to the case of Gaussian $Q := \mathcal{N}(\mu, \Sigma)$. In this case, we can compute the empirical Gibbs risk in Equation (91) in closed form (up to the error function, which has a standard implementation in PyTorch (Paszke et al., 2017)):
$$yw^\mathsf{T}\phi_\theta(x) \sim \mathcal{N}(y\mu^\mathsf{T}\phi_\theta(x), \phi_\theta(x)^\mathsf{T}\Sigma\phi_\theta(x)), \tag{92}$$

$$\text{Pr}\left(yw^\mathsf{T}\phi_\theta(x) < 0\right) = \Phi\left(\frac{-y\mu^\mathsf{T}\phi_\theta(x)}{\sqrt{\phi_\theta(x)^\mathsf{T}\Sigma\phi_\theta(x)}}\right) \tag{93}$$

where $\Phi$ is the standard normal cumulative distribution function and where we have used the fact that $y \in \{-1, +1\}$ so $y^2 = 1$. Now recall that $\Phi$ is related to the error function $\text{erf}(x)$ (as defined in PyTorch) by $\Phi(x) = \frac{1}{2}[1 + \text{erf}(\frac{x}{\sqrt{2}})]$, which gives:
$$\overline{R}_S(Q) = \frac{1}{|S|}\sum_{(x,y)\in S}\frac{1}{2}\left[1 + \text{erf}\left(\frac{-y\mu^\mathsf{T}\phi_\theta(x)}{\sqrt{2\phi_\theta(x)^\mathsf{T}\Sigma\phi_\theta(x)}}\right)\right]. \tag{94}$$

Hence we can backpropagate through the empirical Gibbs risk without the need for Monte Carlo integration over $Q$.

## I.4   Post-Hoc Optimisation of Posteriors

It is well-known that when performing amortised variational inference (VI) (Kingma & Welling, 2014), there is an *amortisation gap* (Cremer et al., 2018), which is the gap between the performance of the amortised inference network, and the performance obtained when optimising each variational problem separately. The meta-learners we consider in Section 4 have similarities with amortised VI, except that PAC-Bayes bound minimisation is being amortised, rather than VI. Similarly, there is an amortisation gap for our meta-learners, which is the gap between the bound obtained by the meta-learner when the posterior that was outputted by the posterior map is directly used, versus the bound we obtain when optimising, for each dataset, the posterior using gradient-based methods (in our case, ADAM (Kingma & Ba, 2015)). Optimising the posterior for each dataset individually is costly, but since we are concerned with obtaining the tightest bounds possible, we perform this *post-hoc optimisation* for all of our meta-learners (including the results reported in Section 4). Fortunately, each optimisation does not take too long, since we can initialise the posterior at the distribution output by the meta-learner.

So far, we have discussed post-hoc optimisation of the PAC-Bayes bound. However, we can also run post-hoc optimisation for the test set bound, *as long as the optimised posterior does not depend on the test set*. We consider post-hoc optimising the *train risk* for each dataset. In principle, this could possibly lead to overfitting the train set. In practice, we observe that this *improves* performance slightly for the MLP-NP (indicating that the MLP-NP test set meta-learner was underfitting the train set somewhat), and leaves performance essentially completely unaffected for the CNN-NP, because the train set risk is *already* essentially zero for the CNN-NP test set meta-learner before post-optimisation. Note that post-hoc optimisation is completely legal as a means of obtaining bounds — it does not affect the validity of the bounds we consider, but merely closes the amortisation gap.

As an ablation study, we can compare the performance of the meta-learners with and without post-hoc optimisation. Figures 7 and 14 show the performance of the CNN-NP and MLP-NP meta-learners *without* post-hoc optimisation, which should be compared to Figures 3 and 13, which show their performance *with post-optimisation*. Comparing Figure 7 with Figure 3 we see that post-hoc optimisation improves the performance of the PAC-Bayes meta-learners slightly but leaves the test set meta-learners essentially unaffected for the CNN-NP. Comparing Figure 14 with Figure 13, we see that post-hoc optimisation tightens the generalisation bounds for all meta-learners slightly. In conclusion, post-hoc optimisation sometimes leads to a small benefit, so we perform it for all meta-learners.

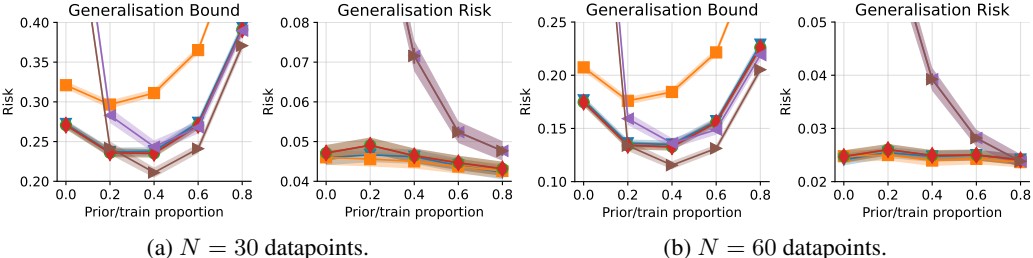

(a) $N = 30$ datapoints.                    (b) $N = 60$ datapoints.

Figure 7: Average generalisation bound and actual generalisation risk **for CNN-NP without post-hoc optimisation** ($\pm$ two standard errors) for Catoni (▼), PAC-Bayes-kl (■), optimisitc PAC-Bayes-kl (●), learned convex (◆), Chernoff test set bound (◄), and binomial tail test set bound (►). All bounds hold with failure probability $\delta = 0.1$ except for conjectured PAC-Bayes-kl which is not proven to be a valid generalisation bound.

## I.5   The Multilayer Perceptron Neural Process

We now describe the multilayer perceptron (MLP)-NP model, which is closely related to (but not identical with) the *conditional neural process* model first described in Garnelo, Rosenbaum, et al.

(2018).[16] When using the MLP-NP, we use an MLP to implement the feature map $\phi_\theta$. Additionally, each of the maps $Q_\theta/P_\theta$ consists of two MLPs: the *encoder* and *decoder*. The encoder $e_\theta$ maps $\mathbb{R} \times \{-1, +1\} \to \mathbb{R}^M$, where $\mathbb{R}^M$, $M \in \mathbb{N}$ is the *representation space*. The decoder $d_\theta$ maps $\mathbb{R}^M \to \mathcal{G}(\mathbb{R}^K)$, where $\mathcal{G}(\mathbb{R}^K)$ is the set of all Gaussian distributions over $\mathbb{R}^K$ (in practice, the decoder outputs a vector in $\mathbb{R}^{K+K(K+1)/2}$, which is converted into the mean of the Gaussian, and also the lower-triangular part of the Cholesky decomposition of the covariance matrix). When given a dataset $S$, the encoder computes a permutation-invariant representation of the dataset as $r_\theta(S) := \frac{1}{N} \sum_{(x,y) \in S} e_\theta((x, y))$. The decoder $d_\theta$ then computes a Gaussian posterior distribution over the hypothesis space as $d_\theta(r_\theta(S))$.

## I.6 The CNN-Based Gaussian Neural Process

In contrast to the MLP-NP, which uses MLPs to implement the feature map $\phi_\theta$, the CNN-Based Gaussian Neural Process (Bruinsma et al., 2021a) (CNN-NP) lets the $k^{\text{th}}$ component of the feature map be $\phi_{\theta,k}(x) = \exp(-\frac{1}{2\ell^2}(x - x_k)^2)$, a Gaussian basis function centred at some fixed input $x_k$, with a learnable lengthscale $\ell$. The centres of the Gaussian basis functions $(x_k)_{k=1}^K$ are evenly spread out through the interval $[-2, 2]$. The CNN-NP lets $Q_\theta$ and $P_\theta$ be parametrisations of maps from datasets to full-covariance Gaussian posteriors over the weights of these basis functions where the maps incorporate a symmetry called *translation equivariance*: if all inputs of the observed data are shifted by some amount, then the posterior over the weights for the basis functions should be shifted accordingly. Translation equivariance enables the CNN-NP to use CNNs for $Q_\theta$ and $P_\theta$ instead of MLPs.

We now give a brief high-level description how the CNN architecture for the posterior mean of the Gaussian works. This follows the way that the mean of the *Convolutional Conditional Neural Process* (ConvCNP) is computed,[17] and we refer the reader to Sec 4 and especially Fig 1 of Gordon et al. (2020) for a full description. First, the dataset is embedded into a 1D function with two channels, known as the *data channel* and the *density channel*. This 1D function is then evaluated on a discretised grid, and then fed into a CNN. The CNN output then defines mean of the Gaussian predictive distribution over functions. However, unlike in Gordon et al. (2020) and Bruinsma et al. (2021a), we modify this setup slightly, so that, instead of interpreting the CNN output as the mean of the Gaussian predictive over functions, it is interpreted as the mean of the Gaussian posterior over *weights* of the basis functions in $\phi_\theta$. Defining the posteriors in weight space instead of function space makes it much easier to compute the KL-divergence.

We also give a brief description of how the CNN architecture for computing the posterior *covariance* works. As this computation is more involved than the computation for the mean, we refer the reader to Sec 3 and App E.2 of Bruinsma et al. (2021a) for a detailed description, `https://github.com/wesselb/NeuralProcesses.jl` for a full implementation, and Bruinsma et al. (2021b) for a useful visual description of the Gaussian neural process architecture, on which we base our CNN-NP architecture used in Section 4. To compute the covariance matrix of the weights of the basis functions, the dataset $S$ is first embedded into three images on $[-2, 2] \times [-2, 2]$. The embedding is performed by placing a Gaussian basis function[18] corresponding to each datapoint along the *diagonal* of the $[-2, 2] \times [-2, 2]$ square. These three images are known as the *data channel*, *density channel* and *source channel* respectively. As explained in Bruinsma et al. (2021a), the data channel incorporates information about the $y$-values of the observations in $S$, the density channel records information about how many points in $S$ are observed at any particular $x$-location, and the source channel is simply in the shape of an identity matrix which, intuitively speaking, allows CNN-NP to begin with a "white noise" covariance matrix that afterwards is modulated to include correlations. These continuous images, after being appropriately discretised on a regular 2D grid[19] are passed through a 2D CNN, which outputs an image which is interpreted as a covariance matrix over the interval $[-2, 2]$. In order

---

[16]The original conditional neural process outputs a Gaussian distribution directly in function space. This leads to complications when considering the KL term in the PAC-Bayes bounds, hence we modify it to output a Gaussian distribution over the parameters of a linear model.

[17]The predictive *mean* of the ConvCNP (Gordon et al., 2020) and that of the later, full-covariance Gaussian Neural Process (Bruinsma et al., 2021a) are computed in the same way.

[18]These basis functions are distinct from the basis functions used to define the feature map $\phi_\theta$.

[19]This discretisation need not be the same as the spacing used for the Gaussian basis functions which make up the feature map $\phi_\theta$.

to ensure that the covariance matrix output is positive semi-definite, we multiply the output by itself: $\Sigma := MM^\mathsf{T}$, where $\Sigma$ is the covariance matrix and $M$ is the $K \times K$ matrix output by the CNN. This covariance matrix is finally interpolated onto the grid defined by the locations of the basis functions in $\phi_\theta$, which then defines the covariance of the weights of the basis functions.

### I.7  Hyperparameters

**General meta-learner training details.** We fix the failure probability at $\delta = 0.1$ for all of the meta-learning experiments. During meta-training, for the PAC-Bayes models we found it was more numerically stable to optimise the logarithm of the objective described in Equation (20). In particular, for the Catoni bound model, we used the numerically stable implementation of $\log(1-e^{-x})$ referenced in `https://github.com/pytorch/pytorch/issues/39242`. For all meta-learners, we use a mini-batch estimate of the objective in Equation (20), with a batch size of 16 datasets. We use a weight decay of $1 \times 10^{-5}$ for all meta-learners.

**MLP-NP hyperparameters.** We use a relatively large architecture for the MLP-NP, as we found during preliminary experiments that larger architectures performed better. The feature dimension of the linear model (see Appendix I.5) was set at $K = 256$. The MLPs implementing the feature map $\phi_\theta$, encoder $e_\theta$ and decoder $d_\theta$ all had two hidden layers, each with a width of $512$. The MLP-NP was trained for $100$ epochs on the meta-train set, with a learning rate of $2 \times 10^{-5}$ (we found that higher learning rates could lead to instabilities during training) with the ADAM optimiser (Kingma & Ba, 2015). We did some manual hyperparameter tuning to choose these hyperparameters, but they were not selected exhaustively. To avoid overfitting to the meta-train set when doing manual hyperparameter tuning, we also sampled a meta-validation set of datasets, which we used when tuning hyperparameters.

**CNN-NP hyperparameters.** For the CNN in the architecture, we use a U-Net (Ronneberger et al., 2015). The U-Net, we use has $12$ layers, with the number of channels in each layer being $8, 16, 16, 32, 32, 64, 64, 64, 64, 32, 32, 16$ respectively. This architecture matches that used by Gordon et al. (2020). The number of Gaussian basis functions per unit of input space (which determines the number of features in $\phi_\theta$) was set at $16$. The discretisation of the Gaussian Neural Process (*i.e.*, the spacing at which the continuous representation is evaluated before passing it through the CNN) is set at $32$ points per unit. Then CNN-NP was trained for $10$ epochs on the meta-train set, with a learning rate of $1 \times 10^{-3}$ with the ADAM optimiser. We did very little manual hyperparameter tuning for the CNN-NP, because we found that it was fairly robust to the choice of learning rate and basis function spacing. In all cases, the CNN-NP optimised much more quickly than the MLP-NP.

**Post-hoc optimisation.** We perform post-hoc optimisation at meta-test time, as discussed in Appendix I.4. Given a dataset, we initialise the posterior at the Gaussian distribution which is output by the NP. We then use the ADAM optimiser (Kingma & Ba, 2015) with a learning rate of $3 \times 10^{-4}$ for a maximum of $3\,000$ optimisation steps to target either the PAC-Bayes bound (for PAC-Bayes meta-learners), or the train risk (for test set meta-learners). If, after $100$ optimisation steps, the generalisation bound has not decreased by at least $0.0001$, then the optimisation is ended early.

**Compute.** We used roughly 500–1000 GPU-hours divided NVIDIA Tesla V100 and GeForce RTX 2080 graphics cards using both an internal cluster and Amazon Web Services. Most of the computational budget was spent on the meta-learning experiments. Of these, the MLP-NP was more costly to run than then CNN-NP, since it took longer to train.

## J  Additional Plots for Synthetic Classification

### J.1  Predictive Distributions

In this appendix we include extra plots of 1D classification tasks from the meta-test set, similar to Figure 4.

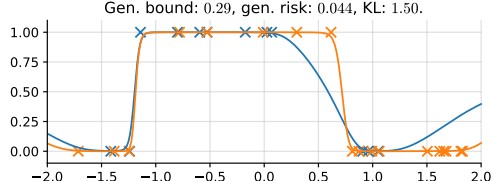 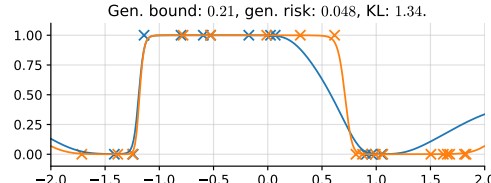

(a) **PAC-Bayes-kl** bound with data-dependent prior, showing the prior (—) and posterior (—) predictive, prior set (✕) of size 12 and risk set (✕) of size 18.

(b) **Catoni** bound with data-dependent prior, showing the prior (—) and posterior (—) predictive, prior set (✕) of size 12 and risk set (✕) of size 18.

Figure 8: Predictions and bounds on one of the 1D datasets with **30 datapoints**. The bounds hold with failure probability $\delta = 0.1$.

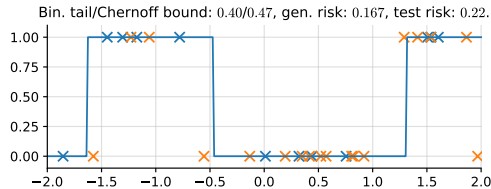 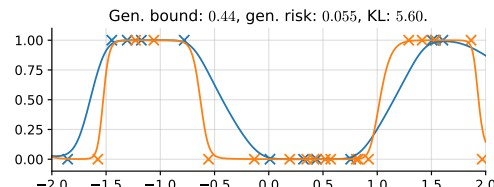

(a) **Test set** bounds, showing the learned hypothesis, (—), the train set (✕) of size 12 and the test set (✕) of size 18.

(b) **Catoni** bound with data-dependent prior, showing the prior (—) and posterior (—) predictive, prior set (✕) of size 12 and risk set (✕) of size 18.

Figure 9: Predictions and bounds on one of the 1D datasets with **30 datapoints**. The bounds hold with failure probability $\delta = 0.1$.

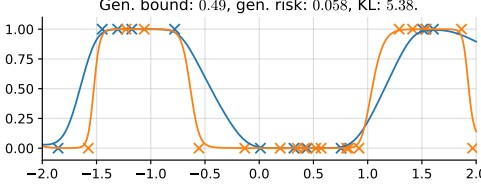 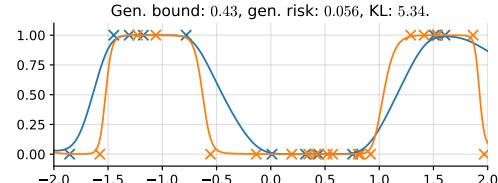

(a) **PAC-Bayes-kl** bound with data-dependent prior, showing the prior (—) and posterior (—) predictive, prior set (✕) of size 12 and risk set (✕) of size 18.

(b) **Learned convex** bound with data-dependent prior, showing the prior (—) and posterior (—) predictive, prior set (✕) of size 12 and risk set (✕) of size 18.

Figure 10: Predictions and bounds on one of the 1D datasets with **30 datapoints**. The bounds hold with failure probability $\delta = 0.1$.

### J.2 Performance of MLP-NP

In the main body, we considered the CNN-NP model, since it performed better while training much faster and requiring fewer parameters then the MLP-NP. In Figures 13 and 14 we also show the performance of the MLP-NP for the test set meta-learners and also the Catoni bound meta-learner, both with and without post-hoc optimisation (see Appendix I.4). We see that the MLP-NP test set meta-learner performs very similarly to the CNN-NP one when $N = 30$, but performs slightly worse when $N = 60$. The MLP-NP Catoni meta-learner is either as tight as the CNN-NP Catoni meta-learner, or slightly looser, except when $N = 30$ and the prior proportion is 0 or 0.8, in which case the MLP-NP seems to have encountered learning difficulties. Also note that generalisation risk is generally higher for the MLP-NP than the CNN-NP.

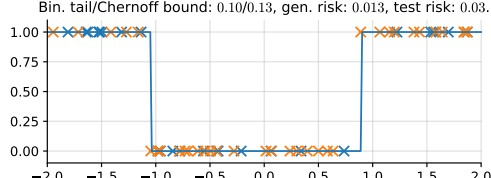

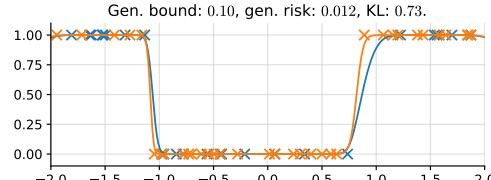

(a) **Test set bounds**, showing the learned hypothesis, (—), the train set (✗) of size 24 and the test set (✗) of size 36.

(b) **Catoni** bound with data-dependent prior, showing the prior (—) and posterior (—) predictive, prior set (✗) of size 24 and risk set (✗) of size 36.

Figure 11: Predictions and bounds on one of the 1D datasets with **60 datapoints**. The bounds hold with failure probability $\delta = 0.1$.

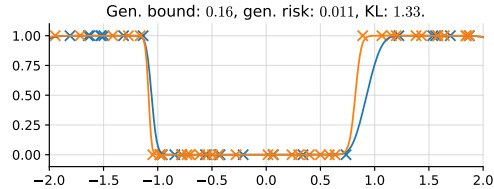

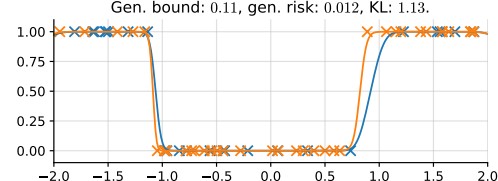

(a) **PAC-Bayes-kl** bound with data-dependent prior, showing the prior (—) and posterior (—) predictive, prior set (✗) of size 24 and risk set (✗) of size 36.

(b) **Learned convex** bound with data-dependent prior, showing the prior (—) and posterior (—) predictive, prior set (✗) of size 24 and risk set (✗) of size 36.

Figure 12: Predictions and bounds on one of the 1D datasets with **60 datapoints**. The bounds hold with failure probability $\delta = 0.1$.

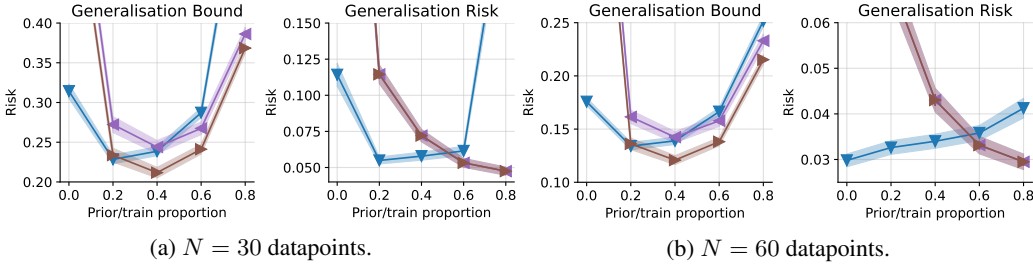

(a) $N = 30$ datapoints.

(b) $N = 60$ datapoints.

Figure 13: Average generalisation bound and actual generalisation risk **for MLP-NP with post-hoc optimisation** ($\pm$ two standard errors) for Catoni (▼), Chernoff test set bound (◀), and binomial tail test set bound (▶). All bounds hold with failure probability $\delta = 0.1$.

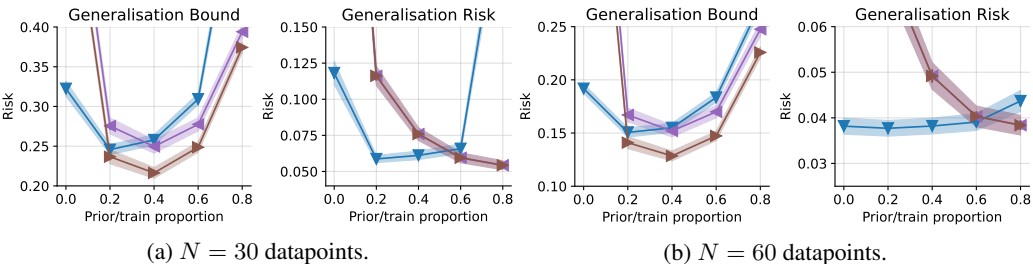

(a) $N = 30$ datapoints.

(b) $N = 60$ datapoints.

Figure 14: Average generalisation bound and actual generalisation risk **for MLP-NP without post-hoc optimisation** ($\pm$ two standard errors) for Catoni (▼), Chernoff test set bound (◀), and binomial tail test set bound (▶). All bounds hold with failure probability $\delta = 0.1$.