# OpenReview forum: "How Tight Can PAC-Bayes be in the Small Data Regime?"
_NeurIPS.cc/2021/Conference — NeurIPS 2021 Poster_

### Official Review · Reviewer_15BA · 2021-07-14

**Rating:** 6
**Confidence:** 4

**Summary:**

The paper first proves that, regarding the choice of convex function Delta in PAC-Bayes generalization bounds (Theorem 3), the family of Delta as in Catoni 2007 always contains a function that minimizes the upper bound on expected risk (Theorem 4). It then proves that for all convex functions Delta, the expectation of the upper bound on expected risk is bounded below (Corollary 3). Finally, the paper studies a meta learning situation (in which some parameters of Delta are fitted based on data) and reports the gap between held-out test error and value of the generalization bound.

**Limitations And Societal Impact:**

A direction for future work is that, does Theorem 4 mean that the best option is to use Catoni’s definition of Delta, allowing for different betas using some union bound argument?

Regarding potential negative societal impact, PAC-Bayes bounds are used to derive/certify the quality of data-driven classifiers, so they have the potential to make unfair classification decisions against protected populations.

**Main Review:**

I have concerns about the clarity of the paper.

Originality. The paper chooses to address an important and unsolved theoretical problem, which is characterizing how Delta, a design choice in PAC-Bayes, affects the best possible generalization bound. Applied researchers have experimented with different families of Delta to attain the best numerical results (generalization bound versus test error). Before experimentation, it is not clear which Delta should be preferred. A theoretical result of the flavor “it is sufficient to search in this/that family of Delta” would be a good tool, since it narrows down the set of options over which a practitioner must search. The use of Figure 2 as an illustration for Theorem 4 is very helpful.

Clarity/quality. My biggest concern is that I am not sure what “meta-learning the tightest bounds” in Section 4 mean. As a result, I do not understand how the numbers being plotted Figure 3 were generated and how to interpret them. It would be good if the paper provided more details to distinguish PAC-Bayes in the meta-learning setting with PAC-Bayes in the classical iid setting. For instance, Equation 6 shows that the p_Delta quantity from iid PAC-Bayes is a high-probability upper bound on the risk of a stochastic classifier drawn from Q. Is there an analog derivation for Equation 19 i.e., is the quantity an upper bound for some classifier’s risk? Lines 259-260 appears to be a big departure from the iid PAC-Bayes case. The convex function Delta here has parameters that depend on the data in the meta-learning task. This seemingly contradicts with the comment in line 147, which states that Delta cannot depend on data (this is also clear from Theorem 3, in which Delta needs to be fixed before seeing data). For Figure 3, if the goal is comparing generalization bounds with test error, it would be better to plot the two curves on the same panel / plot the difference between the two of them. Currently, it is hard to compare the generalization bound and test error because they are plotted on different panels and the limits/scales of the y axes do not align with each other. I have some smaller comments. In Equation 7, instead of kl(q, .), it should read barR(Q). The notation of small q is rather confusing, and out of scope (since it’s only in Theorem 4 that q is defined to be equal to barR(Q)). Overall, it might be better to remove the notation q and KL --- currently there is an inconsistency in Equation 7, since it’s using both q and KL(Q || P). The notation barR(Q) conveys the dependence of the empirical risk on data better than just q.

Significance. Theorem 4/Corollary 3 are exciting theoretical results, of use to researchers searching for PAC-Bayes bound that closely approximate test error from above.

Updates after initial rebuttal: The authors gave a concrete way of improving clarity in Section 4.


**Time Spent Reviewing:**

3.5

---

> ### Author Response · Authors · 2021-08-09
> **Response to 15BA**
>
> Thank you for spending significant time reviewing our paper. We are pleased to hear that you find Theorem 4 / Corollary 3 exciting results and of use to researchers looking for tight bounds.
>
> As we understand from your review, your main concern is Section 4, “Meta-Learning the Tightest Bounds for Synthetic Classification”. In the next section, we briefly clarify the meta-learning setup and how it connects to the usual i.i.d. setting. If our clarification of section 4 resolves your questions, we hope that you will reconsider your score.
>
> # Clarification of Section 4
>
> In Section 4, we consider a large collection of independent one-dimensional classification tasks (a multitude of data sets) with similar statistical structure, where each individual task is in the classical i.i.d. setting. The goal of the section is, for any particular one of those tasks, to produce a generalisation bound which is valid and as tight as possible. [See “Task Distribution”.] For the PAC-Bayes bounds, we do this as follows: To begin with, we use a data-dependent prior, which means that, in any of the tasks, we use part of the data to generate a prior. [See “Priors”.] The mapping from part of the data to the prior for the PAC-Bayes bound is implemented with a _neural process_, which depends on some parameters. [See “Parametrising the Meta-Learning and Hypothesis Space”.] But how do we find these parameters? To train the neural process, we use part of the large collection of independent tasks as “training tasks”—we call these the _meta-train set_. [See “The Meta-Learning Objective”.] After the neural process is trained, we take any task not in the meta-train set—we call this remainder the _meta-test set_—run the neural process on part of the data of that task to generate a prior, and use the other part of the data of the task to produce a valid PAC-Bayes bound. In other words, after the neural process is trained, it functions as a data-dependent prior in the classical i.i.d. setting and _produces valid bounds for each individual dataset_. We emphasise, however, that to train the neural process (and consequently produce good data-dependent priors), you must have access to a large collection of independent tasks with similar statistical structure which you can use as training data and do not require valid generalisation bounds for.
>
> Having clarified the setup of Section 4, let us now respond to your particular questions about the section.
>
> > “As a result, I do not understand how the numbers being plotted [in] Figure 3 were generated and how to interpret them.”
>
> To generate the left sides of Figures 3(a) and 3(b), we produced generalisation bounds for every task in the meta-test set. As a measure of performance, we then computed the average of these bounds where the uncertainty intervals show the error on that average. On the $x$-axis, we vary the proportion of data used for the data-dependent prior (or proportion used for the train set, for the test set bounds). Additionally, for every task, we held out some extra data to closely approximate the true generalisation risk. Averages of the generalisation risks (and corresponding errors of the averages) are shown on the right sides of Figures 3(a) and 3(b).
>
> > “It would be good if the paper provided more details to distinguish PAC-Bayes in the meta-learning setting with PAC-Bayes in the classical iid setting.” “Lines 259-260 appears to be a big departure from the iid PAC-Bayes case.”
>
> The meta-learning setting that we consider consists of many instances of the classical i.i.d. setting. These many instances are only necessary to train the neural process. After the neural process is trained, we can produce valid PAC-Bayes bounds for single instances of the classical i.i.d. setting which are not seen during training.
>
> > “Is there an analog derivation for Equation 19 i.e., is the quantity an upper bound for some classifier’s risk?”
>
> We solely use (19) to train the neural process. Because the prior and the learned convex delta then depend on all the training data, which is everything in the meta-train set, we are unable to produce valid generalisation bounds for the meta-train set. For the meta-test set, however, the model produces completely valid bounds.
>
> > “For Figure 3, if the goal is comparing generalization bounds with test error, (...) with each other.”
>
> The goal of this section is not to compare generalisation bounds with generalisation risks. Rather, in this section, we are concerned with producing generalisation bounds which are as tight as possible. In particular, we compare two ways to produce tight generalisation bounds: (1) PAC-Bayes bounds with a data-dependent prior and (2) traditional test set bound for some held-out data. Therefore, in the left sides of Figures 3(a) and 3(b), the feature of interest is how the purple and brown lines (test set bounds) compare with the other colours (PAC-Bayes bounds).
>
> # Other Comments
>
> Thank you for catching the out-of-place $q$. That should indeed be $\bar{R}_S(Q)$. We will change that in the writing.
>
> Finally, you are right that Theorem 4 implies that considering the Catoni bound where the parameter is optimised with a union bound gives the best possible PAC-Bayes bound within the generic PAC-Bayes theorem up to the slack introduced by the union bound. It would be even better if we wouldn’t have to consider a union bound; see also Open Problem 1. Optimising the parameter of the Catoni bound was considered by Catoni; see page 13 of his monograph (2007).

---

> > ### Comment · Reviewer_15BA · 2021-08-24
> > **Update in score**
> >
> > I appreciate the authors' engagement with my reviews. The clarification was helpful. I have increased my score.
> >
> > One suggestion to improve clarity of Section 4 is to distinguish between the paper's meaning of "meta-learning" with the usual "meta-learning" (in the sense of Amit and Meir [2018], for instance) at an earlier point in Section 4. I understand that bringing up "meta-learning" is useful because it conveys the situation where there are many iid task distributions of interest. But as the authors say in lines 275-276, the goal here is not to make good predictions on an unseen task. Rather, the many iid tasks serve to capture the sampling variation of the generalization bounds and the generalization risks of various algorithms.

---

### Official Review · Reviewer_d92V · 2021-07-16

**Rating:** 6
**Confidence:** 4

**Summary:**

This paper investigates the relative tightness of PAC-Bayesian and test set bounds (Langford, 2002) in the small data regime for losses bounded in the unit interval.


**Limitations And Societal Impact:**

Addressed by the authors. I do not foresee any potential negative societal impact of this work.

**Main Review:**

Focusing on the small dataset regime, the paper investigates the relation between test set bounds (Langford, 2002) and PAC-Bayesian bounds, notably, (1) the family of bounds due to Catoni (parameterized by $\beta$), (2) Maurer's PAC-Bayes-kl-inequality and (2) the generic PAC-Bayesian bound due to Germain et al (2009), Begin et al (2016) that involves a free parameter (convex function $\Delta$). Maurer's (non-parametric) bound behaves like Catoni's bound (and it's relaxation, McAllester's linear bound) but holds uniformly for all $\beta$ at the cost of a $O\left(\tfrac{\ln \sqrt{n}}{n}\right)$ term that comes by way of a union bound argument. It might benefit the reader to clearly state this fact just after the statements of Corollaries 1 and 2.

For a fixed dataset, the authors show that Catoni's bound achieves the tightest version of the generic PAC-Bayesian bound (involving the free parameter $\Delta$). On the other hand, in the random dataset setting, Catoni's bound is not optimal. I checked the proof of Theorem 4 and it appears correct. The authors also experimentally demonstrate in a controlled setting that PAC-Bayesian bounds are competitive with the Chernoff test set bound.

The analysis in this paper applies only to loss functions bounded in the unit interval. Though the authors acknowledge this as a limitation and hint at the potential of extending the analysis beyond some of the restrictions (l.344-349), it is not immediately clear, for example, if the analysis can be extended to unbounded losses. As a minor gripe, this comes about as a bit unsatisfactory given that there are well-known “generic” PAC-Bayes inequalities for unbounded losses to start with (see, e.g., Theorem 2.1 in Tong Zhang, 2006 that recovers Catoni's bound when restricted to the 0-1 loss; http://tongzhang-ml.org/papers/it06-bound.pdf). Overall, I found the paper well-written with a clear motivation albeit with only incremental contributions.

**Time Spent Reviewing:**

3

---

> ### Author Response · Authors · 2021-08-09
> **Response to d92V**
>
> Thank you for spending considerable time to review our paper.  We appreciate your suggestion to include a remark that Catoni’s bound holds uniformly for all $\beta$ at the cost of a $O(\log(\sqrt{n}) / n)$ term, which we will add to the writing after the statements of Cors 1 and 2.
>
> We understand that your main criticism is that the paper is too restricted in scope because we only consider bounded loss functions. While we agree that extensions of our analysis to unbounded loss functions would be very interesting, bounded losses are in line with very large portions of the literature on PAC-Bayes. Moreover, unbounded losses introduce technical challenges which cannot be immediately resolved using the approach we take. Please see the section below. Finally, the criticism that the paper is too restricted in scope was not shared by the other reviewers. On the contrary, R2 “[is] leaning to say that this work is likely to generate fruitful discussions and follow-up works.”, and R4 thinks that “Theorem 4/Corollary 3 are exciting theoretical results, of use to researchers (...)”. In light of this, we hope that you will reconsider your score.
>
> # Unbounded Losses
>
> To generalise our analysis to unbounded losses, we could try to generalize the proof technique introduced by Germain, 2009 to unbounded losses. Following the proof in Begin et al., 2016, we note that the assumption that the loss is bounded is not used until the second-to-last step “Binomial law” in Figure 1. Notably, this step makes use of the form of an expectation for binomial random variables. Our analysis crucially relies on this form, and therefore cannot be readily generalised to unbounded random variables.
>
> Thank you for the reference to Zhang, 2006. While theorem 2.1 in Zhang appears to be of interest and perhaps could be the starting point of an analysis for unbounded losses, it has a different form than the theorem from Germain et al 2009 we begin with. In particular, there is no general convex function $\Delta$ that defines a class of bounds (one for every convex $\Delta$). Our paper investigates which convex function in the bound due to Germain, 2009 leads to the tightest PAC-Bayes bound. Since the bound from Zhang, 2006 does not have such a $\Delta$, we cannot perform our study in the paper for the bound from Zhang, 2006. Perhaps the Zhang, 2006 bound could be generalised to include such a convex function $\Delta$, but deriving a new bound of this form as well as analysing it is beyond the scope of this paper.

---

### Official Review · Reviewer_4Rmy · 2021-07-19

**Rating:** 8
**Confidence:** 5

**Summary:**

As far as I see, this submission is motivated by the desire to understand the ability of PAC-Bayes bounds to give tight numerical values such that they could be used as numerical measures intended to certify the post-training performance of randomised classifiers. The submission leverages the so-called generic PAC-Bayes theorem which is stated in terms of an arbitrary convex function $\Delta : [0,1]\times[0,1] \to \mathbb{R}$ and which can be seen as the basis for deriving the known PAC-Bayes bounds (the one of McAllester, the one of Langford and Seeger, the parametric family of bounds of Catoni, possibly others) as shown by Germain et al. [2009].

In particular, the submission compares theoretically the best bounds that could be hoped for when using the parametrised family of bounds of Catoni (there's one such bound for each $\beta>0$), and the generic bound for a given $\Delta$. Then it also discusses what's the best one can hope for when optimising $\Delta$ from a class of convex functions. Then in some empirical studies with (artificial) datasets of small size the values of the bounds are evaluated and compared. For individual (as opposed to randomised) predictors, the values of two "test set bounds " are also shown: the binomial tail inversion bound and the Chernoff relative entropy bound. If I understand correctly, the intended goal is to compare the tightness of the PAC-Bayes bounds (for randomised predictors) with the tightness of the test set bounds (for individual predictors). The experiments culminate with a setting where the parameters of the bounds are meta-learned, and the claim is that in this setting the PAC-Bayes bounds give better results than the test set bounds. The discussions, however, suggest settings in which test set bounds, in particular the so-called binomial tail bound, dominate.

**Limitations And Societal Impact:**

Sure.

**Main Review:**

ORIGINALITY

The generalisation bounds (PAC-Bayes and other) that appear in this paper are all known. An novel thing that this submission does is leverage the generic PAC-Bayes theorem which holds for an arbitrary convex function $\Delta$ of two arguments, and convex analysis tools (convex duality) in order to reason about the best $\Delta$ that could be used and the tightness of the resulting bound. The related literature is acknowledged (some comments on the cited literature of missing references are pointed below).

SIGNIFICANCE

I am leaning to say that this work is likely to generate fruitful discussions and follow-up works. I admire the effort to use sound theoretical analysis in order to reason about generalisation of learning methods using a language and experimental demonstrations that may appeal to the less theory-minded part of the machine learning community, it could have a positive effect of leading by example.

CLARITY / QUALITY /  READABILITY

The paper is relatively well written and the organisation is okay. Some parts require attention to make them more clear/readable. A few parts require mode delicate attention to rule out sources of possible confusion or misleading the community. My comments are below, together with some questions and there is also more extensive feedback of editorial nature.

EDITORIAL FEEDBACK

Comment on the abstract: On a first reading of this abstract I had the impression that it is a bit overloaded and it did not clearly summarise the paper. After a pass at the whole paper, I think the abstract is a bit misaligned. Try to make it concise and simple while keeping it aligned with the paper, essentially address these questions in the abstract: What problem the paper is about (from the title, it looks that the paper is about tightness of PAC-Bayes bounds in the small data regime); how does the paper approach the said problem (e.g. by analytical and numerical comparison with other kinds of generalisation bounds); what are the most salient observations (e.g. in one setting bound A is tighter than B, but in another setting the reverse order holds); and explicitly advertise the novelties (if any) here. Also, on the note of simplicity, or as matter of style, I recommend typing in (last)names rather than using \cite in the abstract, and keeping citations to a minimum. The last point can be accommodated e.g. like ​this: "generic PAC-Bayes theorem shown by Germain and others." and "family of PAC-Bayes bounds considered by Catoni." and "the family of bounds of Catoni can be suboptimal." and "proof framework of the generic PAC-Bayes theorem, we establish [...]"

L.32: " significantly harming post-training performance due to a reduced training set size."

L.33: this sentence calls for a reference, Perez-Ortiz et al. [2020] is relevant here.

L.36: As fas as I know, the generic PAC-Bayes theorem is originally due to Germain et al. [2009], while others (e.g. Begin et al. [2016] and Rivasplata et al. [2020]) restructured the presentation of the proof or gave extensions to other divergences (Begin et al.) or to setting with relaxed restrictions (Rivasplata et al.) hence it would be more accurate and fair to write something like "remarkably, Germain et al. [2009]  showed that a wide range of bounds can be obtained as special cases of a single \emph{generic PAC-Bayes theorem} that captures the central ideas of many PAC-Bayes proofs [see also Begin et al., 2016 and Rivasplata et al., 2020]."

L.39: "holds for any convex function $\Delta : [0,1]\times[0,1] \to \mathbb{R}$."

L.40: After "Catoni [2007]" insert "and other bounds"

L.41: "to this setup."

Warning note: Two distinct types of applications of PAC-Bayes bounds are: (1) using such a bound for evaluating a numerical risk certificate for (the randomised predictor defined by) some distribution over hypotheses learned by some method; and (2) using such a bound as optimisation objective by declaring to learn a distribution over hypotheses by minimizing the bound. The second kind of use is possible because PAC-Bayes bounds hold simultaneously (i.e. uniformly) for all distributions over a given hypothesis class. See e.g. the corresponding discussion given by Tolstikhin & Seldin [2013] or by Thiemann et al. [2017]. It appears that this submission is not making a clear distinction between these two uses, and perhaps conflating them when claiming that PAC-Bayes allows to use of all the available data that avoid splitting the data while at the same time getting a certificate on the post-training performance (see corresponding discussion given by Perez-Ortiz et al. [2020]). This must be addressed for the sake of clarity and to avoid misleading follow-up research.

L.46-47: This passage needs clarification/elaboration. Risk bounds (aka generalisation bounds, including PAC-Bayes bounds) studied in statistical learning theory are inequalities that hold with high probability with respect to the random choice of datasets (of a given size). Hence reading "fixed dataset" here raises a question mark that begs for clarification. On the other hand, in the typical problems studied by the machine learning community, datasets are fixed (e.g. MINIST, CIFAR-10, ImageNet) and there is the question of whether or under what cases the assumptions of the risk bounds (e.g. i.i.d. data, or at least randomly generated data) are met or can be considered to be reasonable for a given data set so that using the high-probability bounds is justified.

L.48: Replace "stochastic dataset" with "random dataset"

Reason: "random" refers to something that is picked according to a given *fixed* probability distribution (e.g. a random digit, or a random matrix of size 10x10) while "stochastic" in my mind refers to a *changing* probability distribution, e.g. changing with respect to a time parameter (e.g. a Brownian motion) or a space parameter (e.g. a Poisson point process). In your setting, it appears that the assumption on the dataset is that it is generated randomly according to a fixed (but unknown) data-generating distribution $D$. Accordingly, calling it a "random dataset" is preferable. (Alternatively, justify why to call it stochastic.)

L.51: A bit surprising to read that a neural network parametrises a convex function.

L.52: " bound of Langford and Seeger, and relaxes" (space-saver, since the year has been acknowledged already,)

L.53: insert a reference for the Chernoff test set bound (or insert "(see Theorem 2 below)" since it provides the reference)

L.54-60: by the end of reading this paragraph it should be clear to the reader whether you are using bounds for numerical evaluation of certificates only, or if you are using the bounds for learning (by bound-minimisation) and for numerical evaluation of the certificate. In other words, what I missed in this introduction is a high-level description of what kinds of predictors are being considered, for which the certificates on post-training performance are applied.

L.62 (and rest of the paper): I recommend the capital calligraphic $\mathcal{X}$ and $\mathcal{Y}$  for the sets that encompass all possible inputs and all possible labels (resp.), and $\mathcal{Z} = \mathcal{X}\times\mathcal{Y}$ for the set of all possible input-label pairs. Then this will be consistent with your use of $\mathcal{H}$ for the set of all possible hypotheses, etc.

L.77-78: "For the zero-one loss $\ell(h,(x,y)) = \mathbb{I}[h(x) \neq y]$ we have the binomial random variable [..] with parameters [...]"
Also, the number of trials should be "$N_{\mathrm{test}}$" I think?

L.81: the sum should start from 0?

L.86: For the same reason explained above, I recommend replacing "stochastic" with "randomised"
(regardless of what the PAC-Bayes literature has done before, unfortunately)

L.87: As mentioned above, a claim "does not require discarding data" needs to be supported (clarification or reference(s))

L.88-89: "Germain et al. [2009] prove a very general form of the PAC-Bayes theorem which encompasses many of these [see also Begin et al., 2016 and Rivasplata et al., 2020]."

L.90-21: The part of taking supremum over the risk is special to Begin et al. I think
(i.e. Germain et al. [2009] did not do this last step as fas as I remember)

L.96: this theorem needs to clarify that the probability is over the random choice of dataset $S$ of size $N$.
(i.e. could replace $\operatorname{Pr}$ with $D^{\otimes N}$, since the probability is w.r.t. distribution of the size-$N$ random sample)

L.97-98: I did not understand this remark. Note that the choice $\Delta(x,y) = 2(x-y)^2$ is a possible choice for $\Delta$, which gives the classical form of the PAC-Bayes bound, i.e. the one that resembles the bound of McAllester.

L.99: Germain et al. [2009] discussed how Catoni's bound follows from their generic PAC-Bayes theorem, hence in this case I feel that the attribution to Begin et al. is a bit misleading.

L.103-104: "we obtain the bound of Langford and Seeger [2001], also called the PAC-Bayes-kl bound [e.g. Seldin, 2012], but with the slightly sharper dependence on $N$ established by Maurer [2004]:"

L.105: Note that in fact this inequality holds for all $N \geq 1$, the small $N$ cases have been verified numerically
(see "Risk bounds for the majority vote: From a PAC-Bayesian analysis to a learning algorithm")

L.106: Why not refer to this, more suggestively, as the PAC-Bayes-kl bound?
(I am objecting the use of the acronym MLS due to being uninformative and potentially misleading)

L.107: The accurate thing to comment is that Corollary 2 is a special case of Theorem 3 in which the $\mathcal{I}_N(\Delta)$ has been upper-bounded by $2\sqrt{N}$ using Stirling's formula, as shown by Maurer [2004]. It is misleading to say that Corollary 2 is looser than Theorem 3, since the theorem is generic and the $\mathcal{I}_N(\Delta)$ has not been controlled.

L.108-110: Again, the two different uses of PAC-Bayes bounds are being conflated in this passage.
Also, the list of references in the square brackets should start with Langford and Caruana 2001 and end with Perez-Ortiz et al., 2020
(Note that the publication year of "(Not) Bounding the True Error" is 2001, regardless that Google Scholar incorrectly gives 2002)

L.112: [McAllester, 1999, 2003]

L.123: replace "value" with "form"

L.132-133: There are significant differences between the three references that are worth bringing to attention: Langford & Caruana [2001] trained a randomised neural network classifier by some method (which has nothing to do with PAC-Bayes, it was by some sort of sensitivity analysis) and used the PAC-bayes-kl bound to evaluate a numerical bound on the risk of this classifier; while Dziugaite & Ropy [2017] trained a randomised neural net classifier by a PAC-Bayes method (a training objective based on McAllester's bound) and used the PAC-Bayes-kl bound to compute a numerical bound on the risk of the classifier, so the last step is identical to the corresponding step done previously by Langford & Caruana, but the novelty was that D&R demonstrated *non-vacuous* bound values for an architecture of like two hidden layers of 600 hundred units per hidden layer (the demonstration was on a "binarised" version of MNIST (where the digits 0, 1, 2, 3, 4 were declared to be a single class and the digits 5, 6, 7, 8, 9 another single class), hence the first instance of a non-vacuous bound for the overparametrised setting where there are a lot many more trainable parameters than training data); and Perez-Ortiz et al. [2020] trained randomised neural net classifiers by several PAC-Bayes methods (including the one used by D&R 2017) and used the PAC-Bayes-kl bound to compute numerical bounds on the risk of these classifiers (like D&R and L&C before), and the novelty of this work is demonstrating *tight* bound values (not just non-vacuous) on full 10-class MNIST and on CIFAR-10, not just feed-forward architectures but also convolutional architectures, and showing that learning the prior on part of the data is key for obtaining tight values of the numerical risk certificates, and called it "self-certified learning" because the whole procedure outputs simultaneously a predictor learned from data and a certificate on the risk of this predictor, hence this work is quite related to your work making similar claims, and they also did come other things like comparing several predictors (fixed, randomised, ensemble, classic ERM, and "Bayes by backprop") and showing that the PAC-Bayes-kl bound can do model selection. [The important point of the work of Perez-Ortiz et al. is that the whole training data was used to learn the predictor, and the certificate was evaluated using part of this training data (the part that was not used to learn the prior), and the value of the certificate is reasonably close to the test set errors (the latter is evaluated on a held-out set), and this tightness indicates that the computed certificates are informative of the post-training performance of the randomised classifiers learned by their methods, hence it makes sense to call it a certificate, unlike a case where you have a non-vacuous numerical bound value of say $\sim 20$ percent for a classifier with test error $\sim$[single digit] percent, in which case the non-vacuous value is uninformative because it is not tight.]

L.141: "This upper bound (Them 3) can be "inverted" to obtain [...]"

L.146: "random dataset"

L.146-147: I am not sure what is meant by these cases of "fixed dataset" and "random dataset" --- as far as I understand, the theorems hold with high probability (of at least $1-\delta$) over the random datasets, hence when you have a fixed instance of a dataset (one realisation of the random dataset, generated by a distribution that meets the requirements of the theorems) you apply the inequality given by the theorem to this fixed dataset, and the reasoning is that this fixed dataset could be one of the bad datasets for which the inequality does not hold, but the probability that it is a bad dataset is small (of at most $\delta$).

L.147: Replace "Next" with "Later"

L.149: "realistic case of a random dataset"



L.151-154 (Theorem 4): This theorem needs to be reformulated according to the intended meaning and

L.156-158 (Remark 2): since the lower bound gives infimum over $\beta>0$, there might exist values of $\beta$ for which $p_{\Delta}$ is less than $p_{C_\beta}$?

L.161-162: This passage is missing some justification for the implicit claim that a one-hidden-layer neural network with positive weights [...] parametrises a general $\Delta \in \mathcal{C}$. Or insert a reference for this?

L.162-163: This describes the same idea used by the numerical inversion of the binary kl which has been used extensively in the PAC-Bayes literature [Langford & Caruana, Ambroladze et al, Parrado-Hernandez et al., many many others]

L.167-168: the claim that this difference converges to 0 from above suggests that there are no $\beta$ for which $p_{\Delta}$ is less than $p_{C_\beta}$? Then if that is true then in fact $p_{\Delta}$ would be at least as large as the sup of $p_{C_\beta}$.

L.170-171: Still it is not clear how the cases of "fixed data" and "random data" give different things.
This suggests at the very least that clarification is needed.

L.172: This line is comparing the average values of the inversions corresponding to different $\Delta$'s, is this legit?

L.178-180: The statement of this Corollary, or some discussion before or after it, should comment on how to make sense of comparing the expectations of the upper bounds, versus comparing the upper bounds for given settings of their components.

Only skimmed at the Proof of Theorem 4 but happy to get back to it between rebuttal and final decision if we get there.
A quick question: Care to give some intuition on why lower-bounding $N^{-1}\mathcal{I}$ and upper-bounding $\Delta$?
At first look it seems to make sense upper-bounding $\mathcal{I}/N$ and lower-bounding $\Delta$, feel free to check me what I misunderstood.

L.230: In "minimise the expected bounds" does this mean minimising the expectation of the bounds, with the expectation over the distribution of the random datasets of a given size? If this is what is meant, some justification is needed for this procedure, taking into account that the PAC-Bayes bounds are high-probability bounds over such distribution.

L. 235-239: This paragraph is missing some references for the meta-learning literature especially the literature based on PAC-Bayes bounds. The works of Maurer, Amit and Meir, and some others might be relevant here.

L.236: This suggests that there is one hypothesis space (i.e. a function class) $\mathcal{H}_{\theta}$ for each possible $\theta$ --- is that correct? Also, the "posterior map" is formally a stochastic kernel (aka Markov kernel).

L.241: The notation $\mathcal{T}$ needs to be declared. Is it intended for denoting the set of all possible tasks?
I think the distribution over tasks should use a different notation than the distribution that generates random dataset for a given task.

L.247: "Perez-Ortiz et al., 2020" is relevant to these references for data-dependent priors gaining attention.

L.249: replace "empirical risk" with "risk bound"

L.250: This is precisely the protocol used by Perez-Ortiz et al. for learning the data-dependent prior say on data $S_1$ and evaluating the risk certificate on dataset $S_2$ while the randomised predictor (the "posterior") is learned on $S = S_1 \cup S_2$. this procedure was used before by Ambroladze et al. and Parrado-Hernandez et al. for SVMs, while Perez-Ortiz et al. were the first to use it for deep learning.

L.251-252: "and Perez-Ortiz et al. demonstrated empirically that learning priors from data is key for obtaining tight risk certificates"
(the latter are much (!) better than non-vacuous bound values)

L.253: The mapping at the end of this line is better formalised as a stochastic kernel.

L.255: this choice to "ease notation" has the effect of obscuring the dependencies --- can it be fixed?

L.261: "learned convex function"

L.267: If $t$ denotes an individual task, and $D_{t}$ the distribution that generates data for this task, then it is clear that the notation $S_t \sim D_{t}^N$ stands for a dataset of size $N$ for task $t$ (but probably $N_t$ should be used for sample size if diferent sizes were to be used for different tasks). However, it is unclear what the notation "$D_t \sim \mathcal{T}$" is trying to communicate. In case, as a few lines above, $\mathcal{T}$ is the set of all possible tasks, and $D \in M_1(\mathcal{T})$ is the distribution according to which one can select a random task, then writing $t \sim D$ is okay, and then writing $S_t \sim D_{t}^N$ for the chosen $t$ (but I think this is missing some conditionals to clearly communicate the order in which the random choices are made, i.e. $D_t$ is in fact a conditional probability distribution, and some regularity must be assumed on the space $\mathcal{T}$ to justify that one can calculate with these distributions in a way that resembles calculations with unconditional distributions).

In general, my impression is that this section could improve a lot by better/explicit math notation and definitions.

L.279: It might be relevant to comment whether the feature space dimension $K$ is set a priori, or it depends on data and is only known a posteriori as in the case of kernel learning (and corresponding RKHS).

L.292-293: This is expected in view of the higher representational power of the CNN compared to MLP?

L.296: Confused by "the test set classifier" --- should this be "the test set bound"? (Or else, explain.)
L.297: The "PAC-Bayes classifier" is the randomised classifier, according to the PAC-Bayes posterior distribution?

L.304-305: The "binomial tail test set bound with a train set proportion 0.4" --- this needs clarification. Is it that the procedure consists of using 40% of the training data for learning a predictor, and the remaining 60% of the data for evaluating the binomial tail test set bound?

Okay, I guess it lakes sense to compare the binomial tail bound (Theorem 1) and the Chernoff bound (Theorem 2) in Fig.4(a) because both these bounds apply to individual predictors $h$, while the separate Fig.4(b) plots the PAC-Bayes bound (is this correct?) which applies to randomised predictors (i.e. distributions over the $h$'s)

L.340: and "Rivasplata et al. [2020]"

L.343: Not sure about the claim that "Gibbs classifiers are not often used in practice" --- could this be substantiated?

Overall this is an interesting/informative discussion in Section 5. however I'd like to flag one thing: It is not clear to me if the analyses carried out in this paper are specific to the small data regime. It if is, this should be highlighted (places in the paper where the analysis holds for small $N$ but not for large $N$). in fact my impression is that the theoretical arguments put forward here might be valid for arbitrary $N$, while the *experiments* were done on small datasets (N=30 and N=60). Interesting paper, regardless.

Would like for all my feedback above to be addressed in the response.


**Time Spent Reviewing:**

6 hours

---

> ### Author Response · Authors · 2021-08-09
> **Response to 4Rmy [1/4]**
>
> Before anything else, we would like to thank you for your extremely detailed and helpful review. We especially appreciate your line-by-line editorial comments, which we will use to further streamline the exposition.
>
> # Revision of Abstract
>
> Thank you for your constructive feedback on the abstract. We’re slightly shortened and reorganised it and incorporated your style suggestion. Please find the revision below.
>
> In this paper, we investigate the question: _Given a small number of datapoints, for example $N = 30$, how tight can PAC-Bayes and test set bounds be made?_ For such small datasets, test set bounds adversely affect generalisation performance by withholding data from the training procedure. In this setting, PAC-Bayes bounds are especially attractive, due to their ability to use all the data to simultaneously learn a posterior and bound its generalisation risk. We focus on the case of i.i.d. data with a bounded loss and consider the generic PAC-Bayes theorem of Germain et al. While their theorem is known to recover many existing PAC-Bayes bounds, it is unclear what the tightest bound derivable from their framework is. For a fixed learning algorithm and dataset, we show that the tightest possible bound coincides with a bound considered by Catoni; and, in the more natural case of distributions over datasets, we establish a lower bound on the best bound achievable in expectation. Interestingly, this lower bound recovers the Chernoff test set bound if the posterior is equal to the prior. Moreover, to illustrate how tight these bounds can be, we study synthetic one-dimensional classification tasks in which it is feasible to meta-learn both the prior and the form of the bound to numerically optimise for the tightest bounds possible. We find that in this simple, controlled scenario, PAC-Bayes bounds are competitive with comparable, commonly used Chernoff test set bounds. However, the sharpest test set bounds still lead to better guarantees on the generalisation error than the PAC-Bayes bounds we consider.
>
> # Editorial Comments
> In what follows, as requested by you, we provide responses to all your comments.
>
> > L.32: " significantly harming post-training performance due to a reduced training set size."
>
> Thank you for this suggested phrasing change. We will make this edit.
>
> > L.33: this sentence calls for a reference, Perez-Ortiz et al. [2020] is relevant here.
>
> We will add an early reference which states that PAC-Bayes holds uniformly over all posteriors, which allows the full dataset to be used during training.
>
> > L.36: As far as I know, the generic PAC-Bayes theorem is originally due to Germain et al. [2009], while others (e.g. Begin et al. [2016] and Rivasplata et al. [2020]) restructured the presentation of the proof or gave extensions to other divergences (Begin et al.) or to setting with relaxed restrictions (Rivasplata et al.) hence it would be more accurate and fair to write something like "remarkably, Germain et al. [2009] showed that a wide range of bounds can be obtained as special cases of a single \emph{generic PAC-Bayes theorem} that captures the central ideas of many PAC-Bayes proofs [see also Begin et al., 2016 and Rivasplata et al., 2020]."
>
> We agree that more directly attributing the theorem to Germain et al [2009] would be more clear about the history of this bound. We had initially cited Begin at all due to the addition of the sup (as you note later) but agree this is a rather minor addition to the proof (though important for numerical computation).
>
> > L.39: "holds for any convex function."
>
> We introduced $\Delta$ in this sentence to allow ourselves to refer to the convex function succinctly in text later in the paragraph. We can separate it with a comma from the sentence if that would improve readability.
>
> > L.40: After "Catoni [2007]" insert "and other bounds"
>
> Thank you for pointing this out. We will make this change.
>
> > L.41: "to this setup."
>
> Set-up is an accepted spelling of this word, especially as a noun, and the document is internally consistent in its use.
>
> > Warning note: Two distinct types of applications of PAC-Bayes bounds are: (1) using such a bound for evaluating a numerical risk certificate for (the randomised predictor defined by) some distribution over hypotheses learned by some method; and (2) using such a bound as optimisation objective by declaring to learn a distribution over hypotheses by minimizing the bound. The second kind of use is possible because PAC-Bayes bounds hold simultaneously (i.e. uniformly) for all distributions over a given hypothesis class. See e.g. the corresponding discussion given by Tolstikhin & Seldin [2013] or by Thiemann et al. [2017]. It appears that this submission is not making a clear distinction between these two uses, and perhaps conflating them when claiming that PAC-Bayes allows to use of all the available data that avoid splitting the data while at the same time getting a certificate on the post-training performance (see corresponding discussion given by Perez-Ortiz et al. [2020]). This must be addressed for the sake of clarity and to avoid misleading follow-up research.
> Thank you for your note regarding that PAC-Bayes can be used either for learning or for evaluation of models. While the two uses are distinct, the second use is naturally followed by the first use: once the bound, which is used as an objective, is minimised, the final objective function value gives a numerical risk bound. This is what we do and what has been done in several previous works that you point to above. We will explicitly state in the text that PAC-Bayes bounds hold for all Gibbs’ predictors, hence can be used for assessing models regardless of whether or not the model was selected by minimising a PAC-Bayes bound.
>
> > L.46-47: This passage needs clarification/elaboration. Risk bounds (aka generalisation bounds, including PAC-Bayes bounds) studied in statistical learning theory are inequalities that hold with high probability with respect to the random choice of datasets (of a given size). Hence reading "fixed dataset" here raises a question mark that begs for clarification. On the other hand, in the typical problems studied by the machine learning community, datasets are fixed (e.g. MINIST, CIFAR-10, ImageNet) and there is the question of whether or under what cases the assumptions of the risk bounds (e.g. i.i.d. data, or at least randomly generated data) are met or can be considered to be reasonable for a given data set so that using the high-probability bounds is justified.
>
> We agree that the phrasing is unusual and in conflict with the probabilistic nature of the bound. What we mean is that the numerical value of the bound is a function of the empirical risk and the KL-divergence (both random quantities). For any fixed empirical risk and KL divergence, there exists a Catoni bound that is at least as tight as any other bound of the form given in equation 3. For a given learning algorithm these quantities are fixed once the dataset is.  Please see a more detailed answer to this in our response to your question on Line 146-147.
>
> > L.48: Replace "stochastic dataset" with "random dataset"
>
> > Reason: "random" refers to something that is picked according to a given fixed probability distribution (e.g. a random digit, or a random matrix of size 10x10) while "stochastic" in my mind refers to a changing probability distribution, e.g. changing with respect to a time parameter (e.g. a Brownian motion) or a space parameter (e.g. a Poisson point process). In your setting, it appears that the assumption on the dataset is that it is generated randomly according to a fixed (but unknown) data-generating distribution. Accordingly, calling it a "random dataset" is preferable. (Alternatively, justify why to call it stochastic.)
>
> We are unable to find a clear reference stating that stochastic implies an index set. For example, the terms “stochastic process” and “random process” are used interchangeably to refer to time-indexed random variables. In general, it seems that these words are used interchangeably (see for example the wikipedia page for stochastic), with the possible difference that stochastic might be favored for a model of something random as opposed to the object itself.
>
> > L.51: A bit surprising to read that a neural network parametrises a convex function.
>
> We could be more clear that the neural network considered may not allow us to parameterise all convex functions, but only some subset. The function output by the network is convex as we use a convex non-linearity and non-negative weights in the top layer, as described in line 161.
>
> > L.52: " bound of Langford and Seeger, and relaxes" (space-saver, since the year has been acknowledged already,)
>
> Thanks. We will make the change.
>
> > L.53: insert a reference for the Chernoff test set bound (or insert "(see Theorem 2 below)" since it provides the reference)
>
> We will include a forward reference to the theorem.
>
> > L.54-60: by the end of reading this paragraph it should be clear to the reader whether you are using bounds for numerical evaluation of certificates only, or if you are using the bounds for learning (by bound-minimisation) and for numerical evaluation of the certificate. In other words, what I missed in this introduction is a high-level description of what kinds of predictors are being considered, for which the certificates on post-training performance are applied.
>
> We use the bounds for both learning (in our case learning a mapping from datasets to Gaussian measures over some parameters of the model) and for evaluating the bound numerically. We will clearly state that learning is performed using the bounds.

---

> ### Author Response · Authors · 2021-08-09
> **Response to 4Rmy [2/4]**
>
> > L.62 (and rest of the paper): I recommend the capital calligraphic and for the sets that encompass all possible inputs and all possible labels (resp.), and for the set of all possible input-label pairs. Then this will be consistent with your use of for the set of all possible hypotheses, etc.
>
> We agree that this notational change will improve notational consistency. We will make this adjustment.
>
> > L.77-78: "For the zero-one loss we have the binomial random variable [..] with parameters [...]" Also, the number of trials should be "" I think?
>
> Yes, thank you for pointing out this typo.
>
> > L.81: the sum should start from 0?
>
> Yes, thank you again for catching this typo.
>
> > L.86: For the same reason explained above, I recommend replacing "stochastic" with "randomised" (regardless of what the PAC-Bayes literature has done before, unfortunately)
>
> See our earlier response.
>
> > L.87: As mentioned above, a claim "does not require discarding data" needs to be supported (clarification or reference(s))
>
> We will clarify that this means that all data can be used for selecting the hypothesis. We will also include a citation.
>
> > L.88-89: "Germain et al. [2009] prove a very general form of the PAC-Bayes theorem which encompasses many of these [see also Begin et al., 2016 and Rivasplata et al., 2020]."
>
> Thank you for the suggested improvement to the citation. We will make the change.
>
> > L.90-21: The part of taking supremum over the risk is special to Begin et al. I think (i.e. Germain et al. [2009] did not do this last step as far as I remember)
>
> Yes, we agree and will be more explicit in citing the several steps of the proof.
>
> > L.96: this theorem needs to clarify that the probability is over the random choice of dataset of size . (i.e. could replace with , since the probability is w.r.t. distribution of the size- random sample)
>
> The only random quantity in the theorem is the dataset and this notation has appeared at various times in the literature. See for example Guedj [2019]. We can make this more explicit by stating in the theorem that the $N$ datapoints in the dataset are sampled i.i.d. from $D$.
>
> > L.97-98: I did not understand this remark. Note that the choice is a possible choice for , which gives the classical form of the PAC-Bayes bound, i.e. the one that resembles the bound of McAllester.
>
> What we intend to say in this remark is that, for any $\Delta$ there exists a $\Delta’$ that is monotonic in the second argument and produces a bound at least as tight as the bound produced by $\Delta$. See Appendix D for a proof. It may be the case that $\mathcal{I}\_{\Delta}$ is easier to evaluate than $\mathcal{I}\_{\Delta’}$. In that case, the construction of $\Delta’$ shows that $\mathcal{I}\_{\Delta’} \leq \mathcal{I}\_{\Delta}$, so we can always use an upper bound on $\mathcal{I}\_{\Delta}$ instead. We conclude that in terms of both tightness and tractability, so we lose no generality in using $\Delta’$ in place of $\Delta$.
>
> > L.99: Germain et al. [2009] discussed how Catoni's bound follows from their generic PAC-Bayes theorem, hence in this case I feel that the attribution to Begin et al. is a bit misleading.
>
> We agree this citation should be fixed. We will make the change.
>
> > L.103-104: "we obtain the bound of Langford and Seeger [2001], also called the PAC-Bayes-kl bound [e.g. Seldin, 2012], but with the slightly sharper dependence on established by Maurer [2004]:"
>
> Thank you for the suggested improvement to phrasing. We will make the change.
>
> > L.105: Note that in fact this inequality holds for all , the small cases have been verified numerically (see "Risk bounds for the majority vote: From a PAC-Bayesian analysis to a learning algorithm")
>
> Thank you for pointing this out. We will add a remark.
>
> > L.106: Why not refer to this, more suggestively, as the PAC-Bayes-kl bound? (I am objecting the use of the acronym MLS due to being uninformative and potentially misleading)
>
> Thank you for the suggestion. We agree and will change “MLS bound” to the more informative “PAC-Bayes-kl bound”.
>
> > L.107: The accurate thing to comment is that Corollary 2 is a special case of Theorem 3 in which the has been upper-bounded by using Stirling's formula, as shown by Maurer [2004]. It is misleading to say that Corollary 2 is looser than Theorem 3, since the theorem is generic and the has not been controlled.
>
> In the case when  $\Delta = \operatorname{kl}$, the supremum is obtained for any $r \in [0,1]$. This leaves a finite sum that can be explicitly evaluated (especially for small $N$) and hence can be controlled. The bound stated in Corollary 2 applies Stirling’s formula to upper bound this sum as shown by Maurer [2004].  This is particularly useful for large $N$.
>
> > L.108-110: Again, the two different uses of PAC-Bayes bounds are being conflated in this passage. Also, the list of references in the square brackets should start with Langford and Caruana 2001 and end with Perez-Ortiz et al., 2020 (Note that the publication year of "(Not) Bounding the True Error" is 2001, regardless that Google Scholar incorrectly gives 2002)
>
> See our response to your earlier note regarding this. Thank you for pointing out the incorrect citation year.
>
> > L.112: [McAllester, 1999, 2003]
>
> Thanks. We will make the change.
>
> > L.123: replace "value" with "form"
>
> Thanks. We will make the change.
>
> > L.132-133: There are significant differences between the three references that are worth bringing to attention: (...) in which case the non-vacuous value is uninformative because it is not tight.]
>
> We appreciate your detailed comment. We agree that each of the above papers had a distinct and significant contribution giving insight into the application of PAC-Bayes to neural networks. The purpose of L.132–133 is to simply point out to the reader that PAC-Bayes has also been applied to neural networks, without going into detail. If space allows, we will try to expand on the differences.
>
> > L.141: "This upper bound (Thm 3) can be "inverted" to obtain [...]"
>
> Thank you. We’ll insert the reference to Theorem 3.
>
> > L.146: "random dataset"
>
> Please see L. 48 above.
>
> > L.146-147: I am not sure what is meant by these cases of "fixed dataset" and "random dataset" --- as far as I understand, the theorems hold with high probability (of at least $1 - \delta$) over the random datasets, hence when you have a fixed instance of a dataset (one realisation of the random dataset, generated by a distribution that meets the requirements of the theorems) you apply the inequality given by the theorem to this fixed dataset, and the reasoning is that this fixed dataset could be one of the bad datasets for which the inequality does not hold, but the probability that it is a bad dataset is small (of at most $\delta$).
>
> The high-probability upper bound $\overline{p}_\Delta$ from (6) can be viewed as a map from $\Delta$, $S$, $P$, and $Q$ to the definition of $\overline{p}_\Delta$:
>
>  $$\overline{p}_\Delta(S, P, Q) := B[\Delta(\overline{R}_S(Q),\cdot), \frac1N(\operatorname{KL}(Q\|P)+ \log\frac{\mathcal{I}_\Delta(N)}{\delta})].$$
>
> When viewed as a map like this, we can study the dependence of the map $\overline{p}_\Delta(S, P, Q)$ on its four arguments; and we can do this study without any regard for the validity of the resulting generalisation bound. In particular, in Theorem 4, we fix the arguments $S$, $P$, and $Q$ to some arbitrary data set, prior, and posterior; and, for those arguments, we attempt to minimise the quantity $\overline{p}_\Delta(S, P, Q)$ with respect to the convex function $\Delta$. Because we fixed the argument $S$, the value for $\Delta$ which numerically minimises $\overline{p}_\Delta$ depends on $S$. As we point out in L. 147, this is not allowed: this “best” choice for $\Delta$ does not necessarily yield a valid generalisation bound. It does, however, give a lower bound on the tightest possible generalisation bound, which leads to Cor. 3.
>
> We understand that investigating the dependence of $\overline{p}_\Delta$ on $\Delta$, $S$, $P$, and $Q$ without any regard for the validity of the bound is uncommon. Since “fixing a dataset” was also confusing for R1, we will clarify the writing and make it unmistakingly clear what precisely we’re doing.
>
> > L.147: Replace "Next" with "Later"
>
> Thank you. We’ll make the edit.
>
> > L.149: "realistic case of a random dataset"
>
> Thank you. We’ll make the edit.
>
> > L.151-154 (Theorem 4): This theorem needs to be reformulated according to the intended meaning and
>
> Your comment here appears unfinished. Could you please clarify? For now, we’ll assume that your comment was intended to pertain to your comment on L. 146–147 above, confusion surrounding the phrase “fixed dataset”. In line with the clarifications there, we will also adjust the theorem statement to resolve any confusion about the procedure of “fixing $S$, $P$, and $Q$”.
>
> > L.156-158 (Remark 2): since the lower bound gives infimum over $\beta > 0$, there might exist values of $\beta$ for which $p_\Delta$ is less than $p_{C_\beta}$?
>
> Certainly! Looking at (4), if $\overline{R}_S(Q) > 0$, then the Catoni bound goes to one as $\beta$ goes to infinity.
>
> > L.161-162: This passage is missing some justification for the implicit claim that a one-hidden-layer neural network with positive weights [...] parametrises a general $\Delta \in \mathcal{C}$. Or insert a reference for this?
>
> We are not aware of a theorem which guarantees that our proposed neural network is able to approximate any convex function. Moreover, for our experiments, we do not rely on this claim: using our architecture, we have been able to find convex functions which perform well; the true infimum can only do better.
> We will remove the word “general” from L. 161.

---

> ### Author Response · Authors · 2021-08-09
> **Response to 4Rmy [3/4]**
>
> > L.162-163: This describes the same idea used by the numerical inversion of the binary kl which has been used extensively in the PAC-Bayes literature [Langford & Caruana, Ambroladze et al, Parrado-Hernandez et al., many many others]
>
> Thank you. We’ll add the references. We also note that our method is distinct from a common method for inversion of the binary kl using Newton’s algorithm.
>
> > L.167-168: the claim that this difference converges to 0 from above suggests that there are no $\beta$ for which $p_{\Delta}$ is less than $p_{C_\beta}$? Then if that is true then in fact $p_{\Delta}$ would be at least as large as the sup of $p_{C_\beta}$.
>
> As we point out in L. 169–170, Theorem 4 implies that $\inf\_{\Delta \in \mathcal{C}} \overline{p}\_\Delta = \inf\_{\beta > 0} \overline{p}\_{C\_\beta}$, which is what we’re evidencing in Figs 2(a) and 2(b).
>
> > L.170-171: Still it is not clear how the cases of "fixed data" and "random data" give different things. This suggests at the very least that clarification is needed.
>
> Please see L. 146–147 above.
>
> > L.172: This line is comparing the average values of the inversions corresponding to different $\Delta$'s, is this legit?
>
> We compared the minimum and maximum over ten runs to diagnose the behaviour of the optimisation problem: Was there one lucky run? Was there one run suffering from numerical issues which produced crazy values? Did all runs behave reasonably and similarly? To reduce down the 10 runs to a single run, which is a useful visual aid, we decided to use the mean value. Note that the minimum and maximum over 10 runs appear to converge on the right sides of the plots, so any other statistic (median) could also have been used.
>
> > L.178-180: The statement of this Corollary, or some discussion before or after it, should comment on how to make sense of comparing the expectations of the upper bounds, versus comparing the upper bounds for given settings of their components. <newline> Only skimmed at the Proof of Theorem 4 but happy to get back to it between rebuttal and final decision if we get there. A quick question: Care to give some intuition on why lower-bounding $N^{-1} \mathcal{I}$ and upper-bounding $\Delta$? At first look it seems to make sense upper-bounding $\mathcal{I}/N$ and lower-bounding $\Delta$, feel free to check me what I misunderstood.
>
> From the definition of $B[f, y]$ (L. 140), $B[f_1, y_1] \le B[f_2, y_2]$ if $f_1 \ge f_2$ and $y_1 \le y_2$. In Theorem 4, we fix $\Delta$ and find a Catoni bound which is tighter, which we achieve by upper bounding the first argument, $\Delta$  ($f_1$), and lower bounding the second argument $\mathcal{I}$ ($y_1$). This property of $B$ is used in (17).
>
> > L.230: In "minimise the expected bounds" does this mean minimising the expectation of the bounds, with the expectation over the distribution of the random datasets of a given size? If this is what is meant, some justification is needed for this procedure, taking into account that the PAC-Bayes bounds are high-probability bounds over such distribution.
>
> Yes, that’s right. For a justification, we refer to the section “The paper studies minimizing high-probability PAC-Bayes bounds in expectation. Shouldn’t we be using bounds on the expected generalization error?” in Appendix J of “On the Role of Data in PAC-Bayes Bounds” by Dziugaite et al. (2020) (https://arxiv.org/pdf/2006.10929.pdf). We will add a reference in the writing.
>
> > L. 235-239: This paragraph is missing some references for the meta-learning literature especially the literature based on PAC-Bayes bounds. The works of Maurer, Amit and Meir, and some others might be relevant here.
>
> The way meta-learning is applied in our work contrasts with previous PAC-Bayes meta-learning papers. We explain this on lines 273-276 (where we also cite Amit and Meir, along with more recent PAC-Bayes meta-learning works such as Rothfuss et al 2020, Liu et al 2021 and Farid and Majumdar 2021). The key difference is that in those works, PAC-Bayes bounds are applied to the meta-learning problem itself. That is, in those works the tasks are viewed as being generated iid from a task generating distribution. The generalisation performance of the meta-learner is defined as its performance on new tasks sampled from the task-generating distribution. We note that Maurer 2005, “Algorithmic Stability and Meta-Learning”, falls into this category: on the second page they state “This paper describes a mechanism by which… samples, drawn from different learning tasks… can be used to improve and predict the
> performance of a learner on an unknown future task” (although, we are happy to provide a citation in order to contrast their setting from ours).
>
> In contrast to all these papers, in our work we consider the setting where we seek generalisation bounds for each task separately, where the datapoints in each task are generated iid from their respective task distributions. We do not use PAC-Bayes to bound generalisation performance on new tasks. Rather, for each individual task, the meta-learner outputs a task-specific classifier. We then use PAC-Bayes to bound the generalisation performance for each such classifier separately. What we report is then the average generalisation bounds obtained in this way over many tasks.
>
> > L.236: This suggests that there is one hypothesis space (i.e. a function class) H_\theta for each possible \theta --- is that correct? Also, the "posterior map" is formally a stochastic kernel (aka Markov kernel).
>
> This is correct, for each $\theta$ there is a corresponding hypothesis space. Hence our meta-learners can learn distinct hypothesis spaces for each bound. Note that this still leads to completely valid generalisation bounds for each dataset when the meta-learner is shown a dataset in the meta-test set. Please see “Clarification of Section 4” in our response to 15BA. Given the parameters $\theta$ that we have obtained during training, our meta-learning setup on a new dataset is identical or very similar to other applications of data-dependent priors (that have been formalised more carefully). Thank you for pointing out that the posterior map is formally a Markov kernel, although we felt that the introduction of measure-theoretic terminology was not necessary for clarity in this line (in line with our remark in footnote 1).
>
> > L.241: The notation T needs to be declared. Is it intended for denoting the set of all possible tasks? I think the distribution over tasks should use a different notation than the distribution that generates random dataset for a given task.
>
> Thank you for pointing this out. $\mathcal{T}$ here is defined as a distribution (ie, a probability measure) over tasks. A task is viewed as a distribution that generates random datasets. Formally speaking, each task $D \in \mathcal{M}_1(Z)$ is a probability measure over $Z = X \times Y$. Here $\mathcal{M}_1(Z)$ is the set of all probability measures over $Z$. $\mathcal{T}$ is then the distribution over tasks, ie, a probability measure over $\mathcal{M}_1(Z)$, with an appropriate sigma-algebra.
>
> > L.247: "Perez-Ortiz et al., 2020" is relevant to these references for data-dependent priors gaining attention.
>
> Thanks for pointing this out, we will add this citation to point out the recent attention given to data-dependent priors in stochastic deep neural networks.
>
> > L.249: replace "empirical risk" with "risk bound"
>
> Thanks, we will make this change.
>
> > L.250: This is precisely the protocol used by Perez-Ortiz et al. for learning the data-dependent prior say on data $S_1$ and evaluating the risk certificate on dataset $S_2$ while the randomised predictor (the "posterior") is learned on $S = S_1 \cup S_2$. this procedure was used before by Ambroladze et al. and Parrado-Hernandez et al. for SVMs, while Perez-Ortiz et al. were the first to use it for deep learning.
>
> Thanks for pointing this out. In this paper we were not applying PAC-Bayes to stochastic deep neural networks, as is done in Perez-Ortiz et al (our meta-learners output distributions over the weights of linear models with non-linear features). However, we can add this citation here for context.
>
> > L.251-252: "and Perez-Ortiz et al. demonstrated empirically that learning priors from data is key for obtaining tight risk certificates" (the latter are much (!) better than non-vacuous bound values)
>
> Thanks, we will add this observation here.
>
> > L.253: The mapping at the end of this line is better formalised as a stochastic kernel.
>
> Thanks for the suggestion. Please see the comment on line 236 above.
>
> > L.255: this choice to "ease notation" has the effect of obscuring the dependencies --- can it be fixed?
>
> Yes we can adopt a more explicit notation. We can subscript the parameters theta so that the parameters that define the prior map are denoted $\theta_P$ and the parameters that define the posterior map are denoted $\theta_Q$. Then, during meta-training, both of these sets of parameters $\{ \theta_P, \theta_Q \}$ are optimised together.
>
> > L.261: "learned convex function"
>
> Thanks, we’ll make this change.

---

> ### Author Response · Authors · 2021-08-09
> **Response to 4Rmy [4/4]**
>
> > L.267: If  denotes an individual task, and  the distribution that generates data for this task, then it is clear that the notation stands for a dataset of size  for task  (but probably should be used for sample size if diferent sizes were to be used for different tasks). However, it is unclear what the notation "" is trying to communicate. In case, as a few lines above,  is the set of all possible tasks, and is the distribution according to which one can select a random task, then writing  is okay, and then writing for the chosen  (but I think this is missing some conditionals to clearly communicate the order in which the random choices are made, i.e.  is in fact a conditional probability distribution, and some regularity must be assumed on the space  to justify that one can calculate with these distributions in a way that resembles calculations with unconditional distributions). In general, my impression is that this section could improve a lot by better/explicit math notation and definitions.
>
> Thanks for this suggestion. As noted in the comment on line 241 above, $\mathcal{T}$ does not denote the set of all possible tasks, but rather a distribution over distributions over datapoints (ie, a meta-data distribution). However, we could also use the notation you suggest, which can be used to convey the same sampling procedure. Specifically, we sample a data-generating distribution $D_t$ from $\mathcal{T}$, and then sample a random dataset of size $N$ from that data-generating distribution. We repeat this $T=80000$ times to define the objective in Equation 20. We specify the detailed sampling procedure for generating tasks, and datapoints from those tasks, in Appendix I.1.
>
> > L.279: It might be relevant to comment whether the feature space dimension $K$ is set a priori, or it depends on data and is only known a posteriori as in the case of kernel learning (and corresponding RKHS).
>
> $K$ is set a priori. We will state this explicitly.
>
> > L.292-293: This is expected in view of the higher representational power of the CNN compared to MLP?
>
> This is expected in light of the fact that the convolutional architecture has been shown empirically in the past to provide superior performance for 1D regression meta-learning experiments compared to MLP-based architectures. This is shown in the cited papers Gordon et al., 2020, Foong et al., 2020 and Bruinsma et al., 2021. We will state this explicitly.
>
> > L.296: Confused by "the test set classifier" --- should this be "the test set bound"? (Or else, explain.)
>
> No, this refers to the test-set classifier itself returning deterministic predictions. In Section 4, we train a variety of meta-learners. Each meta-learner is trained to optimise, in expectation, the value of a certain generalisation bound. Please see “Clarification of Section 4” in our response to 15BA. We observed that when training the meta-learner that optimises the test-set bound, the resulting learned predictions are nearly deterministic (hence the sharp decision boundaries in Figure 4a). This is in contrast to the meta-learners that optimise PAC-Bayes bounds. Appendix I.2 provides a further discussion of this.
>
> > L.297: The "PAC-Bayes classifier" is the randomised classifier, according to the PAC-Bayes posterior distribution?
>
> Yes, that is correct.
>
> > L.304-305: The "binomial tail test set bound with a train set proportion 0.4" --- this needs clarification. Is it that the procedure consists of using 40% of the training data for learning a predictor, and the remaining 60% of the data for evaluating the binomial tail test set bound?
>
> Yes, that is correct. We will state this more explicitly.
>
> > Okay, I guess it lakes sense to compare the binomial tail bound (Theorem 1) and the Chernoff bound (Theorem 2) in Fig.4(a) because both these bounds apply to individual predictors , while the separate Fig.4(b) plots the PAC-Bayes bound (is this correct?) which applies to randomised predictors (i.e. distributions over the 's)
>
> This is almost correct. What we do to train the test-set meta-learners is use the setup described on line 279: that is, the meta-learner outputs a Gaussian distribution over the weights of the predictors. We do not constrain this to be deterministic. However, during the course of training by optimising the test-set bound on the meta-train set, we find that the classifier learns to become nearly deterministic (which is why Figure 4a shows hard decision boundaries). Hence the freedom to learn stochastic classifiers is unimportant for test-set bounds (whereas it is crucial for PAC-Bayes bounds, as we can see in Figure 4b that both the prior and posterior classifiers are stochastic, leading to soft decision boundaries). This is discussed in further detail in Appendix I.2. Another reason why we plot the Chernoff and binomial tail test set classifiers together is that they learn the exact same classifier. This can be seen by noting that, for both these test-set bounds, the meta-learner that optimises these bounds should simply learn to minimise the test risk on each dataset, since the bounds are monotonic in the test risk (see lines 262-263, 271).
>
> > L.340: and "Rivasplata et al. [2020]"
>
> Thanks, we will add this citation.
>
> > L.343: Not sure about the claim that "Gibbs classifiers are not often used in practice" --- could this be substantiated?
>
> Our point here is simply that Gibbs classifiers are less commonly used by practitioners than deterministic classifiers (e.g., most feed-forward supervised neural networks). For example, there is a line of work on derandomising PAC-Bayes bounds for this very reason (see Viallard et al 2021, “A General Framework for the Derandomization of PAC-Bayesian Bounds” for a recent example). We are happy to change the wording to “Gibbs classifiers are less commonly used than deterministic classifiers in practice".
>
> > Overall this is an interesting/informative discussion in Section 5. however I'd like to flag one thing: It is not clear to me if the analyses carried out in this paper are specific to the small data regime. It if is, this should be highlighted (places in the paper where the analysis holds for small  but not for large N). in fact my impression is that the theoretical arguments put forward here might be valid for arbitrary N, while the experiments were done on small datasets (N=30 and N=60). Interesting paper, regardless.
>
> Thank you for your comments. You are right in saying that the theoretical analysis holds for any N, and only the experiments are done for small N. However, as we state on line 34, the small data setting is such that lower-order terms in PAC-Bayes bounds become more important, relatively. For example, the difference between the MLS bound and the optimistic MLS bound is a term that is O(log N /N), which goes to zero as N goes to infinity. Hence our theoretical results hold for all N, but are most important when N is small.

---

> ### Comment · Reviewer_4Rmy · 2021-09-03
> **post-rebuttal update**
>
> Many thanks for your responses to my feedback.
>
> My review and score of 7 were of course based on the paper that was submitted alone (no other information). The author responses have addressed all my concerns and questions in a satisfactory way, and have outlined how the paper will be updated. After having a look at the other reviews, and the discussions with the other reviewers, it seems that the authors have responded satisfactorily to their feedback, and also outlined how they will update their paper in view of the feedback. While a major concern pointed out by one reviewer has been cleared. In view of all of this, and the updates that the authors have proposed to carry out on their work, I believe that the final camera ready version is going to be a high quality paper. This is on top of the good features of this work that I commented on. Therefore I increase my score, as a higher score represents my updated belief on this work in view of the author responses and the discussions with the other reviewers.

---

### Official Review · Reviewer_tRQX · 2021-07-20

**Rating:** 7
**Confidence:** 3

**Summary:**

This paper proposes to study the best numerical tightness of PAC-Bayes bounds, it comparison with test set bounds. The main motivation given is that in small sample problems one cannot afford to hold out a subset of the data to compute a test set bound.

The main theoretical results are Theorem 4 with its Corollary 3, presented in Sec. 3. In Thm 4 the dataset is fixed, and the best divergence function is sought for the general PAC-Bayes bound, which produces the tightest bound. This turns out to be one of Catoni's bounds. In Cor 3 the data set is no longer fixed, and the authors find that Catoni's bounds can no longer achieve the tightest bound, and the authors conjecture the "optimistic MLS bound" is a tight lower bound, which they show to be a tight lower bound. Apparently the conjecture was previously stated by Langford. While the present paper still leaves this open, it makes some progress in terms of insights.
The theoretical investigation is complemented with numerical experiments, which also add insight.

Furthermore, Sec 4. presents extensive experimental work addressing the question of which PAC-Bayes bounds are tightest and how they compare with the test set bound, by optimising the bounds numerically, with respect to the algorithm and the task. Results from the experiments are presented in Figs 3 and 4, and several interesting observations are made, including that PAC-Bayes can be as tight as the test-set bound, and that the "optimistic MLS bound" is indistinguishable from Catoni's. While it is indeed interesting to see that PAC-Bayes bounds can be really tight, of course it doesn't mean they will be tight when applied on some given task and some given algorithm.

The main outcomes of the study are summarised in Sec 5. in the for of 2 open problems: 1. echoes the one of Langford, 2. asks for the possibility that a PAC-Bayes bound be as tight (at its best) as the tightest set set bound.


**Limitations And Societal Impact:**

Yes.

**Main Review:**

Originality: While the bounds studied here are not new, and the open problems that result from the study are also existing ones, the questions formulated and the methodology to pursue them are original. It is one of those studies where "the journey is more important than the destination". It is insightful to see those connections and limitations of the PAC-Bayes methodology, despite there is no better alternative is being proposed. In addition, the study also has pedagogical value as it could easily serve to attract newcomers to the field.

Quality: The study is well designed, and rigorous. The paper is well written. I found Sec 3 stronger than Sec 4. The latter seems more exploratory.

Clarity: The presentation is mostly clear, except few places.
- I wasn't sure about the pairs of plots in Fig 3 - is the first plot the upper bound on the second, and the third plot the upper bound on the fourth? If so, my concern would be not so much about numerical tightness but the differences in behaviour - eg. generalisation risk decreasing when the bound is decreasing? I suspect I misunderstand something regarding this figure, could the authors clarify?
- the reduction of PAC-Bayes to test set bound should be explained with more care because the purpose of these two types of bounds is quite different and the latter uses a test set whereas the former doesn't.
- In statement of Thm 4 "a fixed dataset" - do you mean any fixed dataset? or one particular dataset?
- on page 4, "discretising the inputs to \Delta": \Delta takes 2 arguments, so please clarify what is meant here? Sam in Thm 4: \Delta convex *in its 2nd argument* - right?
- in Sec 4 when parameterising the meal-learner and hypothesis space, and taking Q_{theta} and P_{theta} to be Gaussian, then I think the results be specific to this choice of distributions? If so, is there anything general that we can learn from the results?
- instead of "optimistic MLS" I'd suggest "wishful MLS" (reason being that optimistic bounds normally refer to bounds that exhibit a faster convergence rate).
- I don't quite agree with the statement that the "optimistic MLS" can quantify potential slack - because, as the authors rightly point out, it is not yet proven to be a valid generalisation bound.
- on page 5, "Catoni's bound is sometimes nearly optimal" sounds very sounds like a formal claim, but if I got it right here it refers to a visual assessment of the experimental figure?

Significance:
- The study is exploratory, and provides interesting insights about and connections between state-of-the-art generalisation bounds, but no definite conclusions or improvements of the bounds or bounding techniques. The main outcome of the study is echoing 2 open problems.
Nevertheless the focus on state-of-the-art bounds and the timely questions investigated have good pedagogical value that can easily serve as entry point for newcomers to the field.
- The study is only concerned with numerical tightness of the bounds, not the (increasing/decreasing) behaviour, which is more important for the bounds to be useful in practice.
- As we know, the PAC-Bayes bounds assume a Gibbs classifier, while we sometimes want to apply it to classifiers in the frequentist framework to classifiers that don't have uncertainly in their parameters. Could the authors comment on how to reconcile this?

**Time Spent Reviewing:**

30

---

> ### Author Response · Authors · 2021-08-09
> **Response to tRQX**
>
> Thank you for taking the time to review our paper. We appreciate that you noted the originality of our methods and questions, and also the rigour and pedagogical contributions of our study. We agree that the experimental section of our paper is mostly exploratory. Our intention there was to gain insight in a simple, controlled setting, which nevertheless complements our theoretical results.
>
> # Significance
>
> You have pointed out that we do not definitively improve bounds or bounding techniques. However, the goal of our paper is primarily to investigate the relationships between bounds and provide insight. We believe that the community can benefit significantly from careful, well-scoped studies like these, and that the paper can serve as an example in this respect, as noted by reviewer 4Rmy. If you agree, we hope you will consider increasing your score.
>
> > “The study is only concerned with numerical tightness of the bounds, not the (increasing/decreasing) behaviour, which is more important for the bounds to be useful in practice.”
>
> We believe numerical tightness can be a very important practical consideration for generalisation bounds, and indeed there is a line of work that emphasizes the importance of numerically non-vacuous bounds, which we mention in Section 2. John Langford’s thesis as well as Dziugaite and Roy, 2017 provide examples of works where the majority of the interest in the work is in the numerical properties of the bounds.
>
> Could you please clarify what you mean by “(increasing/decreasing) behaviour” of the bounds? For an explanation of the behaviour in Figure 3, please also see our response on this in the “Clarity” section below.
>
> > “PAC-Bayes bounds assume a Gibbs classifier, while we sometimes want to apply it to classifiers in the frequentist framework to classifiers that don't have uncertainty in their parameters.”
>
> This is a good point, and one that we acknowledge (along with many other possible extensions) in line 343. There is a body of work that seeks to derandomise PAC-Bayes bounds for this setting (eg, Viallard et al, A General Framework for the Derandomization of PAC-Bayesian Bounds, 2021 is a recent example). It would be interesting to see if our results could be combined with these. However, as most of PAC-Bayes literature considers Gibbs risk, we believe an analysis of the ‘classical’ setting is the right place to start, and we defer extensions such as these to future work.
>
> # Clarity
>
> > “I wasn't sure about the pairs of plots in Fig 3 - is the first plot the upper bound on the second, and the third plot the upper bound on the fourth?”
>
> Yes, this is correct: generalisation bounds in the figure seek to bound the generalisation risks in the figure with high probability. The second and fourth plots show a Monte Carlo estimate of the generalisation risk computed on unseen data.
>
> > “my concern would be not so much about numerical tightness but the differences in behaviour - eg. generalisation risk decreasing when the bound is decreasing? “
>
> The models trained to optimise the test set bound show decreasing risk as the train set proportion is increased. This makes sense because when the train set has more training examples, the classifier is expected to be more accurate. However, when more data is used for the train set, there is less data available for the test set, which makes the test set bounds looser. This shows the tradeoff that test set bounds require - a tighter bound requires using a larger test set, and hence a smaller train set, which means increased generalisation risk.
>
> > “I suspect I misunderstand something regarding this figure, could the authors clarify?”
>
> In this section we train meta-learners that map small 1D regression datasets directly to classifiers. See also “Clarification of Section 4” in our response to 15BA. Each meta-learner is trained to output a classifier that, on average, minimises the expected value of the corresponding generalisation bound. After meta-training, we can evaluate the average risks of these classifiers (2nd and 4th plot) and their corresponding generalisation bounds (1st and 3rd plot) on the held out meta-test set.
>
> > “the reduction of PAC-Bayes to test set bound should be explained with more care because the purpose of these two types of bounds is quite different and the latter uses a test set whereas the former doesn't.”
>
> When we claim that the PAC-Bayes bounds reduce to a test set bound, we mean that when the posterior is chosen to be the (perhaps data-dependent) prior, the bound is identical to the test set bound using the Gibbs classifier associated to this prior.
>
> It is true that test set bounds and PAC-Bayes bounds can be used differently. However, in this paper we are concerned with the question: if we have a fixed number of datapoints, how can we get the tightest bounds? Is it better to split off a test set and use a test set bound, or use the entire dataset together to form a PAC-Bayes bound? Hence we are using both bounds for the same purpose: to get the best possible generalisation guarantees with a fixed number of datapoints. One can see Langford, Tutorial on Practical Prediction Theory for Classification, 2005, Figure 11 for an example of a similar comparison.
>
> > “In statement of Thm 4 "a fixed dataset" - do you mean any fixed dataset? or one particular dataset?”
>
> We mean any fixed dataset. See also L.146–147 in our response to 4Rmy.
>
> > “on page 4, "discretising the inputs to $\Delta$": $\Delta$ takes 2 arguments, so please clarify what is meant here?
>
> The procedure is explained in detail in Appendix F. We seek to compute equation 6, which involves a supremum over the second argument of $\Delta$. It is this second argument that we discretise. We will clarify this by saying “discretising the second argument of $\Delta$” in line 163.
>
> > “Sam in Thm 4: $\Delta$ convex in its 2nd argument - right?”
>
> $\Delta$ is jointly convex in both of its arguments. We can clarify this by explicitly stating joint convexity in the definition of $\mathcal{C}$ in line 72. Joint convexity is required for the application of Jensen’s inequality in the proof of the generic PAC-Bayes theorem, see equation 27 in appendix B.
>
> > “​​in Sec 4 when parameterising the meta-learner and hypothesis space, and taking Q_{theta} and P_{theta} to be Gaussian, then I think the results be specific to this choice of distributions? If so, is there anything general that we can learn from the results?”
>
> You are right in pointing out that we lose some generality for the sake of tractability in specialising to the case of Gaussians. However, we mainly aim to compare the various bounds on an even footing with each other in a controlled setting, while still allowing meta-learning to learn classifiers that are specialised to each bound. We will discuss this in the “Limitations” section. It is worth noting that for the MLP-NP the feature map \phi_{theta} itself is represented by a flexible non-linear function approximator (see appendix I.5).
>
> > “instead of "optimistic MLS" I'd suggest "wishful MLS" (reason being that optimistic bounds normally refer to bounds that exhibit a faster convergence rate).”
>
> Thanks for pointing this out - we will refer to the bound as “conjectured” instead of “optimistic”.
>
> > “I don't quite agree with the statement that the "optimistic MLS" can quantify potential slack - because, as the authors rightly point out, it is not yet proven to be a valid generalisation bound.”
>
> We agree that optimistic MLS is not a proven bound. However, if a practitioner computes a valid PAC-Bayes bound based on the generic PAC-Bayes theorem, and finds that it is close to the optimistic MLS bound, they can be assured by corollary 3 that they would not have gotten a much better bound (in expectation) with any other choice of $\Delta$. We will change the sentence to say that the optimistic MLS bound can be used to prove optimality of a choice of $\Delta$.
>
> > “on page 5, "Catoni's bound is sometimes nearly optimal" sounds very sounds like a formal claim, but if I got it right here it refers to a visual assessment of the experimental figure?”
>
> Do you mean line 300 on page 8? What we mean there is that the Catoni bounds have empirically been observed, in this experiment, to be close to the conjectured MLS bound. We do not intend to make a formal claim. We will clarify this by saying “Empirically this suggests that, in light of Corollary 3, one of the Catoni bounds may be very nearly optimal among all convex functions for this task distribution”.

---

> > ### Comment · Reviewer_tRQX · 2021-08-27
> > **clarification**
> >
> > "Could you please clarify what you mean by “(increasing/decreasing) behaviour” of the bounds?"
> >
> > Data-dependent bounds could be used in practice to select hyperparameters (even without a holdout set in the case of train-set bounds). For this, what we need is that the minimum of the bound occurs at a value of the hyperparameter where the generalisation error is also small (at its minimum, ideally). This doesn't require numerical tightness but it requires that the gap between bound and true error doesn't fluctuate much.

---

> > > ### Author Response · Authors · 2021-08-30
> > > **Response to Clarification**
> > >
> > > > For this, what we need is that the minimum of the bound occurs at a value of the hyperparameter where the generalisation error is also small (at its minimum, ideally). This doesn't require numerical tightness but it requires that the gap between bound and true error doesn't fluctuate much.
> > >
> > > In general, the minimum [maximum] of a[n] upper [lower] bound on a function can be widely different from the minimum [maximum] of the function it bounds.
> > >
> > > Nevertheless, besides hyperparameter selection, we think numerically tight generalisation bounds are important in applications where you simply require low generalisation error (e.g., "I need my generalisation error to be below 30%").

---

### Author Response · Authors · 2021-08-09
**Response to All Reviewers**

The authors would like to thank all reviewers for their careful assessment of our writing and for providing helpful comments. We are pleased that tRQX finds that “Sec 4. presents extensive experimental work”; that 15BA thinks that “Theorem 4/Corollary 3 are exciting theoretical results, of use to researchers (...)”.

Several reviewers felt a weakness of the paper was that the contribution was incremental/exploratory. While we agree that the paper neither directly gives novel bounds nor a state-of-the-art application, it presents a carefully scoped study with novel theoretical results and original experimental methodology on which future work can be built. This was noted by 4Rmy, who is “leaning to say that this work is likely to generate fruitful discussions and follow-up works”, admires the effort to balance theory and experiments, and thinks that the paper “could have a positive effect of leading by example”. Moreover, tRQX thinks that “the focus on state-of-the-art bounds and the timely questions investigated have good pedagogical value that can easily serve as entry point for newcomers to the field.” We believe that carefully scoped papers such as this can be of great use to the community.

Overall, there also seemed to be two points of confusion: (1) the use of “fixed dataset”, see “L. 146–147” in our response to 4Rmy; and (2) the setup of Sec. 4, see “Clarification of Section 4” in our response to 15BA.

---

### Decision · Program_Chairs · 2021-09-27

**Decision:**

Accept (Poster)

**Comment:**

This work deepens our understanding of PAC-Bayes bounds. While the study is mostly exploratory, and perhaps unconventional for a NeurIPS paper, the reviewers believe that there is clear value to the community, as do I. The connections studied in this work are interesting and have pedagogical value, and it is quite likely that interesting discussions will come out of this work. While the open questions/problems are not yet resolved, the understanding gained may lead us closer to their being resolved. In more detail, Theorem 4 is quite interesting (and one reviewer mentioned that the proof checks out). Especially from the discussion phase, all reviewers support acceptance of this paper. This paper will make for an interesting contribution to the NeurIPS proceedings, and I hope serve as a great reference for future works as well.